# E-selectin-mediated rapid NLRP3 inflammasome activation regulates S100A8/S100A9 release from neutrophils via transient gasdermin D pore formation

Monika Pruenster[1,8], Roland Immler [1,8], Jonas Roth[1], Tim Kuchler [1], Thomas Bromberger[2], Matteo Napoli [1], Katrin Nussbaumer[1], Ina Rohwedder[1], Lou Martha Wackerbarth[1], Chiara Piantoni [1], Konstantin Hennis [1], Diana Fink[3], Sebastian Kallabis[3], Tobias Schroll [1], Sergi Masgrau-Alsina[1], Agnes Budke[1], Wang Liu[1], Dietmar Vestweber [4], Christian Wahl-Schott[1], Johannes Roth [5], Felix Meissner [3], Markus Moser[2], Thomas Vogl [5], Veit Hornung [6], Petr Broz [7] & Markus Sperandio [1] ✉

S100A8/S100A9 is a proinflammatory mediator released by myeloid cells during many acute and chronic inflammatory disorders. However, the precise mechanism of its release from the cytosolic compartment of neutrophils is unclear. Here, we show that E-selectin-induced rapid S100A8/S100A9 release during inflammation occurs in an NLRP3 inflammasome-dependent fashion. Mechanistically, E-selectin engagement triggers Bruton's tyrosine kinase-dependent tyrosine phosphorylation of NLRP3. Concomitant potassium efflux via the voltage-gated potassium channel $K_V1.3$ mediates ASC oligomerization. This is followed by caspase 1 cleavage and downstream activation of pore-forming gasdermin D, enabling cytosolic release of S100A8/S100A9. Strikingly, E-selectin-mediated gasdermin D pore formation does not result in cell death but is a transient process involving activation of the ESCRT III membrane repair machinery. These data clarify molecular mechanisms of controlled S100A8/S100A9 release from neutrophils and identify the NLRP3/gasdermin D axis as a rapid and reversible activation system in neutrophils during inflammation.

Neutrophils comprise the largest fraction of circulating white blood cells in humans and are ready to sense infectious or sterile challenges in damaged tissue, thereby functioning as the first line of defense during the inflammatory response[1]. Activation of neutrophils triggers the secretion of intracellular components, including damage-associated molecular pattern molecules (DAMPs), to the extracellular microenvironment. S100A8/S100A9, also known as calprotectin or MRP8/MRP14, is such a DAMP that modulates immune responses after its release

[1]Walter Brendel Centre of Experimental Medicine, Biomedical Center, Institute of Cardiovascular Physiology and Pathophysiology, Ludwig-Maximilians-Universität München, Planegg-Martinsried, Germany. [2]Institute of Experimental Hematology, School of Medicine, Technical University Munich, Munich, Germany. [3]Department of Systems Immunology and Proteomics, Institute of Innate Immunity, Medical Faculty, University of Bonn, Bonn, Germany. [4]Max Planck Institute for Molecular Biomedicine, Münster, Münster, Germany. [5]Institute of Immunology, University of Münster, Münster, Germany. [6]Gene Center and Department of Biochemistry, Ludwig-Maximilians-Universität München, Munich, Germany. [7]Department of Immunobiology, University of Lausanne, Epalinges, Switzerland. [8]These authors contributed equally: M. Pruenster, R. Immler. ✉e-mail: markus.sperandio@lmu.de

from the cytosol of myeloid cells[2]. In human neutrophils, S100A8/S100A9 represents about 40% of the total cytosolic protein content, rendering these cells the major source of this alarmin. Passive release of S100A8/S100A9 into the extracellular space is evoked through tissue damage and is also observed during NETosis. This is in contrast to active secretion triggered by cell stress that operates independently of cell death[2]. Neutrophils have been shown rolling along inflamed endothelium during their recruitment into inflamed tissue and rapidly secreting S100A8/S100A9 via an E-selectin-dependent mechanism[3,4]. This leads to a rise in S100A8/S100A9 levels in the blood circulation, contributing to high serum levels of S100A8/S100A9, which correlates with disease progression in diverse acute and chronic inflammatory disorders[2]. Hence, for many years, S100A8/S100A9 has been used as a valuable clinical biomarker for therapeutic response monitoring. Of note, S100A8/S100A9 serum levels are not only a biomarker but also a disease indicator of pathogenesis of various diseases, including coronavirus disease 2019, chronic tuberculosis and cancer metastasis[5-7].

Similar to interleukin-1β (IL-1β), S100A8/S100A9 lacks a signal peptide required for secretion via the conventional secretory pathway[2]. In 2015, pore-forming gasdermin D (GSDMD) was described as a critical component of IL-1β release from the cytosol into the extracellular space[8]. GSDMD pore formation requires upstream inflammasome activation and subsequent inflammatory caspase and GSDMD cleavage[9]. In contrast to IL-1β, S100A8/S100A9 is prestored within the cytosol in large amounts, and its release is a rapid process, taking place within minutes after stimulation. Here, we show that E-selectin triggers rapid NLRP3 inflammasome activation and caspase 1 cleavage in neutrophils, triggering transient GSDMD pore formation by which S100A8/S100A9 exits cells in a time-restricted manner, which contributes to successful neutrophil recruitment during the acute inflammatory response.

## Results

### E-selectin-induced S100A8/S100A9 release is GSDMD and caspase 1 dependent

Secretion of S100A8/S100A9 by neutrophils is triggered by an E-selectin-dependent process[3,4]. Because pore-forming GSDMD has been proposed to mediate unconventional protein release, we investigated whether rapid S100A8/S100A9 release may be dependent on GSDMD and its activating protease caspase 1. Therefore, we injected tumor necrosis factor (TNF) into the scrotum of C57BL/6 wild-type (WT) mice and mice deficient for GSDMD ($Gsdmd^{-/-}$) or caspase 1 and caspase 11 ($Casp1^{-/-}Casp11^{-/-}$; Fig. 1a). S100A8/S100A9 serum levels were determined by enzyme-linked immunosorbent assay (ELISA) before and 2 h after TNF treatment. Application of TNF, which directly induces upregulation of E-selectin on endothelial cells[10], significantly increased serum levels of S100A8/S100A9 in WT mice. This increase was absent in $Gsdmd^{-/-}$ and $Casp1^{-/-}Casp11^{-/-}$ animals (Fig. 1b). Consistent with the role of E-selectin in S100A8/S100A9 release, E-selectin-deficient ($Sele^{-/-}$) mice did not exhibit increased serum levels of S100A8/S100A9 after TNF stimulation (Extended Data Fig. 1a). For all mouse strains, numbers of blood neutrophils and monocytes did not differ from those observed in WT controls (Extended Data Fig. 1b–g). Next, we stimulated bone marrow neutrophils isolated from WT, $Gsdmd^{-/-}$ and $Casp1^{-/-}Casp11^{-/-}$ mice with E-selectin or PBS (control) for 10 min and determined the amount of secreted S100A8/S100A9 in the supernatants (Fig. 1a). E-selectin stimulation induced rapid S100A8/S100A9 release in WT cells, whereas lack of caspase 1/caspase 11 or GSDMD completely prevented its release (Fig. 1c). Importantly, the overall amount of cytosolic S100A8/S100A9 was similar between WT, $Gsdmd^{-/-}$ and $Casp1^{-/-}Casp11^{-/-}$ neutrophils (Extended Data Fig. 1h). We have shown before that secreted S100A8/S100A9 binds to Toll-like receptor 4 (TLR4) in an autocrine manner, thereby activating β2-integrins on neutrophils[3]. This results in deceleration of rolling neutrophils along the inflamed vessel wall (slow rolling), facilitating firm leukocyte arrest. Therefore, we investigated to what degree loss of GSDMD and caspase 1/caspase 11 influences neutrophil

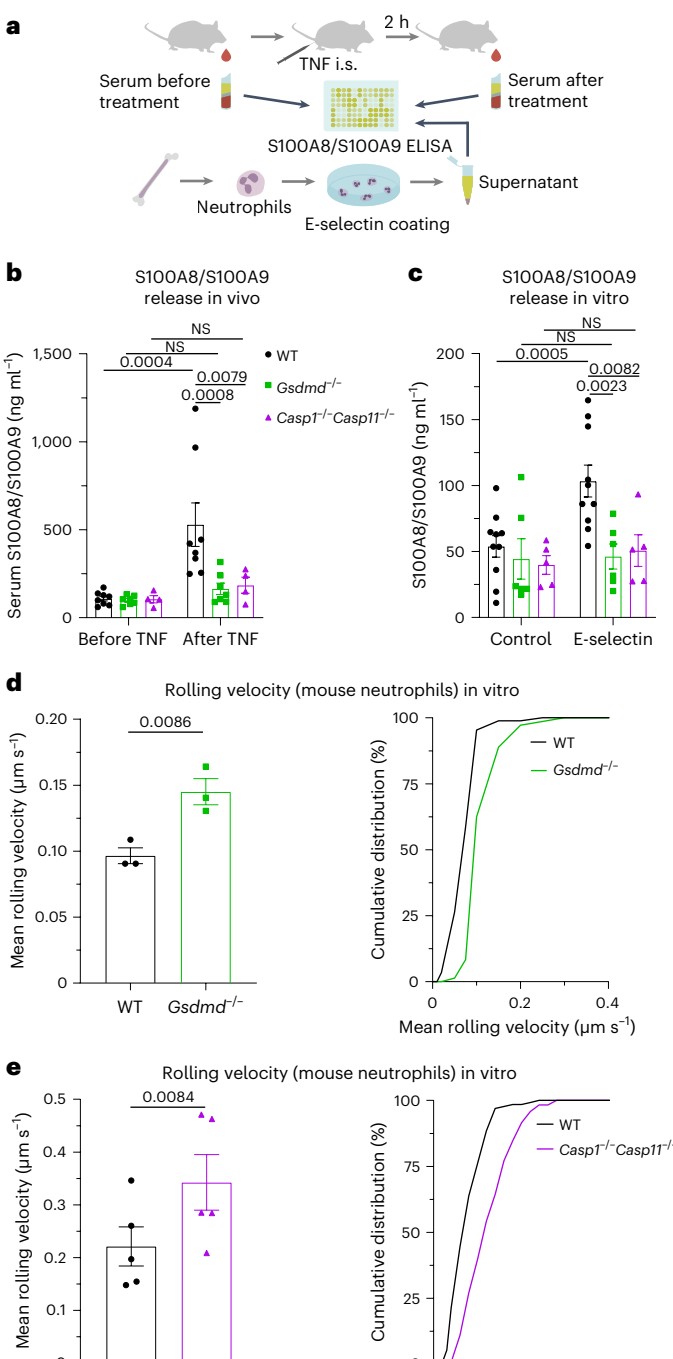

**Fig. 1 | E-selectin-induced S100A8/S100A9 release is GSDMD and caspase 1 dependent. a**, Schematic of the experimental design. **b**, Serum S100A8/S100A9 levels were analyzed by ELISA before and 2 h after intrascrotal (i.s.) TNF application to WT, $Gsdmd^{-/-}$ and $Casp1^{-/-}Casp11^{-/-}$ mice ($n = 8$ (WT), 7 ($Gsdmd^{-/-}$) and 4 ($Casp1^{-/-}Casp11^{-/-}$) mice per group); NS, not significant. **c**, Bone marrow neutrophils from WT, $Gsdmd^{-/-}$ and $Casp1^{-/-}Casp11^{-/-}$ mice were incubated with E-selectin or PBS for 10 min ($n = 10$ (WT), 6 ($Gsdmd^{-/-}$) and 5 ($Casp1^{-/-}Casp11^{-/-}$) mice per group). Supernatants were collected, and S100A8/S100A9 levels were analyzed. **d,e**, Rolling velocities of WT and $Gsdmd^{-/-}$ neutrophils (**d**; 87 (WT) and 72 ($Gsdmd^{-/-}$) cells from $n = 3$ mice per group) and WT and $Casp1^{-/-}Casp11^{-/-}$ neutrophils (**e**; 130 (WT) and 117 ($Casp1^{-/-}Casp11^{-/-}$) cells from $n = 5$ mice per group) were assessed in E-selectin/ICAM-1-coated flow chambers. Data are presented as mean ± s.e.m. and were analyzed by two-way repeated measures analysis of variance (RM ANOVA) with a Sidak's multiple comparison test (**b** and **c**) or are presented as mean ± s.e.m. and cumulative distribution and were analyzed by two-tailed paired Student's *t*-tests (**d** and **e**).

rolling velocities in vitro. *Gsdmd*[−/−] and *Casp1*[−/−]*Casp11*[−/−] neutrophils rolled significantly faster than WT neutrophils (Fig. 1d,e), underlining the inability of *Gsdmd*[−/−] and *Casp1*[−/−]*Casp11*[−/−] neutrophils to decelerate their rolling velocity through secretion of S100A8/S100A9. These results suggest that E-selectin-mediated rapid S100A8/S100A9 release is a process that relies on the expression of GSDMD and caspase 1/caspase 11 in neutrophils.

## E-selectin induces NLRP3 inflammasome activation in neutrophils

The defect in S100A8/S100A9 release observed in *Gsdmd*[−/−] and *Casp1*[−/−]*Casp11*[−/−] neutrophils suggests the involvement of an inflammasome-driven pathway in S100A8/S100A9 release. To investigate whether E-selectin stimulation mediates inflammasome activation and downstream caspase 1 cleavage in neutrophils, we stimulated primary human neutrophils with E-selectin or PBS (control) and performed Western blot analysis (Fig. 2a). To exclude that released S100A8/S100A9 induces TLR4-mediated inflammasome activation, we conducted these experiments in the presence of the TLR4 inhibitor TAC242 and paquinimod, an inhibitor of S100A8/S100A9–TLR4 interaction[3]. We detected cleaved caspase 1 (casp-1 p20/p22) in the supernatants of E-selectin-stimulated neutrophils within 10 min of incubation (Fig. 2b), independent of the presence of TAC242 and paquinimod, demonstrating that rapid caspase 1 cleavage is directly mediated by E-selectin and not by TLR4 signaling. In addition, the presence of the specific NLRP3 inflammasome inhibitor MCC950 prevented E-selectin-induced caspase 1 cleavage (Fig. 2c), suggesting an involvement of the NLRP3 inflammasome. Importantly, we also found increased amounts of cleaved caspase 1 in cell lysates of E-selectin-treated human neutrophils compared to that observed in control cells, demonstrating intracellular activity of caspase 1 downstream of E-selectin stimulation (Extended Data Fig. 2a).

To further validate caspase 1 activation after E-selectin stimulation, we used the fluorescent-labeled inhibitor of caspase 1 (FLICA) reagent (Fig. 2a). We found speck-like aggregates of active caspase 1 in human neutrophils within 10 min of E-selectin treatment (Fig. 2d,e). Furthermore, E-selectin induced an increase in the overall intensity of the FLICA signal compared to PBS (control; Fig. 2f), confirming rapid E-selectin-induced caspase 1 activation. Inhibition of caspase 1 activity using the inhibitor VX-765 prevented an increase in FLICA signal in E-selectin-stimulated cells compared to control-stimulated cells (Extended Data Fig. 2b,c), demonstrating the FLICA signal to be a proxy for intracellular caspase 1 activity.

To test whether E-selectin stimulation of neutrophils would also mediate rapid GSDMD cleavage, we stimulated human neutrophils for 10 min with E-selectin and determined the amount of full-length GSDMD and cleaved N-terminal GSDMD (GSDMD-NT) in the cellular lysates (Fig. 2g). Indeed, E-selectin stimulation induced rapid GSDMD-NT formation in neutrophils. Concomitant incubation of the cells with E-selectin and MCC950 prevented GSDMD cleavage. In addition, we visualized E-selectin-induced GSDMD cleavage and translocation to the cell surface by confocal microscopy using an antibody (clone EPR20829-408) that specifically recognizes GSDMD-NT but not non-active full-length GSDMD[11] (Fig. 2h,i). Both GSDMD cleavage and GSDMD-NT surface translocation were absent in unstimulated neutrophils and E-selectin-stimulated neutrophils pretreated with MCC950, suggesting that canonical NLRP3 activation contributes to GSDMD cleavage and GSDMD-NT surface mobilization following stimulation of neutrophils with E-selectin. To examine in more detail the subcellular localization of GSDMD in E-selectin-stimulated human neutrophils, we performed stimulated emission depletion (STED) microscopy and found colocalization of GSDMD and the plasma membrane marker wheat germ agglutinin (WGA) at the cell surface, suggesting GSDMD pore formation at the plasma membrane downstream of E-selectin stimulation (Extended Data Fig. 2d,e).

Finally, blockade of NLRP3 activation with MCC950 resulted in reduced S100A8/S100A9 release after E-selectin stimulation in vitro and in vivo, similar to the reduction observed in *Casp1*[−/−]*Casp11*[−/−] mice (Extended Data Fig. 2f,g), suggesting a predominant role of the NLRP3 inflammasome pathway in E-selectin-triggered S100A8/S100A9 release. Accordingly, the presence of MCC950 resulted in higher rolling velocities than control cells when perfused through E-selectin- and ICAM-1-coated microflow chambers (Extended Data Fig. 2h,i). Together, the data provide evidence that E-selectin mediates GSDMD-NT pore formation in neutrophils in an NLRP3-dependent manner, resulting in S100A8/S100A9 release.

## Rapid E-selectin-induced inflammasome activation is $K_v1.3$ dependent

$K^+$ efflux is a common trigger of NLRP3 inflammasome activation. Therefore, we wanted to investigate a potential role of $K^+$ efflux in E-selectin-induced rapid inflammasome activation. Indeed, high extracellular potassium (high $[K^+]_{ex}$), which prevents $K^+$ efflux, efficiently attenuated E-selectin-induced caspase 1 cleavage (Fig. 3a) and GSDMD-NT formation in human neutrophils (Extended Data Fig. 3a). Recently, we were able to show that the voltage-gated potassium channel $K_v1.3$ regulates neutrophil recruitment[12]. To study a potential role of $K_v1.3$ in E-selectin-mediated inflammasome activation, we stimulated human neutrophils with E-selectin in the absence and presence of the selective $K_v1.3$ inhibitor 5-(4-phenoxybutoxy) psoralen (PAP-1)[13]. We detected a significant reduction of cleaved caspase 1 in the supernatant of E-selectin-stimulated neutrophils when $K_v1.3$ was blocked (Fig. 3b). In line with human neutrophils, pharmacological inhibition of $K_v1.3$ with PAP-1 or genetic deletion of $K_v1.3$ (*Kcna3*[−/−] mice) in mouse neutrophils resulted in attenuated E-selectin-induced S100A8/S100A9 release in vitro (Fig. 3c,d), suggesting a critical role of $K_v1.3$ in rapid E-selectin-mediated NLRP3 inflammasome activation in neutrophils. Of note, the overall amount of cytosolic S100A8/S100A9 did not differ between WT and *Kcna3*[−/−] neutrophils (Extended Data Fig. 3b). To test whether $K^+$ efflux per se would mediate rapid inflammasome activation and caspase 1 cleavage resulting in S100A8/S100A9 release, we stimulated neutrophils with ATP and nigericin, two common inflammasome activators that mediate potassium efflux[14], and investigated the amount of S100A8/S100A9 in the supernatant. Interestingly, stimulation of neutrophils with ATP or nigericin alone was not sufficient to induce S100A8/S100A9 release within 10 min (Fig. 3e,f). Further, concomitant stimulation of neutrophils with lipopolysaccharide (LPS) and nigericin for 10 min did not result in remarkable S100A8/S100A9 release (Fig. 3f), pointing toward an exclusive role for E-selectin and $K_v1.3$ in mediating rapid NLRP3 inflammasome activation and GSDMD pore formation.

Next, we wanted to elucidate the specificity of E-selectin in inducing S100A8/S100A9 release through binding to its ligand on human neutrophils, L-selectin. Therefore, we stimulated human neutrophils with endoglycan and MAdCAM-1, two L-selectin ligands expressed on endothelial cells, which differ in their binding requirements to L-selectin compared to E-selectin[15]. Neither endoglycan nor MAdCAM-1 was able to induce S100A8/S100A9 release in vitro (Fig. 3g), underlining a unique role for E-selectin as a rapid inflammasome activator. In fact, we detected a time-dependent E-selectin-mediated S100A8/S100A9 release starting already after 1 min, demonstrating extremely rapid inflammasome activation and GSDMD pore formation in neutrophils (Extended Data Fig. 3c). Of note, rapid inflammasome activation did not result in IL-1β release in vitro (Fig. 3h), presumably due to low amounts of prestored IL-1β in unprimed neutrophils. However, as expected, classical priming of neutrophils with LPS for 2.5 h followed by 30 min of nigericin activation induced IL-1β release in neutrophils. Similar to nigericin, E-selectin stimulation induced IL-1β release in LPS-primed neutrophils (Fig. 3i), identifying E-selectin as an alternative and endogenous activation molecule for canonical inflammasome activation.

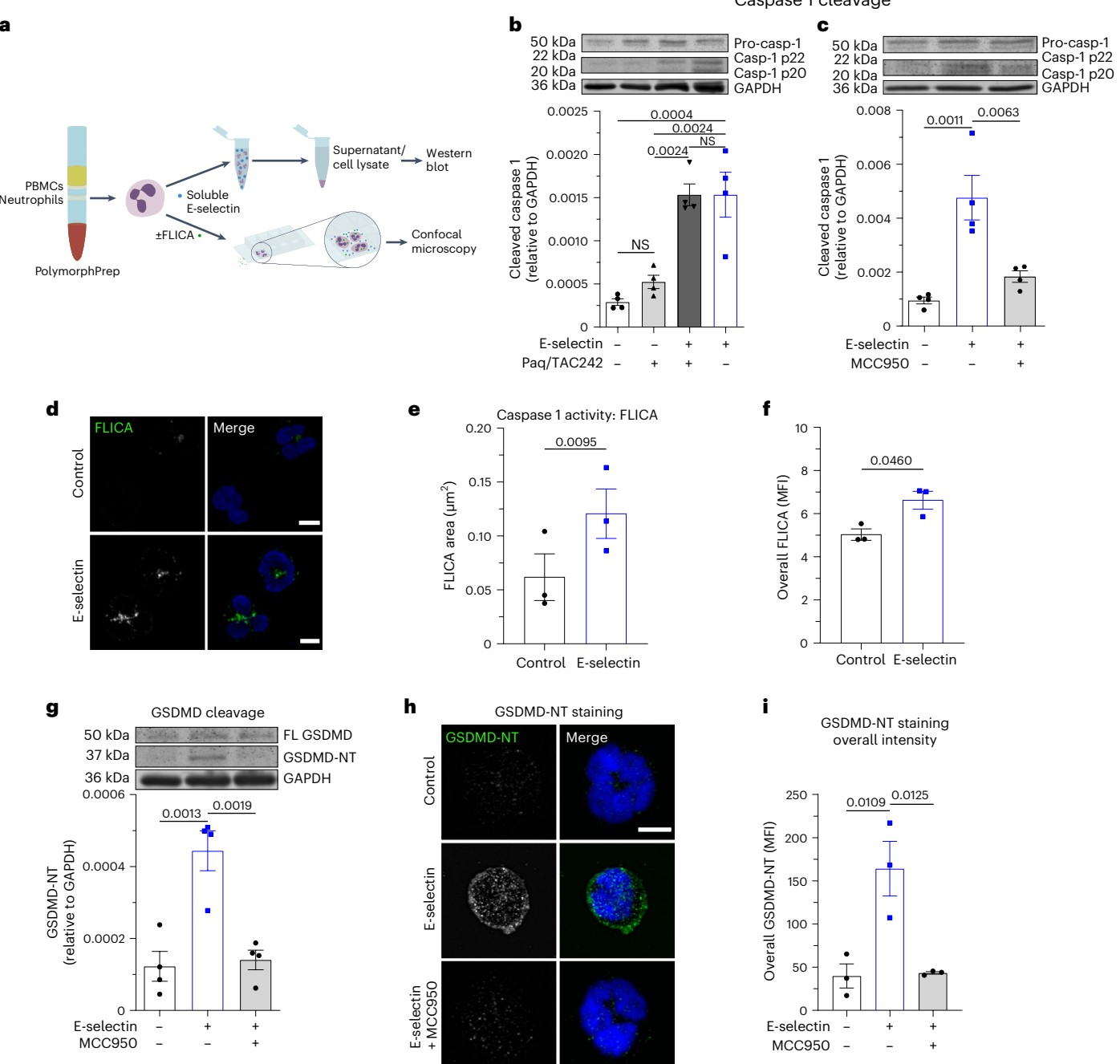

**Fig. 2 | E-selectin induces NLRP3 inflammasome activation in neutrophils.**
**a**, Schematic of the experimental design; PBMCs, peripheral blood mononuclear cells. **b**,**c**, E-selectin-induced cleavage of caspase 1 was assessed in isolated human neutrophils treated for 10 min with PBS/paquinimod/TAC242, PBS (control), E-selectin or a combination of E-selectin/paquinimod/TAC242 (**b**) and with PBS, E-selectin and E-selectin in the presence MCC950 (**c**). The amount of processed caspase 1 (casp-1 p20 and casp-1 p22) was determined in the supernatants, and the amount of procaspase 1 and GAPDH was determined in cell lysates ($n = 4$ independent experiments); Paq, paquinimod. **d**–**f**, Caspase 1 activity was determined in isolated human neutrophils stimulated with E-selectin or PBS (control) for 10 min loaded with FLICA dye and analyzed by confocal microscopy. Representative confocal images (**d**), quantification of mean FLICA+ area (**e**) and overall mean fluorescence intensity (MFI) of FLICA signal per cell

(**f**; 76 (control) and 112 (E-selectin) cells of $n = 3$ independent experiments) are shown. **g**, Isolated human neutrophils were treated for 10 min with PBS (control), E-selectin or E-selectin in the presence of MCC950, and the amount of full-length GSDMD (FL GSDMD), GSDMD-NT and GAPDH was determined in the cell lysates ($n = 4$ independent experiments). **h**,**i**, Representative confocal images (**h**) and overall MFI of GSDMD-NT staining of human neutrophils (**i**) stimulated with PBS (control), E-selectin or E-selectin in the presence of MCC950 for 10 min (49 (control), 75 (E-selectin) and 62 (E-selectin + MCC950) cells of $n = 3$ independent experiments). Data are presented as representative western blots and mean ± s.e.m. (analyzed by one-way ANOVA with Tukey's multiple comparison tests; **b**, **c** and **g**), as mean ± s.e.m. (analyzed by two-tailed paired Student's $t$-tests (**e** and **f**) or one-way ANOVA with a Tukey's multiple comparison test (**i**)) and as representative confocal images (**d** and **h**); scale bars, 5 μm.

To investigate whether E-selectin-induced GSDMD pores act as a conduit for cytosolic small alarmins ($M_w ≤ 50$ kDa) in general, we performed a secretome analysis of neutrophils stimulated with PBS or E-selectin for 10 min (ref. 16). Indeed, we were able to detect a significant increase in the amount of secreted cytosolic small alarmins after E-selectin stimulation (Extended Data Fig. 4). We detected

E-selectin-induced secretion of S100A8, S100A9 and macrophage migration-inhibitory factor in the supernatants and found a significant enrichment of the term 'cytosolic small alarmin' in an unbiased one-dimensional annotation enrichment (Extended Data Fig. 4c). S100A12 was identified in two of three samples and was therefore excluded from statistical analysis. In contrast to cytosolic small alarmins, alarmins located within the nucleus, like HMGB1, were not detected in the secretomes. Further, cytoplasmic alarmins with higher molecular weights, like HSP70 or HSP90, were either not detected in the secretome or secretion was not upregulated after E-selectin stimulation compared to control. In line with the findings shown in Fig. 3h, we did not detect proteins of the IL-1 family (IL-1α, IL-1β and IL-18) in the secretomes of control or E-selectin-stimulated neutrophils. Of note, we also did not observe the induction of granule release after 10 min of E-selectin stimulation as validated by secretome analysis (Extended Data Fig. 5a–c) and flow cytometry (Extended Data Fig. 5d–g), further underlining the specificity of E-selectin in inducing GSDMD pore formation in neutrophils within the vasculature, leading to the specific release of cytosolic small alarmins without an overall activation of neutrophils.

## E-selectin induces NLRP3 tyrosine phosphorylation and ASC oligomerization

Next, we wanted to elucidate in more detail downstream signaling events linking E-selectin stimulation to K$^+$ efflux via K$_V$1.3 and NLRP3 inflammasome assembly. Mueller et al. showed a central role of the Tec family kinase Bruton's tyrosine kinase (BTK) in downstream signaling events induced by E-selectin[17]. Interestingly, BTK has been shown recently to positively regulate inflammasome activation by mediating tyrosine phosphorylation of NLRP3 at the PYD-NACHT linker domain[18,19]. Indeed, we detected tyrosine phosphorylation of NLRP3 in neutrophils after E-selectin stimulation, which was abolished in the presence of ibrutinib, a Food and Drug Administration-approved BTK inhibitor (Fig. 4a). Moreover, BTK inhibition efficiently abolished S100A8/ S100A9 release in vitro (Fig. 4b). Interestingly, E-selectin-induced NLRP3 tyrosine phosphorylation was independent of K$_V$1.3 activity, as we detected NLRP3 tyrosine phosphorylation after E-selectin treatment in *Kcna3*$^{-/-}$ neutrophils (Fig. 4c). This implies that NLRP3 phosphorylation and K$^+$ efflux converge to induce oligomerization of the adaptor protein apoptosis-associated speck-like protein containing a CARD (ASC) and formation of the active inflammasome complex, promoting caspase 1 cleavage downstream of E-selectin treatment[14]. In fact, we detected an increase in ASC oligomers in neutrophils after stimulating the cells with E-selectin compared to treatment with PBS control (Fig. 4d,e). This was dependent on K$^+$ efflux via K$_V$1.3, as *Kcna3*$^{-/-}$ neutrophils stimulated with E-selectin were unable to form ASC oligomers (Fig. 4f,g), indicating that E-selectin stimulation mediates NLRP3 tyrosine phosphorylation and ASC oligomerization; however, only the latter was dependent on K$^+$ efflux via K$_V$1.3.

## E-selectin-induced GSDMD pore formation is transient

To evaluate E-selectin-induced GSDMD pore formation, we used isolated human and mouse neutrophils and live-cell confocal microscopy to visualize the passive uptake of propidium iodide (PI) through GSDMD pores (Fig. 5a)[20]. Stimulation of human neutrophils with E-selectin induced PI uptake, leading to nuclear staining within minutes (Fig. 5b,c). After 10 min of E-selectin stimulation, almost 100% of neutrophils displayed positive PI staining (Fig. 5d). Pretreatment with MCC950 in turn completely prevented PI uptake (Fig. 5b–d). Accordingly, mouse WT neutrophils efficiently took up PI, and over 90% of WT neutrophils displayed positive staining for PI after 10 min of incubation with E-selectin (Fig. 5e–g). Lack of caspase 1 and caspase 11 completely prevented PI uptake (Fig. 5e–g and Supplementary Video 1), demonstrating the dependence of E-selectin-induced rapid pore formation on inflammatory caspases.

Next, we evaluated changes in membrane potential associated with E-selectin-induced pore formation using current clamp experiments (Extended Data Fig. 6a). To do so, we assessed the baseline membrane potential in human neutrophils before application of E-selectin (inset in Extended Data Fig. 6a). Neutrophils displayed a baseline membrane potential ranging from −11 mV to −49 mV with mean values of around −30 mV (Extended Data Fig. 6b). Neutrophils pretreated with PAP-1 or disulfiram, a GSDMD pore formation inhibitor[21], exhibited similar baseline levels. Application of E-selectin resulted in depolarization of membrane potential in neutrophils, which could be completely abolished by preincubation of the cells with PAP-1 or disulfiram, confirming that E-selectin-induced depolarization is dependent on K$_V$1.3 activation and subsequent GSDMD pore formation, allowing moderate, unspecific ion fluxes across the plasma membrane (Extended Data Fig. 6c).

Several reports implicated GSDMD pores to drive neutrophil pyroptosis[22] and to play a role in the generation of neutrophil extracellular traps[23,24]. However, considering the fact that neutrophils are not supposed to lose their cellular integrity during the intravascular phase of their recruitment into inflamed tissue, we hypothesized that GSDMD pores are formed only transiently. Non-lytic functions of GSDMD have emerged recently, including lysis-independent release of mature IL-1β and cytosolic small DAMPs[20,25,26]. To test for GSDMD pore formation without inducing neutrophil cell death, we assessed lactate dehydrogenase (LDH) concentrations as a proxy for lytic cell death in supernatants of neutrophils stimulated with E-selectin (Fig. 5a). LDH is too large to exit through GSDMD pores and can only leave the cell after membrane rupture. E-selectin stimulation of human neutrophils neither induced cell death after 10 min nor after 30 min or 180 min (Fig. 5h–j). In contrast to E-selectin, LPS priming for 2.5 h followed by 30 min of nigericin stimulation induced moderate LDH release in human neutrophils and high LDH release in human monocytes (positive control), respectively (Extended Data Fig. 6d–g). These findings show that E-selectin stimulation, although inducing GSDMD pore formation and S100A8/S100A9 release, does not mediate any form of cell death in neutrophils. To investigate whether E-selectin-induced GSDMD pore formation might be a transient process, we activated isolated human neutrophils with E-selectin and added PI after 5, 10, 15 or 20 min of stimulation (Fig. 5k). Interestingly, continuous stimulation of neutrophils with E-selectin reduced PI uptake in a time-dependent manner (Fig. 5l,m). Prestimulation of cells with E-selectin for 10 min before the addition of PI reduced the amount of PI$^+$ cells to around 70%. After 15 min or 20 min of E-selectin stimulation, respectively, PI uptake was completely abolished. These results suggest that E-selectin-induced GSDMD pore formation in neutrophils is a time-limited and reversible process.

Recently, the endosomal sorting complexes required for transport (ESCRT) machinery was shown to trigger repair programs to remove GSDMD pores from the plasma membrane, thereby counterbalancing cell death[27]. To elucidate a potential role of the ESCRT machinery in this self-repair program, we investigated subcellular localization of CHMP4B, an ESCRT III-associated protein, after E-selectin stimulation for 15 min. ESCRT proteins form a punctate pattern during membrane repair and translocate to the plasma membrane[27,28]. Consistently, stimulation of human neutrophils with E-selectin resulted in the formation of CHMP4B puncta, presumably corresponding to functional assemblies of ESCRT III, whereas unstimulated neutrophils (control cells) displayed a weak and rather diffuse cytoplasmic staining of CHMP4B (Fig. 5n,o). These findings suggest the induction of membrane repair processes in E-selectin-stimulated neutrophils to clear the surface membrane of GSDMD pores.

## GSDMD-dependent pore formation supports neutrophil recruitment

Finally, we wanted to test if GSDMD pore formation supports neutrophil recruitment during acute inflammation in vivo. Therefore, we studied leukocyte recruitment in an acute, predominantly neutrophil-driven,

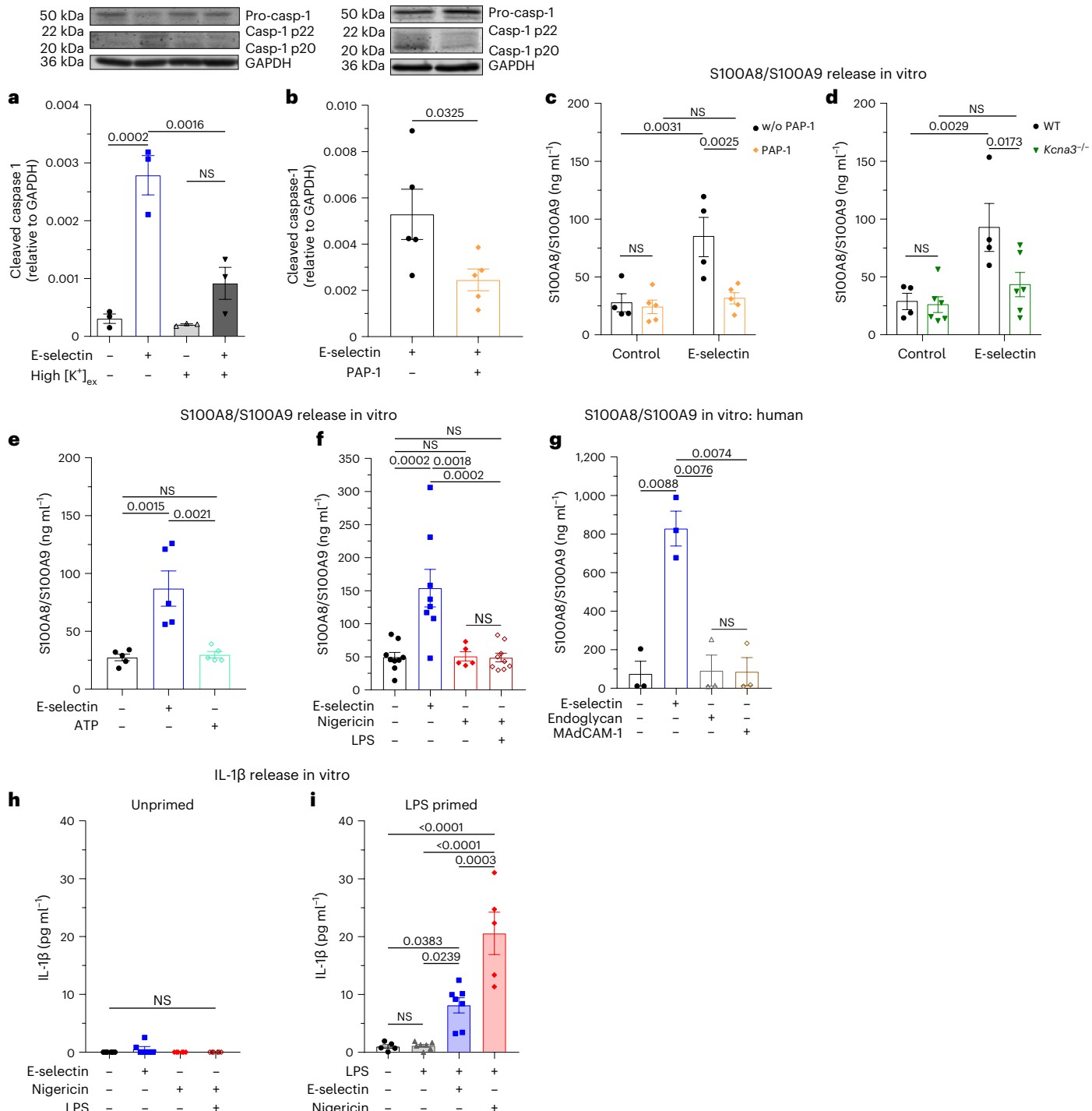

**Fig. 3 | Rapid E-selectin-induced inflammasome activation is $K_V1.3$ dependent. a,b**, Caspase 1 cleavage was assessed in isolated human neutrophils resuspended in normal HBSS and HBSS with high $[K^+]_{ex}$ (**a**) and pretreated with $K_V1.3$ inhibitor PAP-1 (**b**; 50 nM) or vehicle control and stimulated with E-selectin for 10 min. The amount of processed caspase 1 (casp-1 p20 and casp-1 p22) was determined in the supernatants, and the amounts of procaspase 1 and GAPDH were determined in cell lysates ($n = 3$ and 5 independent experiments, respectively). **c,d**, Bone marrow neutrophils from WT mice pretreated with PAP-1 (50 nM) or vehicle control (**c**) or from WT and $Kcna3^{-/-}$ mice (**d**) were incubated with E-selectin or PBS (control) for 10 min ($n = 4$ (control), 5 (PAP-1), 4 (WT) and 6 ($Kcna3^{-/-}$) mice per group); w/o, without. **e,f**, Bone marrow neutrophils from WT mice were incubated with PBS, E-selectin or ATP (**e**) or with PBS, E-selectin, nigericin or a combination of LPS and nigericin (**f**) for 10 min ($n = 5$ (control (**e**), E-selectin (**e**), ATP (**e**) and nigericin (**f**)), 9 (control (**f**), LPS and nigericin

(**f**)) and 8 (E-selectin (**f**)) mice per group). **g**, Isolated human neutrophils were stimulated with PBS, E-selectin, MAdCAM-1 or endoglycan for 10 min ($n = 3$ independent experiments). In **c–g**, supernatants were collected, and S100A8/ S100A9 levels were analyzed by ELISA. **h,i**, IL-1β levels were analyzed by ELISA in the supernatants from WT bone marrow neutrophils stimulated with PBS, E-selectin, nigericin or a combination of LPS and nigericin for 10 min (**h**; $n = 6$ mice per group) or primed with PBS or LPS for 2.5 h and subsequently stimulated for 30 min with PBS, E-selectin or nigericin (**i**; $n = 5$ (control, LPS/nigericin) and 7 (LPS, LPS/E-selectin) mice per group). Data are presented as representative western blots and mean ± s.e.m. (**a** and **b**; data were analyzed by one-way ANOVA with a Tukey's multiple comparison test or two-tailed paired Student's $t$-tests, respectively) or mean ± s.e.m. (**c–i**; data were analyzed by two-way RM ANOVA with a Sidak's multiple comparison test (**c** and **d**) or one-way ANOVA with Tukey's multiple comparison tests (**e–i**)).

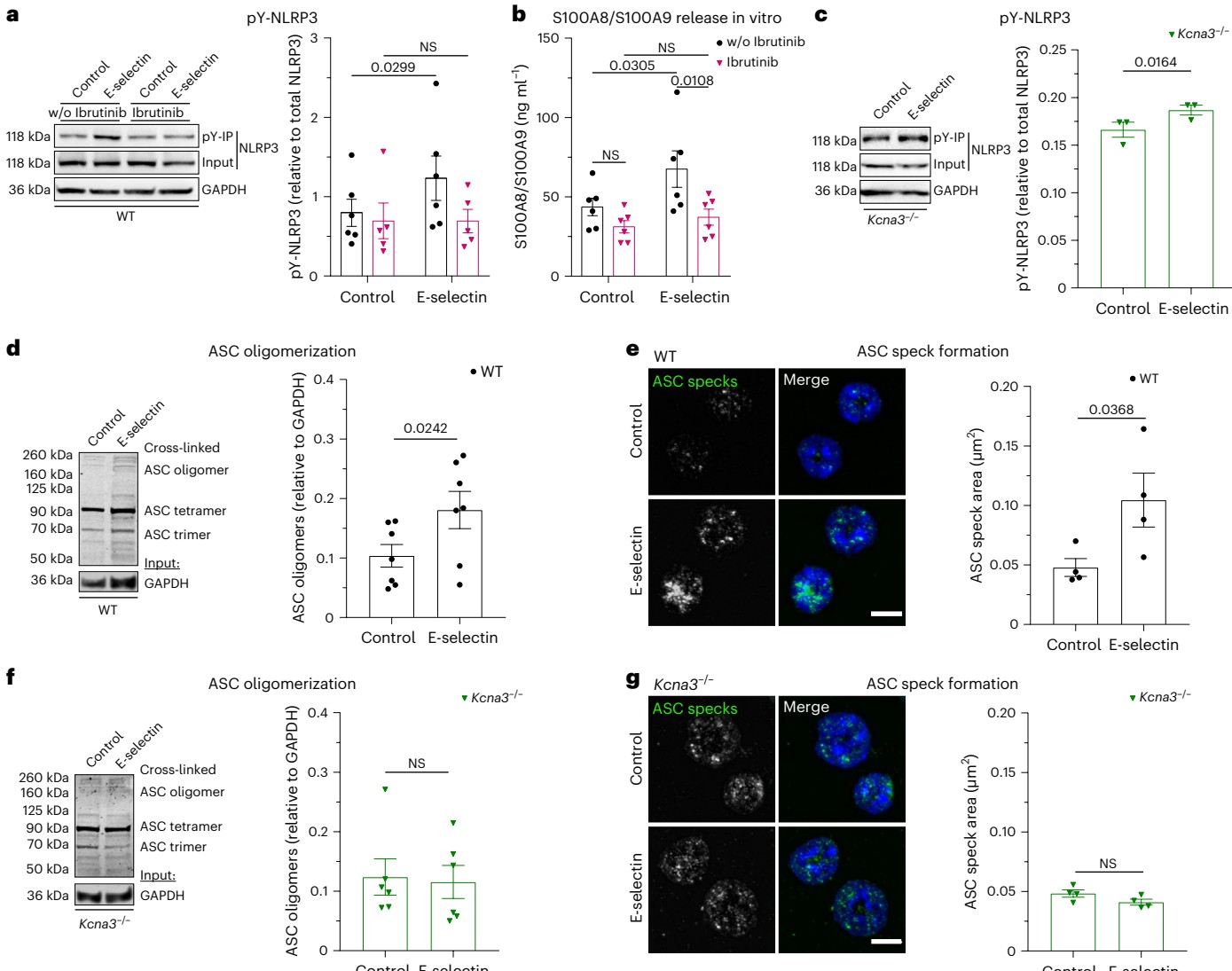

**Fig. 4 | E-selectin induces NLRP3 tyrosine phosphorylation and ASC oligomerization. a**, Tyrosine phosphorylation of NLPR3 after 5 min of PBS or E-selectin stimulation was assessed via phospho-tyrosine immunoprecipitation (pY-IP) in WT bone marrow neutrophils pretreated with the BTK inhibitor ibrutinib (0.6 μM) or vehicle control (*n* = 5 (ibrutinib) and 6 (without ibrutinib) independent experiments). **b**, Bone marrow neutrophils from WT mice were incubated with ibrutinib (0.6 μM) or vehicle control and subsequently stimulated with E-selectin or PBS (control) for 10 min (*n* = 6 mice per group). Supernatants were collected, and S100A8/S100A9 levels were analyzed by ELISA. **c**, Bone marrow *Kcna3*⁻/⁻ neutrophils were stimulated for 5 min with PBS or E-selectin, and tyrosine phosphorylation of NLRP3 was analyzed by pY-IP (*n* = 3 independent experiments). **d**–**g**, Bone marrow neutrophils from WT (**d** and **e**) and *Kcna3*⁻/⁻

(**f** and **g**) mice were stimulated with E-selectin or PBS (control) for 10 min, and ASC oligomerization was analyzed by western blotting (*n* = 7 (WT) and 6 (*Kcna3*⁻/⁻) mice per group) and confocal microscopy (WT: 38 (control) and 32 (E-selectin) cells; *Kcna3*⁻/⁻: 41 (control) and 34 (E-selectin) cells of *n* = 4 mice per group). Data are presented as representative western blots and mean ± s.e.m. (**a**, **c**, **d** and **f**; data were analyzed by two-way RM ANOVA and a Sidak's multiple comparison test (**a**) or two-tailed paired Student's *t*-tests (**c**, **d** and **f**)), mean ± s.e.m. (**b**; data were analyzed by two-way RM ANOVA and a Sidak's multiple comparison test) and representative micrographs and mean ± s.e.m. (**e** and **g**; 51 (WT control), 64 (WT E-selectin), 46 (*Kcna3*⁻/⁻ control) and 48 (*Kcna3*⁻/⁻ E-selectin) cells of *n* = 4 mice per group; data were analyzed by paired Student's *t*-tests); scale bars, 5 μm.

inflammatory setting[29] and assessed leukocyte rolling, leukocyte rolling velocities and number of adherent leukocytes in postcapillary venules of the TNF-stimulated mouse cremaster muscle using intravital microscopy. In addition, we analyzed the number of extravasated neutrophils in the TNF-stimulated cremaster muscle (Fig. 6a). Rolling flux fraction was not affected in the absence of GSDMD (Fig. 6b), demonstrating that GSDMD pores and release of cytosolic proteins are not regulating the number of rolling neutrophils. However, and in line with the flow chamber experiments (Fig. 1d), *Gsdmd*⁻/⁻ cells rolled significantly faster in vivo (Fig. 6c and Supplementary Video 2), indicating that pore formation and release of S100A8/S100A9 is important for the deceleration of leukocyte rolling velocities in vivo. Furthermore, in the absence of

GSDMD, leukocytes were unable to adhere efficiently (Fig. 6d and Supplementary Video 2), and the number of perivascular neutrophils was decreased (Fig. 6e). Notably, overall surface expression of adhesion and extravasation-relevant molecules, such as Mac-1 (CD11b/α$_M$), LFA-1 (CD11a, α$_L$), CD18 (β$_2$) and CXCR2 (CD182), on neutrophils was similar between WT and *Gsdmd*⁻/⁻ mice under baseline conditions (Extended Data Fig. 7). Consistently, MCC950 injection into WT mice 1 h before TNF application increased leukocyte rolling velocity and significantly reduced the number of adherent and extravasated cells in the inflamed cremaster muscle (Extended Data Fig. 8). Importantly, microvascular parameters did not differ between respective groups (Extended Data Table 1). Of note, genetic deletion of *Gsdmd* or pharmacological

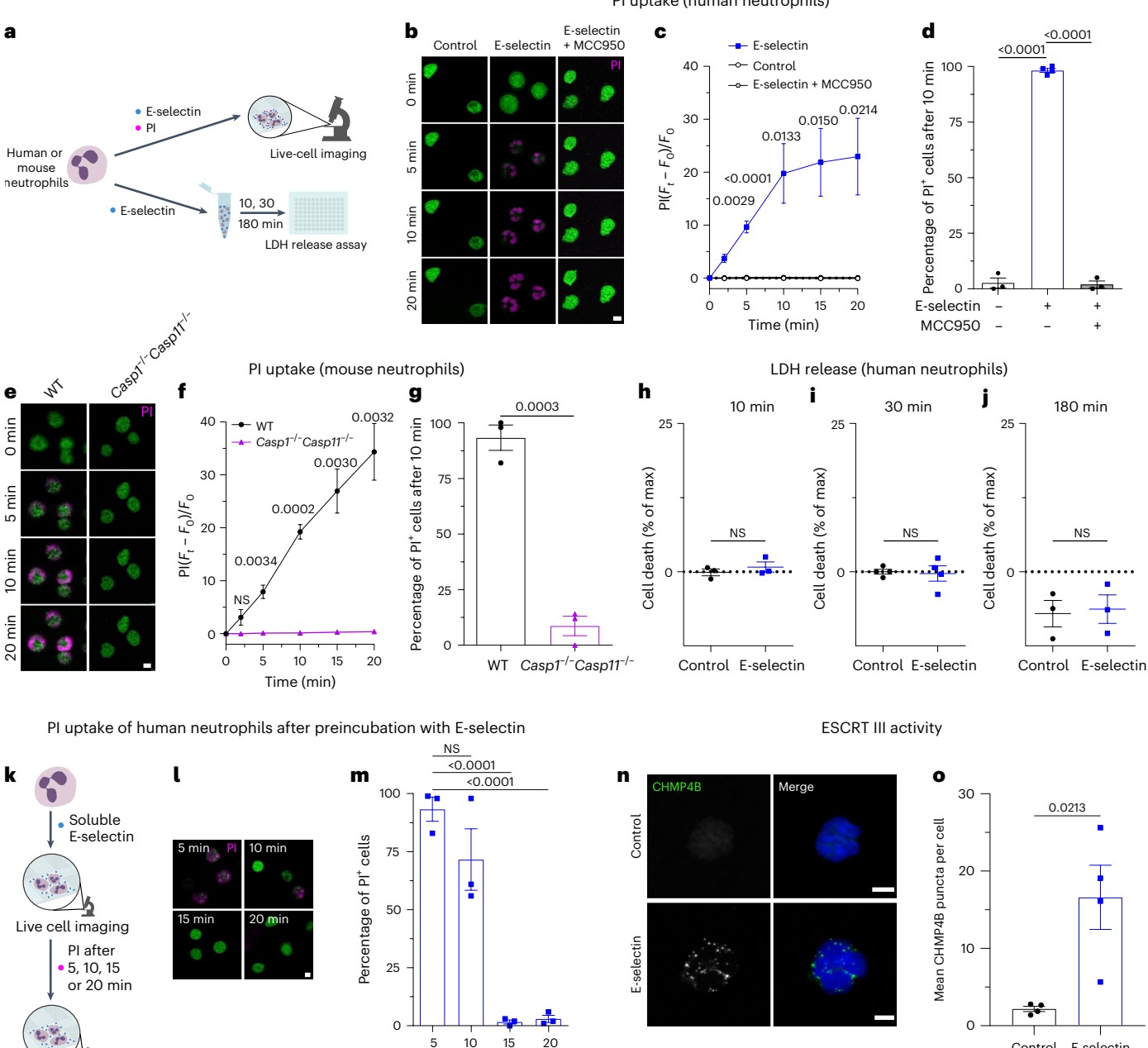

**Fig. 5 | E-selectin-induced GSDMD pore formation is transient. a**, Schematic of the experimental design. **b**, Representative confocal images of PI uptake in isolated human neutrophils stimulated with PBS, E-selectin or E-selectin in the presence of MCC950. **c,d**, Time course analysis (**c**) and percentage of PI⁺ cells after 10 min of stimulation (**d**; 1,029 (PBS), 429 (E-selectin) and 315 (E-selectin/MCC950) cells of $n$ = 3 (control and E-selectin + MCC950) and 4 (E-selectin) independent experiments); $F_t$, fluorescence at the respective time point; $F_0$, fluorescence at $t_0$. **e**, Representative confocal images of PI uptake in bone marrow neutrophils from WT and $Casp1^{-/-}Casp11^{-/-}$ mice treated with E-selectin. **f,g**, Time course (**f**) and percentage of PI⁺ WT and $Casp1^{-/-}Casp11^{-/-}$ neutrophils after 10 min of stimulation (**g**; 190 (WT) and 351 ($Casp1^{-/-}Casp11^{-/-}$) cells of $n$ = 3 mice per group). **h–j**, Cell death of human neutrophils was assessed via LDH release after stimulation with PBS or E-selectin for 10 min (**h**), 30 min (**i**) and

180 min (**j**; $n$ = 3 (10 min and 180 min) and 4 (30 min) independent experiments). **k–m**, PI uptake was measured in isolated human neutrophils after pretreatment of cells with E-selectin for the specified amounts of time (**k**). Representative confocal images of PI uptake (**l**) and percentage of PI⁺ neutrophils (**m**) are shown (274 (5 min), 313 (10 min), 358 (15 min) and 466 (20 min) cells of $n$ = 3 independent experiments). **n,o**, CHMP4B expression was investigated in isolated human neutrophils stimulated with PBS or E-selectin for 15 min. Representative confocal images of CHMP4B localization (**n**) and number of CHMP4B puncta per cell (**o**; 24 (control) and 47 (E-selectin) cells of $n$ = 4 independent experiments) are shown. Data are presented as mean ± s.e.m. (**c**, **d**, **f–j**, **m** and **o**; data were analyzed by one-way ANOVA with a Dunnetts's multiple comparison test (**c**, **d** and **m**), two-tailed unpaired Student's $t$-tests (**f–j**) or two-tailed paired Student's $t$-tests (**m** and **o**)) and representative micrographs (**b**, **e**, **l** and **n**); scale bars, 5 μm.

inhibition of NLRP3 inflammasome activation will also affect inflammasome activation in cell types other than neutrophils, which might additionally contribute to the reduction in the inflammatory response observed in the in vivo setting.

## Discussion

Neutrophils comprise the major part of circulating white blood cells in humans and are critically involved in the early inflammatory response exerted by the innate immune system. The importance of

inflammasome activation in neutrophils during the inflammatory response has been rather neglected in the past, and inflammasome research has focused primarily on monocytes and macrophages. However, neutrophils express and store key components of the inflammasome machinery and are a major source of IL-1β secretion; so the idea of targeting inflammasome activation in neutrophils as a potential anti-inflammatory strategy during overwhelming immune responses or autoimmune disorders became more and more appreciated over the last decade[30–34]. With our present study, we expand the knowledge on inflammasome activation in neutrophils and identify an E-selectin-induced mechanism leading to rapid, but transient, GSDMD pore formation accompanied by S100A8/S100A9 release during inflammation. In contrast to other cell types that need to be 'converted' into a proinflammatory phenotype or primed before being able to secret high amounts of cytokines and other proinflammatory mediators, neutrophils are fast-reacting cells and are fully equipped with the receptor and protein machinery needed to effectively fight against invading pathogens. S100A8/S100A9 is one of the major components within the cytosol of circulating blood neutrophils and an important mediator of inflammation[2]. In its cytosolic form, S100A8/S100A9 is abundantly expressed and completely processed and does not require upregulation by de novo protein synthesis. Instead, IL-1β secretion is a rather slow process that relies on de novo synthesis during the priming phase[9]. Before IL-1β secretion, the inflammasome is primed through engagement of pattern recognition receptors or cytokines to upregulate inflammasome components like NLRP3 and caspase 1. Priming is followed by an activation step, which can be induced by bacterial, viral and fungal products, sterile inflammation or cellular stress. Upstream signals of NLRP3 activation are thought to include efflux of $K^+$ or chloride, calcium influx, lysosomal disruption, metabolic changes, *trans*-Golgi disassembly and mitochondrial dysfunction[14]. Here, we describe a rapid activation process for GSDMD pore formation in neutrophils triggered by E-selectin, which is expressed on inflamed endothelial cells. Importantly, this fast activation process is independent of TLR4 engagement, as disruption of TLR4 downstream signaling with a TLR4 inhibitor and paquinimod does not affect the amount of cleaved caspase 1 within the supernatant of E-selectin-stimulated neutrophils.

Similar to conventional inflammasome activation, E-selectin-induced rapid inflammasome activation and subsequent S100A8/S100A9 release in neutrophils is dependent on $K^+$ efflux, which is provided via the voltage-sensitive $K^+$ channel $K_V1.3$. Interestingly, stimulation of neutrophils over 10 min with nigericin or ATP alone (two known inducers of $K^+$ efflux) or in combination with LPS[35,36] was unable to trigger rapid S100A8/S100A9 release from neutrophils. These findings suggest that E-selectin stimulation not only mediates $K^+$ efflux by $K_V1.3$ but also induces signaling events leading to post-translational NLRP3 modifications, a process required for proper NLRP3 activation[14,37,38]. Indeed, we were able to demonstrate that E-selectin triggers BTK-dependent NLRP3 tyrosine phosphorylation. Accordingly, BTK was reported to positively regulate NLRP3 inflammasome activation by phosphorylating four conserved tyrosine residues on NLRP3. This promoted formation of NLRP3 oligomers and complexes with ASC, which resulted in IL-1β secretion in LPS/nigericin- or LPS/monosodium urate crystal-treated primary immune cells[18]. We show here that E-selectin induces post-translational modifications of NLRP3 independent of $K^+$ efflux, as E-selectin-mediated NLRP3 tyrosine phosphorylation was still present in the absence of $K_V1.3$. However, and in line with LPS/ATP-induced inflammasome activation[39], ASC speck formation occurred downstream of $K^+$ efflux in rapid E-selectin-induced inflammasome activation, as ASC oligomerization and ASC speck formation were absent in E-selectin-stimulated KCNA3-deficient neutrophils.

During NLRP3 inflammasome activation, the 52-kDa GSDMD is processed to the 31-kDa GSDMD-NT fragment, which oligomerizes at

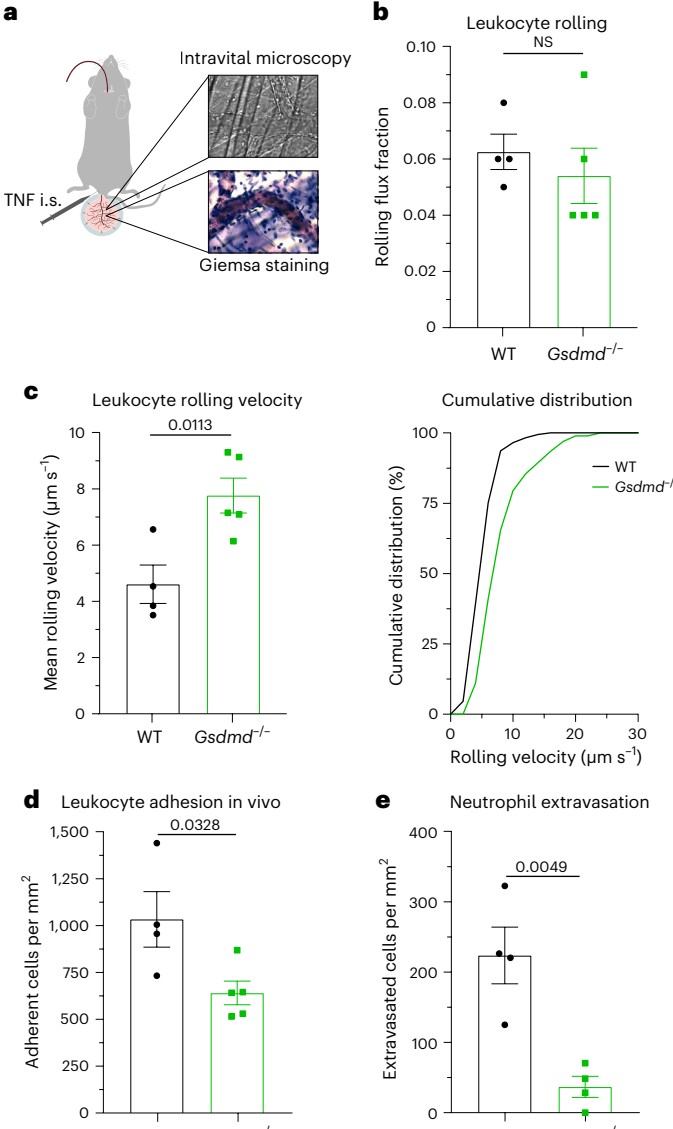

**Fig. 6 | GSDMD-dependent pore formation supports neutrophil recruitment. a**, Schematic of the experimental design. Male WT and *Gsdmd*[−/−] mice were stimulated after intrascrotal administration of TNF (500 ng) 2 h before intravital microscopy of postcapillary venules of the mouse cremaster muscle. **b–d**, Neutrophil rolling (**b**), neutrophil rolling velocity (**c**) and number of adherent neutrophils per vessel surface (**d**) were analyzed (18 and 24 vessels, respectively, of $n = 4$ (WT) and 5 (*Gsdmd*[−/−]) mice per group). **e**, TNF-stimulated cremaster muscles were stained with Giemsa, and the number of perivascular neutrophils was quantified (52 and 31 vessels, respectively, of $n = 4$ mice per group). Data are presented as mean ± s.e.m. (**b–e**) and as cumulative distribution (**c**; right) and were analyzed by two-tailed unpaired Student's *t*-tests.

the plasma membrane and forms pores, thereby increasing permeability for macromolecules and the release of IL-1β in macrophages, dendritic cells and neutrophils[11,25,40]. The pores have an inner diameter of around 20 nm (refs. 41,42) and are predominantly negatively charged, thus predominantly allowing passage of neutral or positively charged proteins[40,42]. With a molecular weight of 24 kDa, S100A8/S100A9 is suitable to exit through GSDMD pores. However, the net charge of released S100A8 and S100A9 is currently unclear but might be influenced by the local influx of $Ca^{2+}$ ions through the open GSDMD pores, favoring $Ca^{2+}$ binding to S100A8/S100A9 and enhancing hydrophobicity and positive charge, as previously reported[43].

Our findings provide evidence that E-selectin-driven S100A8/S100A9 release requires canonical NLRP3 inflammasome activation and GSDMD pores, as inhibition of the NLRP3 inflammasome by MCC950 and genetic deletion of *Gsdmd* completely abolished its release from neutrophils in vitro and in vivo. Initially, cell death has been proposed to be a direct and unavoidable consequence of GSDMD pore formation[44,45]. However, non-lytic functions of GSDMD emerged, such as promoting lysis-independent release of mature IL-1β and cytosolic small DAMPs[20,25,26,46]. Accordingly, the ESCRT machinery was shown to trigger repair programs to remove GSDMD pores from the plasma membrane, thereby counterbalancing cell death[27]. Neutrophils rolling on inflamed endothelium are not supposed to undergo cell death within the vasculature, as neutrophils need to extravasate into inflamed tissue. Within the tissue, neutrophils fight against invading pathogens and clear the area from cell debris or injured tissue during sterile inflammation together with other innate immune cells, like tissue-resident macrophages or dendritic cells. We show that, although almost all neutrophils stimulated with E-selectin displayed positive staining for PI, the amount of LDH release was rather low, suggesting protective mechanisms that allow neutrophils to close/remove GSDMD pores from their surface. In general, neutrophils seem to be less prone to undergo cell death, in particular after canonical inflammasome activation[23,26]. After NLRC4 inflammasome activation, for example, neutrophils are resistant to pyroptotic cell death[26], and GSDMD-independent secretion pathways were described for IL-1β from non-pyroptotic cells, such as neutrophils[40]. Karmakar et al. recently described an additional IL-1β secretory mechanism in neutrophils via an autophagy-dependent, non-lytic pathway. They showed that p31 GSDMD pores do not accumulate in the plasma membrane of neutrophils but associate with azurophilic granules, leading to neutrophil elastase release into the cytosol, mediating a secondary cascade of serine protease dependent-GSDMD processing[11]. In 2018, Rühl et al. showed that calcium influx through GSDMD pores initiates a membrane repair program by recruiting the ESCRT machinery to damaged membrane areas[27]. We detected CHMP4B puncta after stimulating the cells with E-selectin, implying the formation of the ESCRT III membrane repair machinery following rapid E-selectin-induced GSDMD pore formation. These findings support our notion that GSDMD pores appear in a transient fashion, enabling the release of S100A8/S100A9 from recruited neutrophils within the vasculature while preserving their integrity, a prerequisite for subsequent diapedesis.

Taken together, we uncovered a rapid, endogenous and transient E-selectin-dependent activation system for neutrophils within inflamed microvessels, which triggers NLRP3 inflammasome assembly, leading to transient GSDMD pore formation and S100A8/S100A9 release from neutrophils without inducing cell death. Although triggering inflammation within the vasculature, neutrophils remain capable of extravasation and functional to fulfill their role in subsequent immune responses within tissues. Modulating this E-selectin/NLRP3 inflammasome/GSDMD-dependent activation pathway in neutrophils might therefore offer an interesting therapeutic approach for the treatment of acute and chronic inflammatory disorders with unwanted neutrophil accumulation.

## Online content

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

## Methods

### Mice

C57BL/6NCrl (WT) mice were purchased from Charles River Laboratories. $Gsdmd^{-/-}$ (ref. [47]) mice were from P. Broz (University of Lausanne, Switzerland), $Casp1^{-/-}Casp11^{-/-}$ (ref. [48]) mice were from V. Hornung (LMU München, Germany), $Sele^{-/-}$ (ref. [49]) mice were from D. Vestweber (Max Planck Institute for Molecular Biomedicine, Münster, Germany) and $Mrp14^{-/-}$ mice were from Johannes Roth (University of Münster, Germany). $Kcna3^{-/-}$ mice[50] were purchased from Jackson Laboratories and maintained on a C57BL/6NCrl background. WT, $Mrp14^{-/-}$ and $Kcna3^{-/-}$ mice were housed at 20 to 22 °C with a humidity of 45–55% and a 12-h light/12-h dark cycle. Eight- to 25-week-old male and female were used for all experiments. Mice of similar age and sex were used for all experiments except for intravital microscopy studies in the cremaster muscle, where only male mice were used. The allocation of animals to groups was performed randomly or on the basis of their genotype. Animal experiments were approved by the Government of Oberbayern (AZ: ROB-55.2-2532.Vet_02-18-22 and ROB-55.2-2532. Vet_02-17-102) and were performed in accordance with the guidelines from Directive 2010/63/EU. For in vivo experiments, mice were anesthetized via intraperitoneal injection using a combination of ketamine/xylazine (125 mg per kg (body weight) and 12.5 mg per kg (body weight), respectively, in a volume of 0.1 ml per 8 g body weight). All mice were killed by cervical dislocation.

### Human blood sampling

Blood sampling from healthy volunteers was approved by the ethical committee of Ludwig-Maximilians-Universität München (AZ 611-15), in agreement with the Declaration of Helsinki. Recruitment was randomly performed at the Biomedical Center at Ludwig-Maximilians-Universität Munich, and samples were allocated randomly to groups with no relevant covariates. Informed consent was obtained from all participants.

### Isolation of human and mouse neutrophils

Human neutrophils were isolated from healthy blood donors using either Polymorphprep (Axis Shield) or an EasySep direct human neutrophil isolation kit (StemCell Technologies) according to the manufacturers' protocols. Bone marrow mouse neutrophils were isolated using an EasySep mouse neutrophil enrichment kit (StemCell Technologies) according to manufacturer's protocol.

### S100A8/S100A9 release

For the in vivo release assay, recombinant mouse TNF (500 ng per mouse; R&D Systems) was injected into the scrotums of anesthetized male WT, $Gsdmd^{-/-}$, $Casp1^{-/-}Casp11^{-/-}$ and $Sele^{-/-}$ mice, as previously described[3]. In a second set of experiments, WT mice were pretreated with an intraperitoneal injection of MCC950 (10 mg per kg (body weight); Invivogen) or vehicle control 1 h before TNF injection. Mouse blood was collected via retroorbital bleeding before and 2 h after TNF application.

In vitro release of S100A8/S100A9 was assessed as described previously[3]. Briefly, bone marrow neutrophils were isolated from WT, $Gsdmd^{-/-}$, $Casp1^{-/-}Casp11^{-/-}$ and $Kcna3^{-/-}$ mice. Glass cover slips were coated with recombinant mouse E-selectin (rmCD62E-Fc chimera, 10 μg ml$^{-1}$; R&D Systems) or PBS/0.1% bovine serum albumin (BSA) at 4 °C overnight, blocked with 5% casein (Sigma-Aldrich) and washed with PBS. Neutrophils ($5 \times 10^5$) were reconstituted in 500 μl of HBSS and incubated under shaking conditions on coated slides for 10 min at 37 °C with 5% CO$_2$. Depending on the specific groups, cells were pretreated with MCC950 (1 μM, 30 min), PAP-1 (50 nM, 10 min; Sigma-Aldrich), ibrutinib (0.6 μM, 30 min; SelleckChem) or respective vehicle controls at 37 °C and 5% CO$_2$ and/or stimulated with ATP (3 mM; Sigma-Aldrich), nigericin (10 μM; InvivoGen) or a combination of LPS and nigericin (1 μg ml$^{-1}$ and 10 μM, respectively; both InvivoGen). For human in vitro release assays, $5 \times 10^5$ neutrophils were stimulated with recombinant

human E-selectin, recombinant human endoglycan, recombinant human MAdCAM (all 1 μg ml$^{-1}$; R&D Systems) or PBS. Finally, cellular supernatants and serum probes were analyzed by ELISA to determine the concentrations of S100A8/S100A9, as previously described[51].

### Quantification of intracellular S100A8/S100A9 levels

Isolated bone marrow neutrophils ($1 \times 10^6$ cells per 100 μl) were lysed with modified RIPA lysing buffer (150 mM NaCl, 1% Triton X-100 (Applichem), 0.5% sodium deoxycholate (Sigma-Aldrich), 50 mM Tris-HCl (pH 7.3; Merck Millipore) and 2 mM EDTA (Merck Millipore) supplemented with protease/phosphatase inhibitors (Cell Signaling) and 1× Laemmli sample buffer), homogenized and boiled (95 °C, 5 min). Proteins were resolved by SDS–PAGE and electrophoretically transferred to PVDF membranes. After blocking in LI-COR blocking solution, membranes were incubated overnight with primary antibodies. The following primary antibodies were used for detection: polyclonal rabbit anti-mouse S100A8 and polyclonal rabbit anti-mouse S100A9 (1 μg ml$^{-1}$ and 5 μg ml$^{-1}$, respectively; provided by T. Vogl, University of Münster, Germany) and mouse anti-GAPDH (0.15 μg ml$^{-1}$; Calbiochem). Goat anti-rabbit IRDye 800CW and goat anti-mouse IRDye 680RD (0.1 μg ml$^{-1}$; both LI-COR Bioscience) secondary antibodies were used to detect the respective proteins on an Odyssey CLx (LI-COR Bioscience). Blots were analyzed with Image Studio software, and intensities of S100A8/S100A9 were normalized to that of the loading control (GAPDH).

### Mouse and human neutrophil rolling velocities

Ibidi flow chambers (μ Slide VI$^{0.1}$) were coated with recombinant mouse E-selectin/recombinant human E-selectin (20 μg ml$^{-1}$ and 5 μg ml$^{-1}$, respectively; both R&D Systems) and recombinant mouse ICAM-1/recombinant human ICAM-1 (15 μg ml$^{-1}$ and 4 μg ml$^{-1}$, respectively; both R&D Systems) for 3 h at room temperature and blocked with 5% casein overnight at 4 °C. Isolated mouse bone marrow neutrophils ($5 \times 10^5$ cells per ml) or isolated human neutrophils ($5 \times 10^5$ cells per ml) were diluted in prewarmed HBSS (37 °C) and perfused through the flow chamber with a high precision pump (Harvard Apparatus) at a shear stress level of 2 dyn cm$^{-2}$. For NLRP3 inflammasome inhibition, cells were preincubated for 30 min with MCC950 (1 μM) or vehicle control at 37 °C and 5% CO$_2$ before the experiment. Flow assays were conducted as previously described[12].

### Caspase 1 and GSDMD cleavage

Isolated human neutrophils ($1 \times 10^7$) were stimulated for 10 min with soluble recombinant human E-selectin (1 μg ml$^{-1}$) or PBS as a control. To inhibit TRL4 receptor signaling, cells were treated with both the TLR4 inhibitor TAC242 (1 μg ml$^{-1}$; Merck Millipore) and paquinimod (10 μg ml$^{-1}$; Active Biotech) or vehicle control 5 min before stimulation. For NLRP3 inflammasome inhibition, cells were preincubated with MCC950 (1 μM) or vehicle control for 30 min at 37 °C and 5% CO$_2$. To analyze the contribution of K$^+$ efflux or of K$_V$1.3 on rapid caspase 1 and GSDMD cleavage, neutrophils were stimulated in high K$^+$-containing PBS (137 mM KCl) or were pretreated with PAP-1 (50 nM) for 10 min at 37 °C and 5% CO$_2$, respectively. Stimulation was stopped by the addition of ice-cold HBSS, and supernatants and cell pellets were collected. Cell pellets were homogenized in lysis buffer, and protein extraction of supernatants was performed with chloroform–methanol or trichloracetate (Sigma-Aldrich) precipitation. Samples were resolved by SDS–PAGE gels and electrophoretically transferred onto PVDF membranes. The following primary and secondary antibodies were used: polyclonal rabbit anti-human caspase 1 (1:1,000; Cell Signaling), polyclonal rabbit anti-human GSDMD (1:2,000; Cell Signaling) and mouse anti-human GAPDH (0.15 μg ml$^{-1}$). Goat anti-mouse IRDye 680RD- and goat anti-rabbit IRDye 800CW-coupled secondary antibodies (0.1 μg ml$^{-1}$) were used for detection. Intensities of processed forms of GSDMD or caspase 1 were normalized to the expression of the loading control (GAPDH).

## Confocal and STED microscopy of fixed neutrophils

To test E-selectin-dependent caspase 1 activation, we used the FAM-FLICA dye (ImmunoChemistry Technologies). Briefly, $3 \times 10^5$ isolated human neutrophils were loaded with FLICA in HBSS buffer for 30 min at 37 °C and 5% $CO_2$ and stimulated with soluble recombinant human E-selectin (1 µg ml$^{-1}$) or PBS for 10 min. To block caspase 1 activity, cells were incubated with VX-765 (10 µM; InvivoGen) for 10 min before application of FLICA dye. Stimulation was stopped by the addition of ice-cold 5 mM EDTA/HBSS. Cells were washed, transferred onto Ibidi eight-well removable slides precoated with 0.01% poly-L-lysine solution (Sigma-Aldrich) for 45 min and allowed to settle for 20 min. Cells were fixed for 15 min using the provided fixative, stained at room temperature with DyLight 649-conjugated WGA (10 µg ml$^{-1}$; EY Laboratories) for 10 min and DAPI (1 µg ml$^{-1}$; Invitrogen) for 5 min and embedded using VectaShield Antifade mounting medium (Vector Laboratories).

For studying GSDMD cleavage, ASC oligomerization or CHMP4B localization, $3 \times 10^5$ isolated neutrophils were transferred onto poly-L-lysine-coated Ibidi eight-well removable slides and allowed to settle for 20 min. For GSDMD cleavage experiments, cells were preincubated with MCC950 (1 µM) or vehicle control for 30 min at 37 °C and 5% $CO_2$. E-selectin (mouse or human, 1 µg ml$^{-1}$) or PBS (control) was added for 10 min before cells were fixed with 2% paraformaldehyde for 15 min, followed by permeabilization with 0.1% Triton X-100/2% BSA for 1 h. Slides were then incubated with monoclonal rabbit anti-human cleaved GSDMD-NT (2 µg ml$^{-1}$; Abcam), polyclonal rabbit anti-mouse ASC (5 µg ml$^{-1}$; Adipogen) or polyclonal rabbit anti-human CHMP4B (5 µg ml$^{-1}$; Abcam) in antibody diluent solution (Dako) at 4 °C overnight, washed and incubated with donkey anti-rabbit Alexa 488 (5 µg ml$^{-1}$; Thermo Fisher Scientific) or goat anti-rabbit abberior STAR 635P (2 µg ml$^{-1}$; Abberior), respectively, for 1 h at room temperature. Cell nuclei were stained with DAPI (1 µg ml$^{-1}$), and slides were embedded with VectaShield Antifade mounting medium. All images were acquired on a Leica SP8X WLL microscope (equipped with HC PL APO ×93/1.3-NA glycerol immersion, HC PL APO ×63/1.20-NA water immersion and HC PL APO ×40/1.30-NA oil immersion objectives), and semiautomated single-cell analysis was performed using ImageJ software[52]. For analysis of caspase 1 activity, WGA signal was used to define single-cell masks, and overall mean FLICA signal was analyzed in each cell. Average FLICA and ASC speck areas per cell were determined after automatic thresholding (RenyiEntropy) in the FLICA/ASC channels, respectively. GSDMD and CHMP4B intensities/puncta were analyzed on a single-cell basis using ImageJ. All representative micrographs were processed (including background subtraction and noise reduction) using ImageJ.

To analyze GSDMD-NT localization via STED microscopy, human neutrophils were stimulated with PBS (control) or E-selectin and fixed on poly-L-lysine-coated Ibidi slides, followed by incubation with CF594-labeled WGA (2 µg ml$^{-1}$; Biotium) in antibody diluent solution for 10 min at room temperature. Cells were permeabilized with 0.1% Triton X-100/2% BSA for 1 h and incubated with anti-GSDMD-NT (2 µg ml$^{-1}$) in antibody diluent solution at 4 °C overnight and secondary goat anti-rabbit abberior STAR 635P (2 µg ml$^{-1}$) for 1 h at room temperature. Samples were embedded in VectaShield Antifade mounting medium. Z-stack images of 0.33-µm step size were acquired on a Leica SP8X WLL super-resolution microscope (STED module) equipped with an HC PL APO CS2 ×100/1.4-NA oil immersion STED white objective and STED 592 and STED 775 depletion lasers. Tau (τ)-STED images were obtained with LASX software adopting FLIM to improve STED resolution. Single-cell analysis was performed using ImageJ software[52]. MFI values for GSDMD-NT signal were analyzed per z plane in the plasma membrane compartment and in the cytosol and averaged. A plasma membrane translocation index was calculated for each cell (MFI$_{GSDMD-NT}$ (plasma membrane)/MFI$_{GSDMD-NT}$ (cytosol)) and statistically analyzed. All representative micrographs were processed (including background subtraction and noise reduction, filtering and contrast enhancing) using ImageJ.

## IL-1β release

Glass cover slips were coated and handled as described for the S100A8/S100A9 release assay. IL-1β concentrations in supernatants were analyzed by ELISA according to the manufacturer's protocol (Quantkine, R&D Systems).

## Sample preparation for mass spectrometry

Neutrophils ($1 \times 10^6$ ml$^{-1}$) were stimulated with PBS or E-selectin in HBSS buffer (BSA free), and supernatants were collected. Proteins in the supernatant were precipitated at −20 °C overnight with acetone (1:4 (sample:acetone)) and washed twice with ice-cold 80% acetone. Precipitated proteins were dissolved in 8 M urea in 50 mM Tris (pH 8), and disulfide bridges were reduced and alkylated with 10 mM Bond-Breaker TCEP solution and 30 mM chloroacetamide. Digestion was performed overnight at a concentration of 2 M urea with LysC/trypsin mixture (1:50 (enzyme:protein)). Protein digestion was stopped with 1% formic acid, and peptides were desalted on in-house-produced SDB-RPS StageTips.

## Liquid chromatography–tandem mass spectrometry measurements

Samples were measured by liquid chromatography–tandem mass spectrometry. For each sample, 300 ng of tryptic peptides was loaded on an in-house-packed C18 analytical column. Peptides were separated on a nanoflow ultrahigh performance liquid chromatography system (EASY-nLC 1200, Thermo Scientific) using a 120-min non-linear gradient. Eluting peptides were transferred on-line to the quadrupole orbitrap tandem mass spectrometer (Orbitrap Exploris 480, Thermo Scientific) by a nanoelectrospray ionization source (Thermo Scientific). The mass spectrometer was operated in data-independent acquisition mode. Full spectra were recorded with a scan range of 300–1,650 $m/z$ and a resolution of 120,000. The automatic gain control target was set to $3 \times 10^6$ ions and a maximum injection time of 60 ms. MS2 data-independent acquisition scans were acquired at a resolution of 30,000, an automatic gain control target of $1 \times 10^6$ and a maximum injection time of 54 ms. Data-independent acquisition isolation windows were of variable sizes ranging from 16 to 524 $m/z$ and were fragmented with a higher-energy collisional dissociation energy of 27%.

## Quantification and statistical analysis of mass spectrometry data

Peptide identification and protein quantification were performed in library-free mode using DIA-NN 1.8.1 (ref. [53]). Data-independent acquisition tandem mass spectrometry spectra were searched against the UniProt SWISSPROT human proteome (version from 15 February 2023) and filtered on peptide level at a false discovery rate of 1%. Trypsin was set as the digestion enzyme with peptide lengths set from 7 to 30 amino acids, a fixed modification of cysteines (carbamidomethylation) and a maximum of one missed cleavage. The match-between-runs option was enabled. Using the DIA-NN main output table, identified precursors were filtered for Lib.QValue and Lib.PG.Value of <1%. Label-free quantification was performed with the R plugin diann using the maxLFQ algorithm[54] for proteotypic peptides only.

Statistical analyses were performed with the Perseus computational platform (version 1.6.15)[55], and protein LFQ intensities were log$_2$ transformed and filtered for at least three valid values in at least one condition. Random values were used to impute missing data drawn from normal distributions with a 1.8 standard deviation downshift and 0.3 standard deviation spread for each sample. Significantly released proteins were identified by Welch's two-sample $t$-tests (significance: false discovery rate > 0.05, fudge factor S0 = 0.1). Secretory signatures were functionally characterized by one-dimensional annotation enrichment analysis (Benjamin–Hochberg false discovery rate = 0.02).

The following annotations were manually added: alarmins, cytosolic small alarmins, cytosolic big alarmins, granules, azurophilic granules, specific granules and gelatinase granules. The mass spectrometry proteomics data have been deposited to the ProteomeXchange Consortium via the PRIDE[56] partner repository with the dataset identifier PXD041652.

### Detection of granule release via flow cytometry

Isolated human neutrophils ($1 \times 10^6$) were stimulated with PBS (control), E-selectin or phorbol myristate acetate (positive control; 100 nM; Sigma-Aldrich) in HBSS for 10 min at 37 °C under shaking conditions. The reaction was stopped by the addition of FACS Lysing solution (BD Bioscience), and cells were stained with antibodies to CD15, CD63, CD66 and CD11b and their corresponding isotype controls (all Biolegend; 5 μg ml⁻¹). Translocation of granule components was assessed using a CytoFlex S flow cytometer (Beckmann Coulter) and FlowJo software.

### ASC oligomerization

Isolated bone marrow neutrophils were stimulated with recombinant mouse E-selectin (1 μg ml⁻¹) in BSA-free HBSS for 10 min at 37 °C under shaking conditions and subsequently centrifuged for 5 min at 300$g$ and 4 °C. Cells were processed as described elsewhere[57]. The following primary and secondary antibodies were used: polyclonal rabbit anti-mouse ASC (1 μg ml⁻¹; Adipogen), mouse anti-mouse GAPDH (0.15 μg ml⁻¹), goat anti-rabbit IRDye 800CW and goat anti-mouse IRDye 680RD (0.1 μg ml⁻¹). Intensities of ASC oligomers were normalized to those of GAPDH in non-cross-linked loading controls.

### Patch-clamp recordings of human neutrophils

Membrane potential ($V_m$) recordings of isolated human neutrophils were performed in current clamp mode at room temperature using the whole-cell variation of the patch-clamp technique and a HEKA EPC10 USB patch-clamp amplifier (HEKA Elektronik) in combination with Patchmaster software. Origin Pro 2019G software was used for data analysis. Pipettes with a resistance of 4–6 MΩ were fabricated with a DMZ Universal Microelectrode Puller (Zeitz-Instrumente Vertriebs) and filled with intracellular solution (consisting of 140 mM KF, 10 mM NaCl, 2.0 mM MgCl₂, 1.0 mM CaCl₂, 10 mM HEPES and 10 mM EGT, pH 7.2 KOH). After isolation, cells were transferred into extracellular solution (containing 140 mM NaCl, 2.8 mM KCl, 2.0 mM MgCl₂, 1.0 mM CaCl₂, 10 mM HEPES and 11 mM glucose, pH 7.4 NaOH). Once the whole-cell configuration was established, baseline membrane potential was recorded for 1–2 min. Cells were then superfused with extracellular solution containing recombinant human E-selectin (1 μg ml⁻¹) in the presence or absence of PAP-1 (50 nM) or disulfiram (30 μM) for 1 min, and the membrane potential was recorded for an additional 3.5 min. The starting membrane potential was calculated as the 30-s mean $V_m$ value immediately before superfusion of E-selectin. Changes in membrane potential after E-selectin stimulation were calculated as $\Delta V_m$ between the 30-s mean values at different time intervals as indicated and the starting membrane potential (−30 to 0 s). $V_m$ values were corrected for liquid junction potential.

### NLRP3 pY-IP

Tyrosine phosphorylation of NLRP3 was assessed by pY-IP and subsequent western blotting, as previously described[18]. Isolated bone marrow neutrophils ($20 \times 10^6$ cells) were preincubated with ibrutinib (0.6 μM) or DMSO for 30 min at 37 °C and 5% CO₂ and subsequently stimulated with E-selectin (1 μg ml⁻¹) or PBS (control) for 7 min. Cells were washed with PBS and lysed in 1 ml of mammalian protein extraction reagent buffer (Thermo Fisher Scientific) supplemented with phosphatase inhibitors (phosphatase inhibitor cocktails 2 + 3; Sigma-Aldrich) and complete Mini EDTA-free protease inhibitors (Roche). IP of tyrosine-phosphorylated proteins was performed by incubating the lysates with 20 μl of pY-MultiMab rabbit mAb mix

conjugated to magnetic beads (P-Tyr-1000, Cell Signaling) for 2 h at 4 °C with gentle mixing. After washing, bound protein was retained by incubation in 2× Laemmli buffer (Roti-Load 1, reducing, 4× stock, Carl Roth) at 95 °C for 5 min. IP and input control samples were subjected to SDS–PAGE and western blotting. Western blots were probed with the following antibodies diluted in 5% milk/ TBS plus 0.1% Tween20: mouse anti-NLRP3/NALP3 (1:1,000; Adipogen), mouse anti-GAPDH (0.15 μg ml⁻¹) and goat anti-mouse-HRP (1:15,000; Jackson ImmunoResearch). NLRP3 signal was quantified by densitometry using ImageJ. pY-NLRP3 signals as measured in IP samples were normalized to relative expression levels quantified from input controls.

### Live imaging of neutrophils

Isolated neutrophils were stained with Cell Tracker Green CMFDA (Invitrogen) according to the manufacturer's protocol. For NLRP3 inflammasome inhibition, cells were preincubated with MCC950 (1 μM) for 30 min at 37 °C and 5% CO₂. Neutrophils ($1 \times 10^6$) in 300 μl of HBSS were transferred onto poly-L-lysine-coated Ibidi eight-well glass-bottom slides and allowed to settle for 20 min before soluble recombinant mouse E-selectin/recombinant human E-selectin (both 1 μg ml⁻¹) was added to the samples. PI (0.33 μg ml⁻¹; Invitrogen) was used to track pore formation and was applied to the samples concomitantly with E-selectin or with a time delay, as indicated. Samples were imaged for 20 min with 20-s frame intervals by confocal microscopy with a Leica SP8X WLL microscope equipped with an HC PL APO ×40/1.30-NA oil immersion objective. Images were analyzed using ImageJ. PI intensity was quantified by subtracting the PI intensity at time zero ($F_0$) from the PI intensity at the depicted time point ($F_t$) and dividing by the PI intensity at time zero: $PI = (F_t - F_0)/F_0$.

### LDH release assay

Isolated human neutrophils ($5 \times 10^5$) were stimulated with PBS or soluble recombinant human E-selectin (1 μg ml⁻¹) in 500 μl of HBSS for 10, 30 or 180 min. In additional experiments, $5 \times 10^5$ human neutrophils or monocytes were stimulated with PBS or LPS/nigericin (1 μg ml⁻¹ and 10 μM, respectively) for 30 min. Furthermore, $5 \times 10^5$ human neutrophils or monocytes were primed with LPS for 2.5 h and stimulated with nigericin for 30 min (180-min stimulation). LDH levels in the supernatants were quantified using a CyQuant LDH Cytotoxicity Assay (Thermo Fisher Scientific), according to the manufacturer's protocol, and measured using a Tecan SPARK 10M microplate reader.

### Mouse model of acute inflammation in the cremaster muscle

Intravital microscopy of the mouse cremaster muscle was performed as previously described[3]. Rolling flux fraction (number of rolling cells normalized to complete leukocyte flux[29]), leukocyte rolling velocities, number of adherent cells per mm², vessel diameter and vessel length were determined on the basis of the generated movies using ImageJ software.

To assess neutrophil extravasation, cremaster muscles were dissected 2 h after intrascrotal TNF application and stained for perivascular neutrophils as previously described[12].

### Surface expression of adhesion-relevant molecules in mouse neutrophils

Neutrophils were stained with antibodies to CD11a (eBioscience), CD11b (BioLegend), CD18 (Pharmingen) and CXCR2 (R&D Systems) or their corresponding isotype controls (all 5 μg ml⁻¹). Samples were fixed with FACS Lysing solution and analyzed using a Beckman Coulter Gallios flow cytometer with Kaluza Flow analysis software (Beckman Coulter). Neutrophils were defined as Ly6G⁺ cells (5 μg ml⁻¹; Biolegend).

### Statistics

Data are presented as mean ± s.e.m., cumulative frequency or representative images, as detailed in the figure legends. Group sizes were

chosen based on previous experiments. GraphPad Prism 7 software (GraphPad Software) and Adobe Illustrator were used to analyze data and illustrate graphs. Statistical tests were performed according to the number of groups being compared. Data collection and analysis were not performed blind to the conditions of the experiments. No data were excluded from statistical analyses. For pairwise comparisons, an unpaired or paired Student's *t*-test was performed, and for more than two experimental groups, either a one-way or two-way ANOVA with a Tukey's, Dunnetts's or Sidak's post hoc test was performed. Data distribution was assumed to be normal, but this was not formally tested. *P* values of <0.05 were considered statistically significant. No statistical methods were used to predetermine sample sizes, but our sample sizes are similar to those reported in previous publications[3,12].

### Reporting summary

Further information on research design is available in the Nature Portfolio Reporting Summary linked to this article.

### Data availability

Data-independent acquisition tandem mass spectrometry spectra were searched against the UniProt SWISSPROT human proteome (version from 15 February 2023). Mass spectrometry proteomics data have been deposited in the ProteomeXchange Consortium via the PRIDE partner repository with the dataset identifier PXD041652. Source data are provided with this paper. All other data that support the findings of this study are present in the article and Supplementary Information or from the corresponding author upon reasonable request.

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

### Acknowledgements

This work was supported by the German Research Foundation collaborative research grants CRC914 (projects A01 (M.M.), B01 (M.S.) and B11N (M.P.)), TRR332 (# 449437943; projects B5 (Johannes Roth), C2 (M.S.) and C7 (T.V.)), TRR359 (#491676693; project B02 (M.S. and R.I.)), CRC1450 (projects C01 (Johannes Roth) and C03 (T.S.)) and CRU342 (projects P3 (Johannes Roth) and P5 (T.S.)). We thank D. Jenne (Comprehensive Pneumology Center, Institute of Lung Biology and Disease, German Center for Lung Research) and S. Zierler (Institute of Pharmacology, Johannes Keppler University) for fruitful discussions, T. Leandersson (Immunology Group, University of Lund) for providing paquinimod and C. Scheiermann (Geneva Centre for Inflammation Research) for providing animals. In addition, we thank D. Gössel, A. Lübeck and S. D'Avis for excellent technical assistance. We are indebted to the core facilities Bioimaging and Flow Cytometry and Animal Models at the Biomedical Center (Ludwig-Maximilians-University Munich) and U. Schillinger (Gene Center and Department of Biochemistry, Ludwig-Maximilians-University Munich) for their support.

### Author contributions

M.P. and R.I. designed and performed experiments, analyzed data and wrote the manuscript. Jonas Roth, T.K., T.B., M.N., K.N., I.R., L.M.W., C.P., K.H., D.F., S.K., T.S., S.M.-A., A.B., W.L., M.M., T.V. and F.M. performed experiments and analyzed data. D.V., C.W.-S., Johannes Roth, V.H. and P.B. provided their expertise and critical reagents. M.S. designed experiments and wrote the manuscript.

### Competing interests

V.H. serves on the Scientific Advisory Board of Inflazome, Ltd. All other authors have no competing interests.

### Additional information

**Extended data** is available for this paper at https://doi.org/10.1038/s41590-023-01656-1.

**Correspondence and requests for materials** should be addressed to Markus Sperandio.

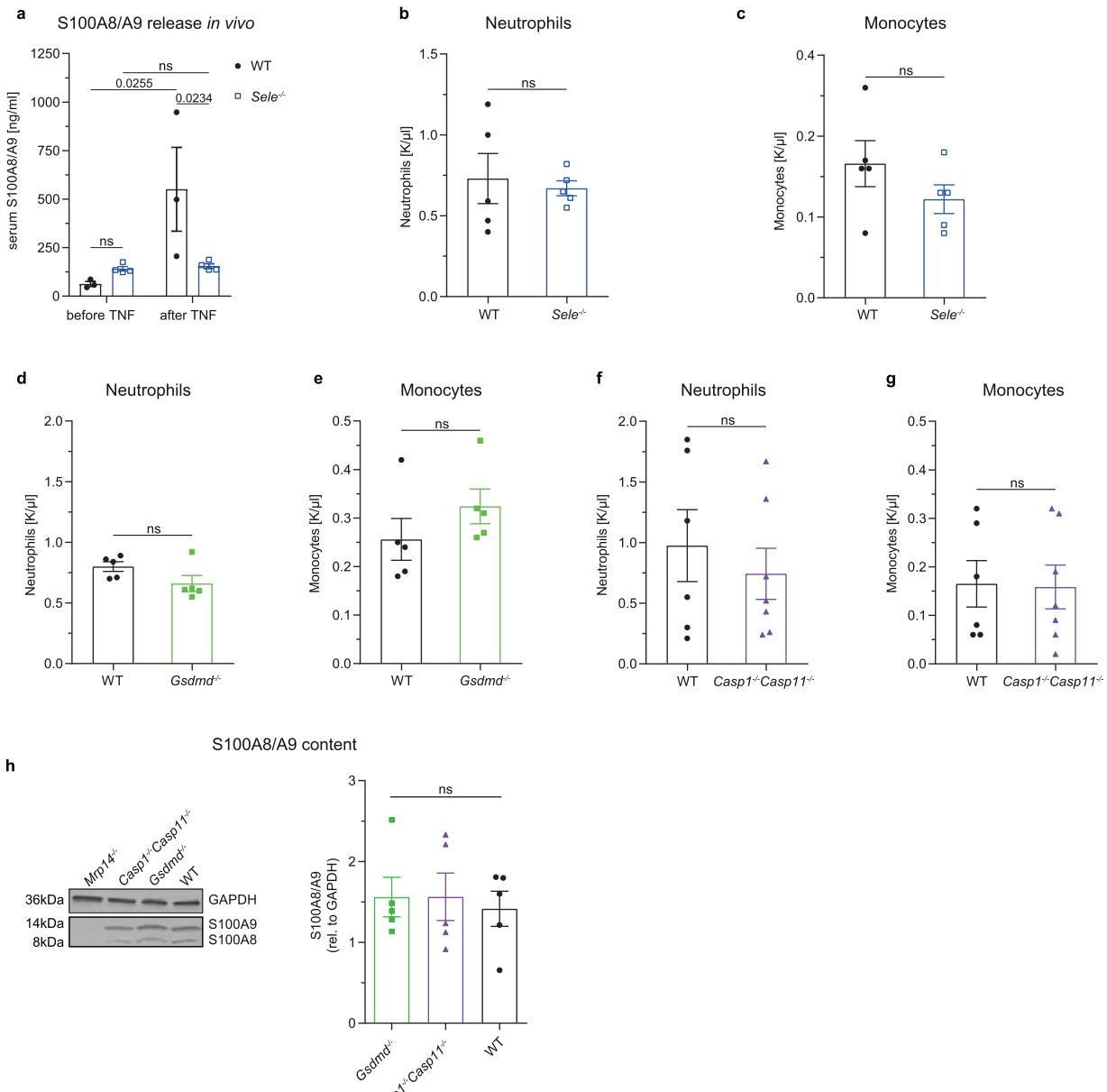

**Extended Data Fig. 1 | S100A8/A9 release depends on E-selectin, GSDMD and inflammatory caspases. a** Serum S100A8/A9 levels were determined by ELISA before and 2 h after intrascrotal TNF application to WT and *Sele*⁻/⁻ mice (n = 3 (WT), 4 (*Sele*⁻/⁻) mice/group). **b-g** Neutrophil and monocyte counts were analyzed in peripheral blood samples from untreated WT, *Sele*⁻/⁻, *Gsdmd*⁻/⁻ and *Casp1*⁻/⁻*Casp11*⁻/⁻ mice (n = 5 (WT **b, c, d, e**, *Sele*⁻/⁻, *Gsdmd*⁻/⁻), 6 (WT **f,g**, *Casp1*⁻/⁻*Casp11*⁻/⁻) mice/group). **h** Overall intracellular S100A8/A9 content in bone marrow neutrophils from WT, *Gsdmd*⁻/⁻ and *Casp1*⁻/⁻*Casp11*⁻/⁻ mice was assessed and quantified by western blotting. *Mrp14*⁻/⁻ neutrophils, which exhibit a loss of both S100A8 and A9, were used as negative control (representative western blot, n = 5 mice/group). Data are presented as mean ± s.e.m. (two-tailed unpaired student's t-test **for b-g**; one-way ANOVA, Tukey's comparison for **h** and two-way RM ANOVA, Sidak's multiple comparison for **a**) and representative Western blot for **h**; ns: not significant.

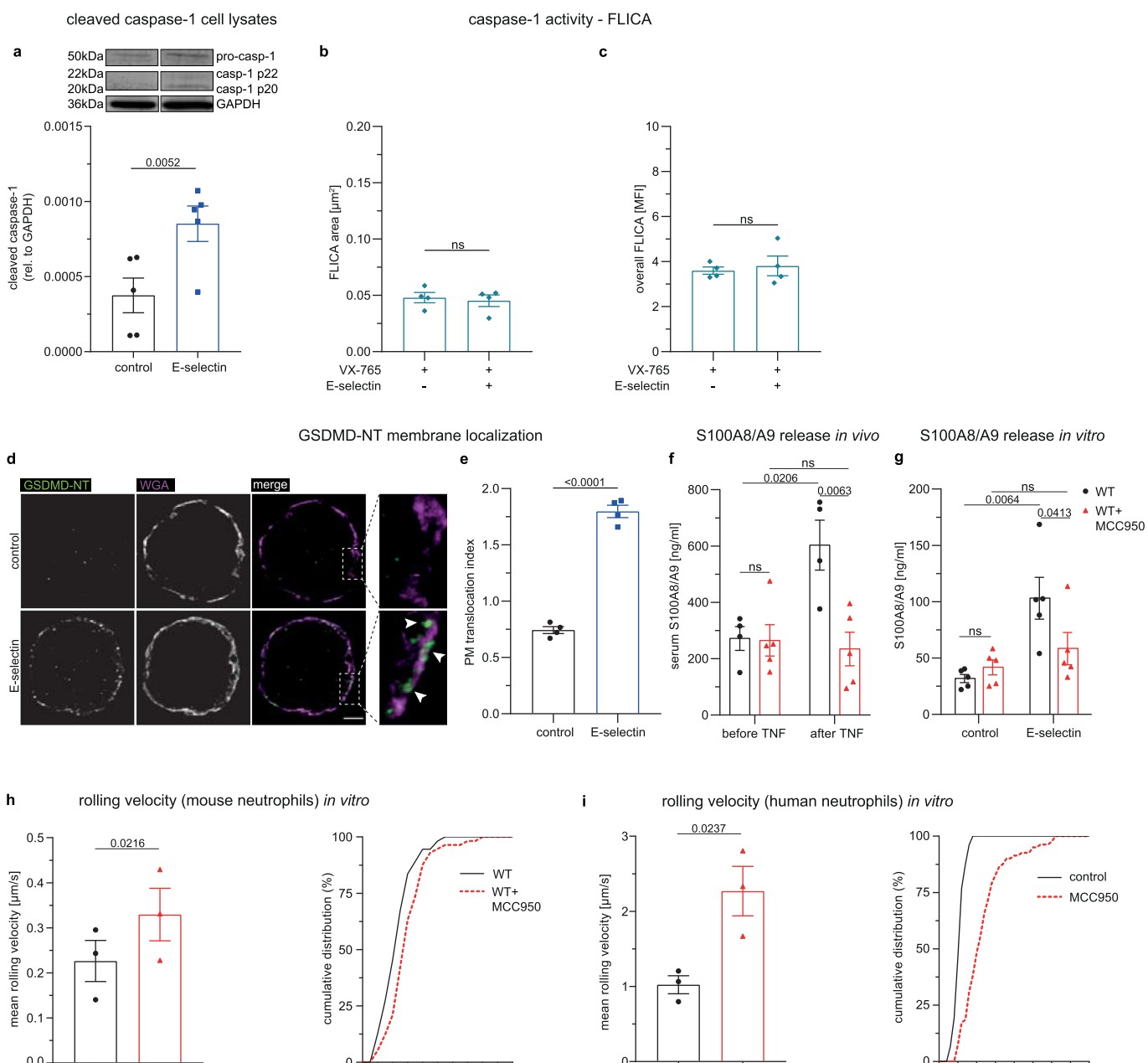

**Extended Data Fig. 2 | S100A8/A9 release is dependent on NLRP3 inflammasome. a** Amount of processed caspase-1 (casp-1 p20/casp-1 p22) in lysates of human neutrophils was determined and normalized to GAPDH (n = 5 independent experiments). **b-c** Caspase-1 activity was determined in FLICA dye-loaded, human neutrophils pretreated with VX-765 (10 μM) upon stimulation with E-selectin or PBS (control) for 10 min. Cells were analyzed by confocal microscopy. Quantification of **b** mean FLICA positive area and **c** overall mean fluorescence intensities (MFI) of FLICA signal/cell (47 (control) and 57 (E-selectin) cells of n = 4 independent experiments). **d** Representative super-resolution micrographs (maximum projection of 10 planes around cell centers; arrows: GSDMD-NT (green) at the plasma membrane (magenta)) and **e** Plasma membrane (PM) translocation index (PM vs cytosol GSDMD-NT MFI ratio) of human neutrophils treated with PBS and E-selectin (19 (control) and 19 (E-selectin) cells of n = 4 independent experiments). **f** WT mice were i.p. injected with NLRP3-inhibitor MCC950 or vehicle control 1 h prior to intrascrotal TNF

application. Serum S100A8/A9 levels were analyzed by ELISA before and 2 h after TNF stimulation (n = 4 (WT), 5 (WT + MCC950) mice/group). **g** Bone marrow neutrophils from WT mice pretreated with MCC950 (1 μM, 30 min) or vehicle control were incubated with E-selectin or PBS. (n = 5 mice/group). S100A8/A9 levels were analyzed in supernatants by ELISA. Rolling velocities of **h** MCC950 (1 μM, 30 min) or vehicle control (65 control and 68 MCC950 treated cells from n = 3 mice/group) pretreated WT neutrophils were assessed in E-selectin/ ICAM-1-coated flow chambers. Rolling velocities were determined in **i** MCC950 (1 μM, 30 min) or vehicle control (104 (control) and 81 (MCC950) cells from n = 3 independent experiments/group) pretreated human neutrophils. Data are presented as mean ± s.e.m., (two-tailed paired student's t-test for **a-c**, **e**, **h** and **i**, two-way ANOVA, Sidak's multiple comparison for **f**; two-way RM ANOVA, Sidak's multiple comparison for **g**), as representative Western blot for **a**, as representative micrographs for **d** and as cumulative distribution for **h** and **i**. ns: not significant.

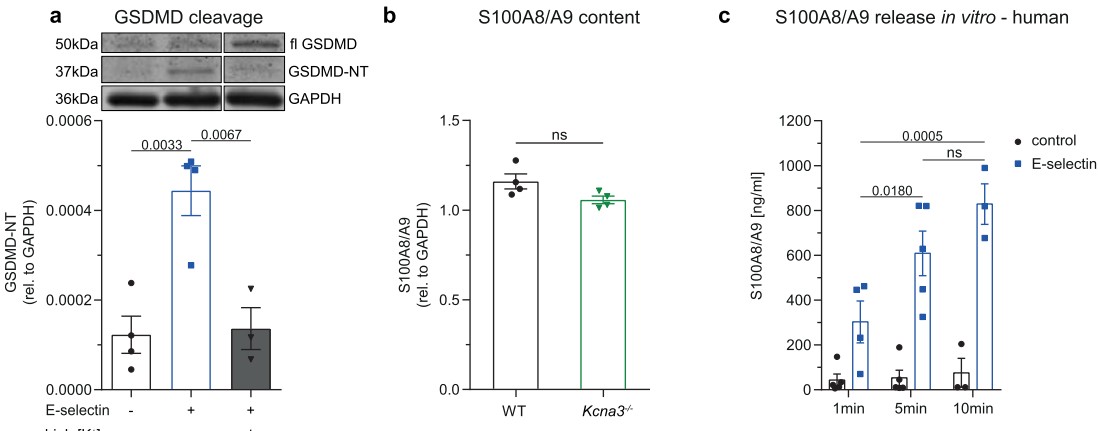

**Extended Data Fig. 3 | E-selectin triggers rapid S100A8/A9 release. a** Isolated neutrophils were treated for 10 min with PBS, E-selectin or E-selectin in the presence of high [K⁺]ₑₓ and the amount of full length GSDMD, GSDMD-NT and GAPDH was determined in the cell lysates (PBS control group and E-selectin stimulated group values and representative blot are the same as in Fig. 2g; n = 4 (control, E-selectin), 3 (E-selectin/ high [K⁺]ₑₓ) independent experiments). **b** Overall intracellular S100A8/A9 content was assessed by Western blotting in bone marrow neutrophils of WT and *Kcna3*⁻/⁻ mice (n = 4 mice/group). **c** Isolated

human neutrophils were stimulated with PBS (control) or E-selectin (1 µg ml⁻¹) for 1 min, 5 min or 10 min and S100A8/A9 levels in the supernatants were assessed by ELISA (n = 3 (control 10 min, E-selectin 10 min), 4 (E-selectin 1 min), 5 (control 1 min, control 5 min, E-selectin 5 min) individual experiments. Values from the 10 min time point (control and E-selectin) are the same as in Fig. 3g. Data are presented as mean ± s.e.m. (one-way ANOVA, Tukey's comparison for **a**, two-tailed unpaired student's t-test for **b** and two-way ANOVA, Sidak's multiple comparison for **c**), as representative Western blot for **a**, ns: not significant.

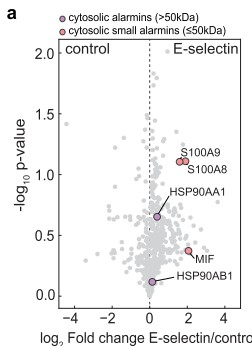

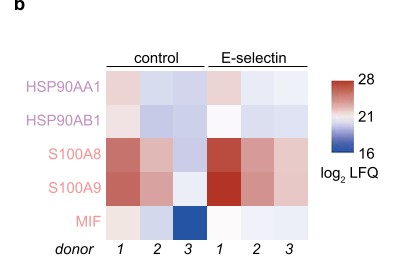

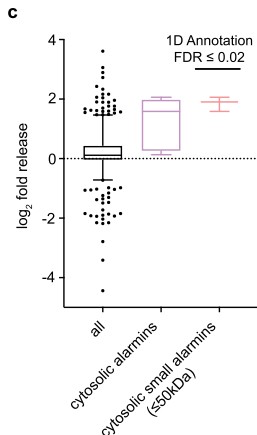

**Extended Data Fig. 4 | E-selectin stimulates secretion of cytosolic small alarmins.** Protein release analyzed by mass spectrometry-based proteomics of supernatants from E-selectin-activated versus PBS-treated isolated human neutrophils (n = 3 independent experiments). **a** Volcano plot showing differentially released proteins, cytosolic small alarmins ($M_w \leq 50$ kDa) and cytosolic alarmins ($M_w > 50$ kDa) as indicated. **b** Release of indicated proteins for each donor, **c** release of all proteins, cytosolic alarmins (no size threshold) and cytosolic small alarmins ($M_w \leq 50$ kDa). Data are presented as volcano plot for **a**, as heat map for **b**, as box-whisker-plots for **c** (50% interquartile range (IQR), median center, whiskers ranging from 5-95 percentiles and dots indicating outliers). -$\log_{10}$ transformed p-values depicted in the volcano plots were determined by Welch's two-sided t-test. Enrichment of alarmins in **c** was determined by a 1D annotation enrichment with Benjamin-Hochberg FDR = 0.02, ns: not significant.

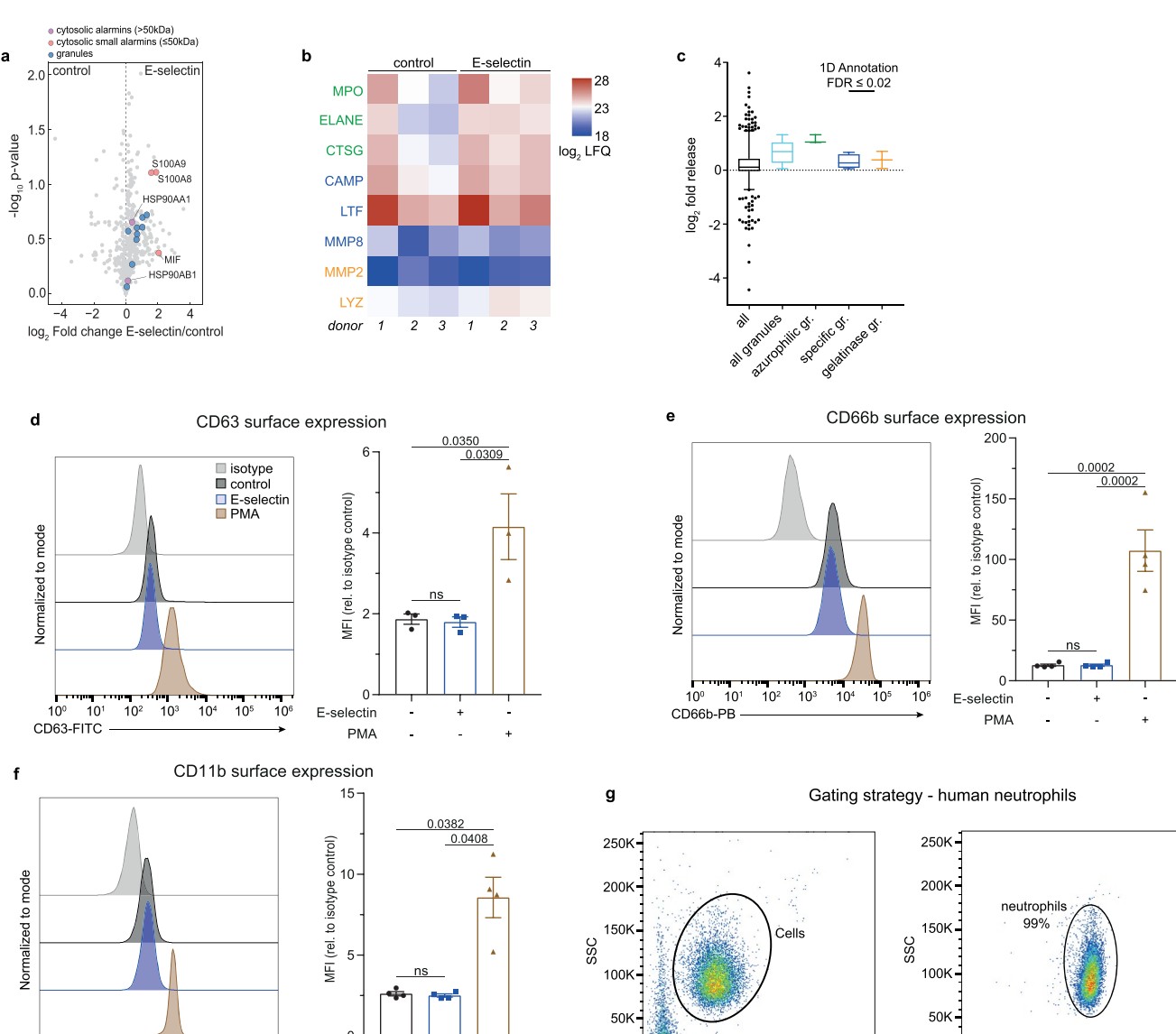

**Extended Data Fig. 5 | E-selectin stimulation does not result in granule release.** Enrichment of granule content (azurophilic gr., specific gr., gelatinase gr.) in supernatants of E-selectin (1 µg ml⁻¹) and PBS (control) stimulated human neutrophils was analyzed by mass spectrometry (n = 3 independent experiments). **a** Volcano plot showing differentially released proteins, cytosolic small alarmins (M$_w$≤50 kDa), cytosolic alarmins (M$_w$>50 kDa) and granules are indicated. **b** Release of indicated proteins for each donor, **c** release of all proteins, all granules, azurophilic granules, specific granules and gelatinase granules. **d-g** Isolated human neutrophils were stimulated with E-selectin (1 µg ml⁻¹), PMA (100 nM, positive control) or PBS (control) for 10 min. Surface

expression of **d** CD63, **e** CD66b and **f** CD11b was assessed using flow cytometry (n = 4 independent experiments). **g** gating strategy. Data are presented as mean ± s.e.m. (one-way ANOVA, Tukey's comparison for **d-f**, as volcano plot for **a**, as heat map for **b**, as box-whisker-plots for **c** (50% IQR, median center, whiskers ranging from 5-95 percentiles and dots indicating outliers), as representative flow cytometry dot blot image for **g**, and as representative histograms for **d-f**. -log$_{10}$ transformed p-values depicted in the volcano plots were determined by Welch's two-sided t-test. Enrichment of granule content in **c** was determined by a 1D annotation enrichment with Benjamin-Hochberg FDR = 0.02. ns: not significant.

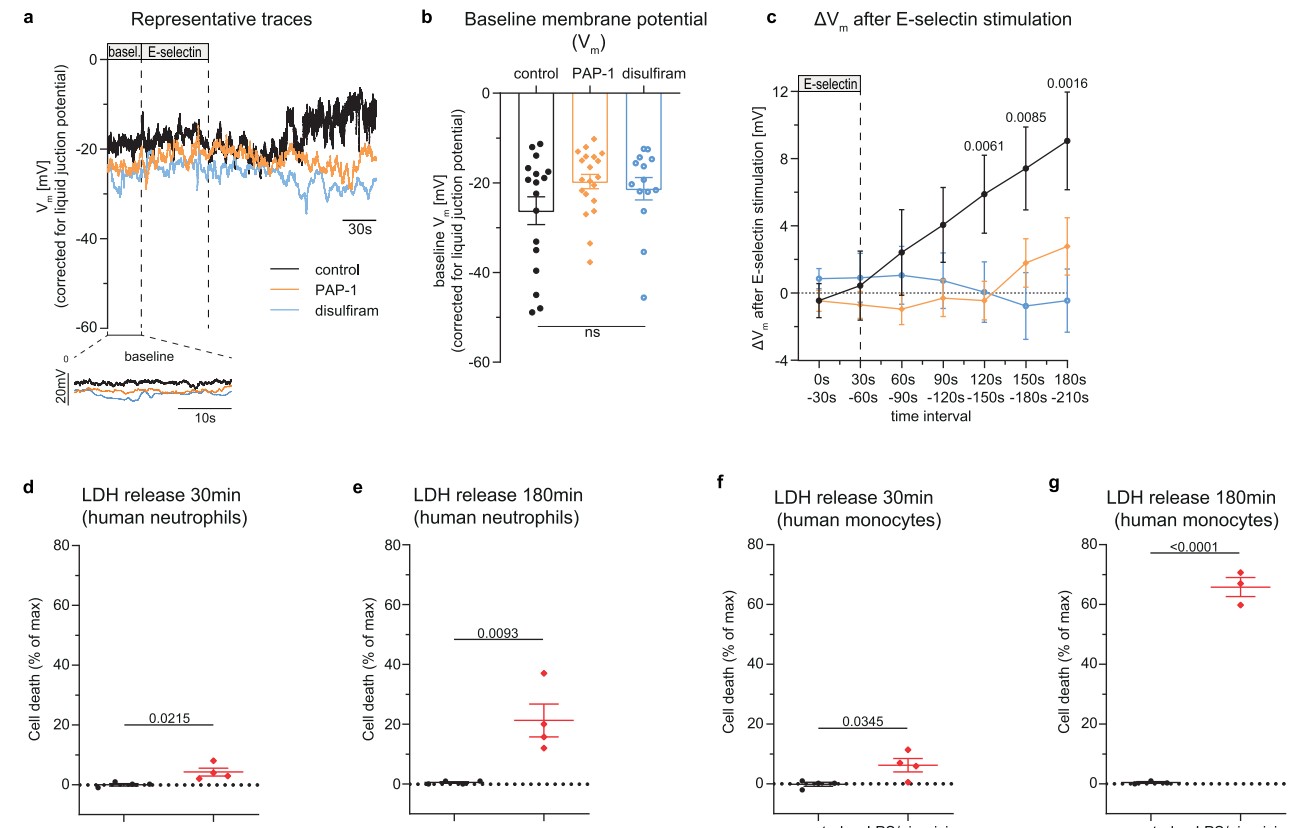

**Extended Data Fig. 6 | E-selectin induces depolarization of membrane potential in neutrophils. a-c** Changes in membrane potential of isolated human neutrophil pretreated with PAP-1 (50 nM), disulfiram (30 μM) or vehicle (control) before (baseline) and after E-selectin stimulation was measured by current clamp, **a** representative voltage membrane traces (n = 14 (disulfiram), 17 (control), 20 (Pap-1) cells of 7 different blood donors). **b** Mean baseline membrane potential $V_m$ (averaged from $t_{(-30)}$-$t_{(0)}$) and **c** changes in $V_m$ compared to baseline ($\Delta V_m$) upon E-selectin stimulation were analyzed. **d** and **e** Cell death

of human neutrophils and **f** and **g** monocytes was assessed by LDH release after stimulation with LPS/nigericin for 30 min and 180 min (n = 3 (monocytes 180 min), 4 (neutrophils 30 min, 180 min, monocytes 30 min) independent experiments). Data are presented as mean ± s.e.m. (two-tailed unpaired student's t-test for **d**-**g**, one-way ANOVA, Tukey's multiple comparison for **b** and two-way ANOVA, Tukey's multiple comparison for **c**) and as representative traces for **a**, ns: not significant.

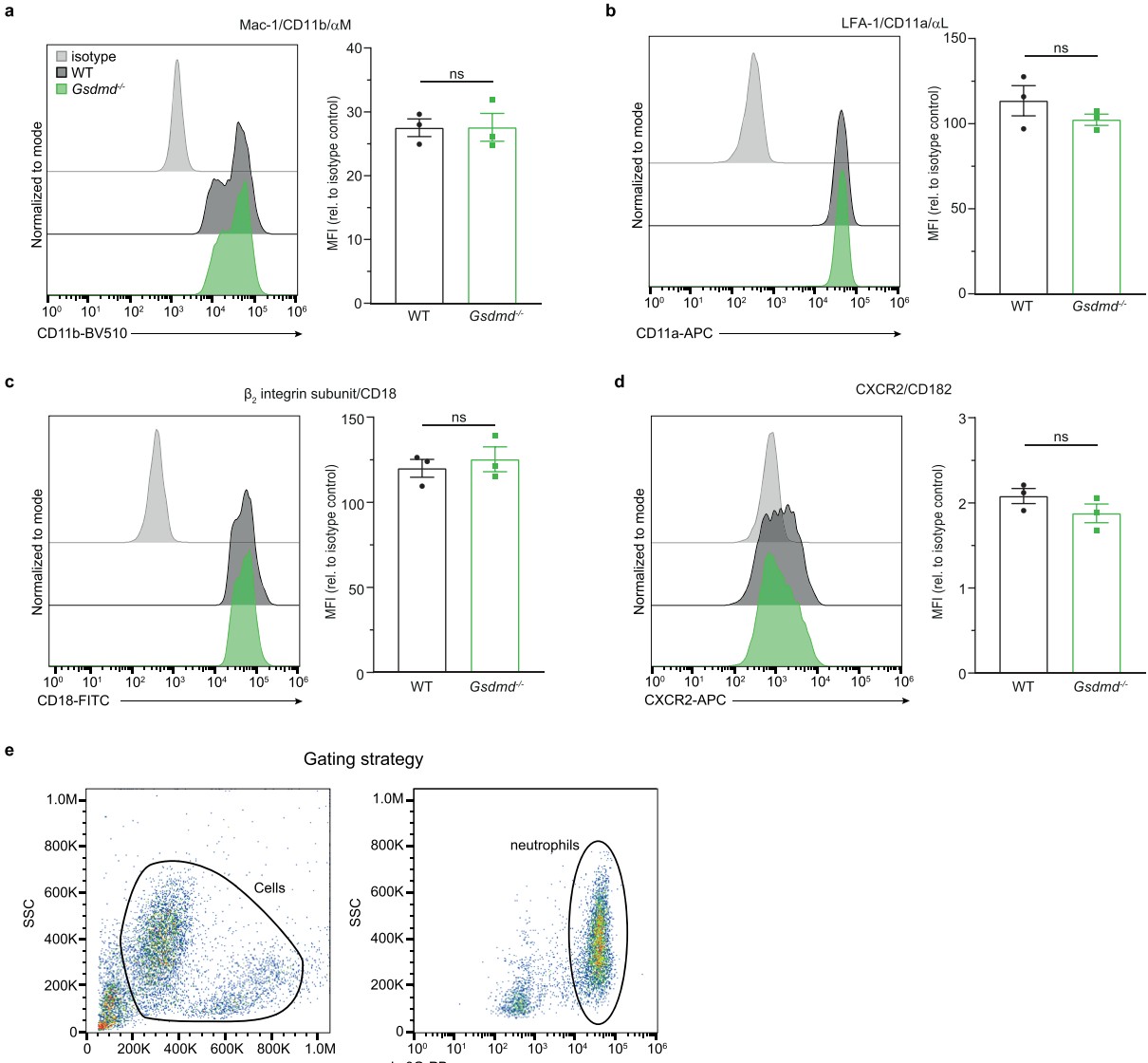

Surface expression levels of adhesion and extravasation relevant molecules
(mouse neutrophils)

**Extended Data Fig. 7 | Receptor expression of WT and *Gsdmd*-/- neutrophils.**
Surface expression levels of **a** Mac-1/CD11b/αM, **b** LFA-1/CD11a/αL, **c** β2 integrin subunit/CD18 and **d** CXCR2/CD182 were determined on bone marrow neutrophils from WT and *Gsdmd*-/- mice by flow cytometry (n = 3 mice/group). **e** Mouse neutrophils were defined as Ly6G positive cells using a PB conjugated rat anti-mouse Ly6G antibody. Data are presented as mean ± s.e.m. (two-tailed unpaired student's t-test) and as representative histograms for **a-d** and as representative flow cytometry dot blot image for **e**. ns: not significant.

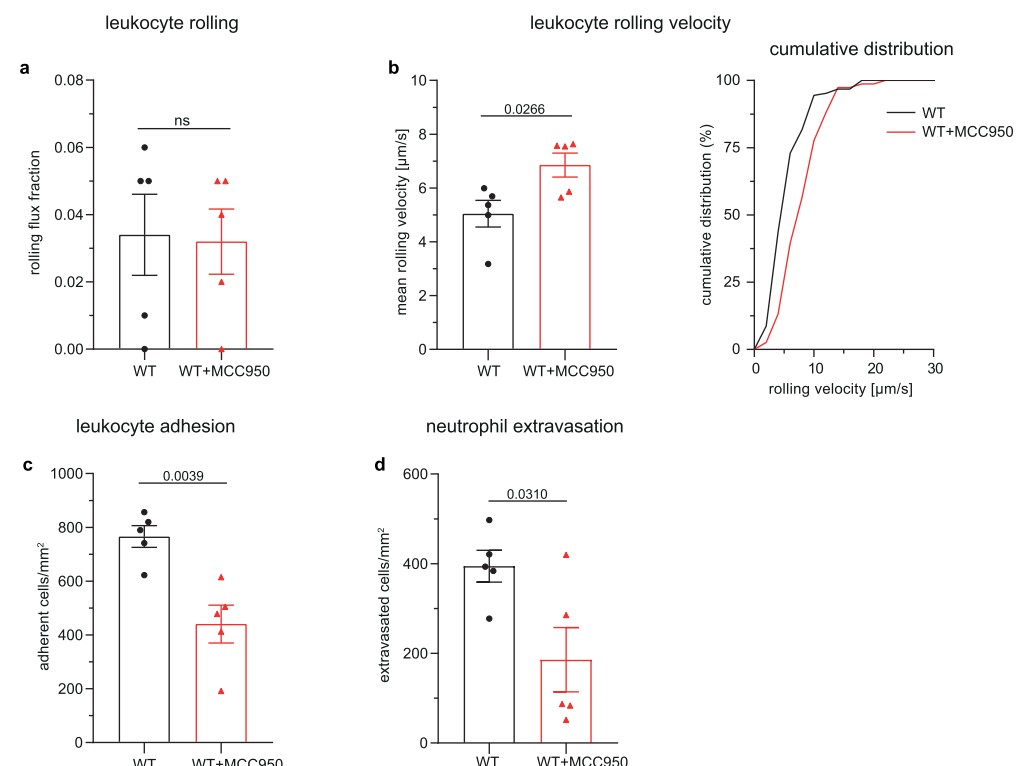

**Extended Data Fig. 8 | Neutrophil recruitment *in vivo* requires NLRP3 inflammasome activity.** Male WT mice were injected i.p. with MCC950 (10 mg kg$^{-1}$) or vehicle control 1 h before i.s. application of TNF (500 ng mouse$^{-1}$). Two hours later, intravital microscopy of postcapillary venules of the mouse cremaster muscle was performed and **a** neutrophil rolling, **b** neutrophil rolling velocity and **c** number of adherent neutrophils per vessel surface was analyzed (26 and 24 vessels, respectively, of n = 5 mice/group). **d** TNF stimulated cremaster muscles were stained with Giemsa and number of perivascular neutrophils was quantified (35 and 25 vessels, respectively, of n = 5 mice/group). Data are presented as mean ± s.e.m. (two-tailed unpaired student's t-test for **a**, **b**, **c** and **d**) and as cumulative distribution for **b**; ns: not significant.

**Extended Data Table 1 | Microvascular parameters of cremaster experiments**

| | n (mice) | n (venules) | Diameter [μm] | Centerline velocity [μm s⁻¹] | Wall shear rate [s⁻¹] | WBC [μl⁻¹] |
|---|---|---|---|---|---|---|
| **WT** | 4 | 18 | 30±1 | 2000±216 | 1606±160 | 3500±726 |
| **Gsdmd⁻/⁻** | 5 | 24 | 30±1 | 1929±269 | 1574±137 | 3796±375 |
| | | | ns. (p=0.9309) | ns. (p=0.6076) | ns. (p=0.9543) | ns. (p=0.3173) |
| **WT** | 5 | 26 | 31±1 | 1918±120 | 1598±132 | 2044±258 |
| **WT + MCC950** | 5 | 24 | 31±1 | 1738±135 | 1420±141 | 1978±216 |
| | | | ns. (p=0.6320) | ns. (p=0.3214) | ns. (p=0.3594) | ns. (p=0.8494) |

Vessel diameter, centerline velocity, wall shear rate and white blood cell count (WBC) of TNF-stimulated WT and *Gsdmd*⁻/⁻ mice and TNF-stimulated WT mice pretreated with either MCC950 or vehicle control (mean±s.e.m.; data were analyzed by two-tailed unpaired Student's I-test).

# Reporting Summary

## Statistics

For all statistical analyses, confirm that the following items are present in the figure legend, table legend, main text, or Methods section.

| n/a | Confirmed | |
|---|---|---|
| ☐ | ☒ | The exact sample size (*n*) for each experimental group/condition, given as a discrete number and unit of measurement |
| ☐ | ☒ | A statement on whether measurements were taken from distinct samples or whether the same sample was measured repeatedly |
| ☐ | ☒ | The statistical test(s) used AND whether they are one- or two-sided *Only common tests should be described solely by name; describe more complex techniques in the Methods section.* |
| ☒ | ☐ | A description of all covariates tested |
| ☐ | ☒ | A description of any assumptions or corrections, such as tests of normality and adjustment for multiple comparisons |
| ☐ | ☒ | A full description of the statistical parameters including central tendency (e.g. means) or other basic estimates (e.g. regression coefficient) AND variation (e.g. standard deviation) or associated estimates of uncertainty (e.g. confidence intervals) |
| ☐ | ☒ | For null hypothesis testing, the test statistic (e.g. *F*, *t*, *r*) with confidence intervals, effect sizes, degrees of freedom and *P* value noted *Give P values as exact values whenever suitable.* |
| ☒ | ☐ | For Bayesian analysis, information on the choice of priors and Markov chain Monte Carlo settings |
| ☒ | ☐ | For hierarchical and complex designs, identification of the appropriate level for tests and full reporting of outcomes |
| ☒ | ☐ | Estimates of effect sizes (e.g. Cohen's *d*, Pearson's *r*), indicating how they were calculated |

*Our web collection on statistics for biologists contains articles on many of the points above.*

## Software and code

Policy information about availability of computer code

| Data collection | Western blots were scanned using the Odyssey® CLx Imaging System (LI-COR); MetaMorph software was used to generate rolling velocity movies; Flow cytometry was performed using a Beckman Coulter Gallios flow cytometer or CytoFLEX S, both Beckman Coulter; LDH levels and Il1beta in the supernatants were measured using aTecan SPARK 10M microplate reader. Patch clamp recordings were conducted using a HEKA EPC10 USB patch-clamp amplifier (HEKA Elektronik, Germany) in combination with Patchmaster software (v2x91). Postcapillary venules of the mouse cremaster muscle were recorded using a Olympus WX51 intravital microscope and VirtualDub software (version 1.10.4). Flow chamber experiments were conducted on a Zeiss Axiovert 200 microscope and MetaMorph software (version 6.2r6). Analysis of neutrophil extravasation was carried out on a Leica DM2500 microscope. Fluorescence images/time laps movies were aquired on a Leica SP8X WLL confocal microscope, equipped with a STED module using LAS X software (Leica, version 3.5 and newer). Mass spectometry was carried out using a quadrupole orbitrap tandem mass spectrometer (Orbitrap Exploris 480). |
|---|---|
| Data analysis | Blots were analyzed with Image Studio software (LI-COR; Lite version 5.2); Patch clamp recordings were analyzed with Origin Pro 2019G (version 9.6.0.172). Leukocyte rolling velocities, rolling flux fraction, number of adherent cells/mm2, vessel diameter and vessel length was determined on the basis of the generated movies using ImageJ software (version 1.52v) including M-TrackJ Plugin (version 1.5.1). Fluorescence images/time laps movies were analyzed using ImageJ software (version 1.52v). Flow cytometry data were analyzed using Kaluza (version 1.2) or Flowjo software (version 10.7 and newer). Mass spectometry data quantified with the R plugin DIA-NN (version 1.8.1) and statistically analyzed with the Perseus computational plattform (version 1.6.15). GraphPad Prism software (GraphPad Software Inc., version 7.05 and newer) and Adobe Illustrator (version 27.8) were used for statistical analyses and Figure design. |

For manuscripts utilizing custom algorithms or software that are central to the research but not yet described in published literature, software must be made available to editors and reviewers. We strongly encourage code deposition in a community repository (e.g. GitHub). See the Nature Portfolio guidelines for submitting code & software for further information.

## Data

Policy information about availability of data

All manuscripts must include a data availability statement. This statement should provide the following information, where applicable:

- Accession codes, unique identifiers, or web links for publicly available datasets
- A description of any restrictions on data availability
- For clinical datasets or third party data, please ensure that the statement adheres to our policy

All data that support the findings of this study are available from the corresponding author upon reasonable request. DIA MS/MS spectra were searched against the Uniprot SWISSPROT human proteome (version from 2023/02/15). The mass spectrometry proteomics data have been deposited to the ProteomeXchange Consortium via the PRIDE partner repository with the dataset identifier PXD041652. Source data are provided with this paper.

## Research involving human participants, their data, or biological material

Policy information about studies with human participants or human data. See also policy information about sex, gender (identity/presentation), and sexual orientation and race, ethnicity and racism.

| Reporting on sex and gender | Human neutrophils were isolated from healthy male and female donors. Due to low sample numbers we did not differentiate between male and female. |
| --- | --- |
| Reporting on race, ethnicity, or other socially relevant groupings | Human neutrophils were isolated from healthy male and female donors. Due to low sample numbers we did not differentiate between race, ethnicity or social relevant groupings. |
| Population characteristics | Blood samples were taken from healthy volunteers 20-40 years of age. |
| Recruitment | Recruitment of healthy volunteers was randomly performed at the Biomedical Center at LMU Munich, Germany, through announcements at the institute's dash board. Informed consent was obtained from all participants. |
| Ethics oversight | Blood sampling from healthy volunteers was approved by the ethical committee of Ludwig-Maximilians-Universität München, Munich, Germany (Az. 611-15) in agreement with the Declaration of Helsinki. |

Note that full information on the approval of the study protocol must also be provided in the manuscript.

# Field-specific reporting

Please select the one below that is the best fit for your research. If you are not sure, read the appropriate sections before making your selection.

☒ Life sciences        ☐ Behavioural & social sciences        ☐ Ecological, evolutionary & environmental sciences

For a reference copy of the document with all sections, see nature.com/documents/nr-reporting-summary-flat.pdf

# Life sciences study design

All studies must disclose on these points even when the disclosure is negative.

| Sample size | We have not performed any sample size power calculations. In our studies, we based the sample size on optimization experiments or previously conducted experiments (in vivo and in vitro experiments) and used a minimum of 3-10 individual samples per group. Published sample sizes from our group in the literature include (Kurz AR, Pruenster M, Rohwedder I, Ramadass M, Schäfer K, Harrison U, Gouveia G, Nussbaum C, Immler R, Wiessner JR, Margraf A, Lim DS, Walzog B, Dietzel S, Moser M, Klein C, Vestweber D, Haas R, Catz SD, Sperandio M. MST1-dependent vesicle trafficking regulates neutrophil transmigration through the vascular basement membrane. J Clin Invest. 2016 Nov 1;126(11):4125-4139. doi: 10.1172/JCI87043) and (Pruenster M, Kurz AR, Chung KJ, Cao-Ehlker X, Bieber S, Nussbaum CF, Bierschenk S, Eggersmann TK, Rohwedder I, Heinig K, Immler R, Moser M, Koedel U, Gran S, McEver RP, Vestweber D, Verschoor A, Leanderson T, Chavakis T, Roth J, Vogl T, Sperandio M. Extracellular MRP8/14 is a regulator of β2 integrin-dependent neutrophil slow rolling and adhesion. Nat Commun. 2015 Apr 20;6:6915. doi: 10.1038/ncomms7915) and (Rohwedder I, Kurz ARM, Pruenster M, Immler R, Pick R, Eggersmann T, Klapproth S, Johnson JL, Alsina SM, Lowell CA, Mócsai A, Catz SD, Sperandio M. Src family kinase-mediated vesicle trafficking is critical for neutrophil basement membrane penetration. Haematologica. 2020 Jul;105(7):1845-1856. doi: 10.3324/haematol.2019.225722) |
| --- | --- |
| Data exclusions | No data was excluded. |
| Replication | All experiments were performed at least three times and were successfully replicated. The number of experiments is indicated in the respective figure legends. |
| Randomization | Mice of similar age and sex were used for all experiments except for intravital microscopy studies in the cremaster muscle, where male mice were used only. Allocation of animals to groups was performed randomly or on the basis of their genotype. Human blood samples, all obtained from healthy volunteers (please also refer to Recruitment) were allocated randomly to groups with no relevant covariates. |
| Blinding | Most of the conducted experiments were performed in a non-blinded fashion but kept as unbiased as possible. Individual in vivo and in vitro |

| Blinding | experimental series contained appropriate internal controls and normalization methods (as described in Methods) and were conducted by the same researcher to guarantee reproducibility. In vitro S100A8/A9 release assays were analyzed in a blinded fashion using numerical keys for sample and group allocation. |
|---|---|

# Reporting for specific materials, systems and methods

We require information from authors about some types of materials, experimental systems and methods used in many studies. Here, indicate whether each material, system or method listed is relevant to your study. If you are not sure if a list item applies to your research, read the appropriate section before selecting a response.

## Materials & experimental systems

| n/a | Involved in the study |
|---|---|
| ☐ | ☒ Antibodies |
| ☒ | ☐ Eukaryotic cell lines |
| ☒ | ☐ Palaeontology and archaeology |
| ☐ | ☒ Animals and other organisms |
| ☒ | ☐ Clinical data |
| ☒ | ☐ Dual use research of concern |
| ☒ | ☐ Plants |

## Methods

| n/a | Involved in the study |
|---|---|
| ☒ | ☐ ChIP-seq |
| ☐ | ☒ Flow cytometry |
| ☒ | ☐ MRI-based neuroimaging |

## Antibodies

| Antibodies used | 1) polyclonal rabbit anti-mouse MRP8 (provided by Thomas Vogl, Muenster, Germany, 1µg/ml)<br>2) polyclonal rabbit anti-mouse MRP14 (provided by Thomas Vogl, Muenster, Germany, 5µg/ml)<br>3) mouse anti GAPDH (Calbiochem, #CB1001, clone 6C5, 0.15µg/ml)<br>4) goat anti-rabbit IRDye 800CW (LI-COR Bioscience, #926-3211, 0.1µg/ml)<br>5) goat anti-mouse IRDye 680RD (LI-COR Bioscience, #926-68070, 0.1µg/ml)<br>6) polyclonal rabbit anti human caspase-1 (Cell Signaling, #2225S, 1:1000)<br>7) polyclonal rabbit anti human GSDMD (Cell Signaling, #96458S, 1:2000)<br>8) rabbit anti-human cleaved n-terminal (NT-)GSDMD antibody (Abcam, #ab215203, clone EPR20829-408, 2µg/ml)<br>9) donkey anti-rabbit Alexa488 (Thermo Fisher, #A21206, Lot: 802706, 5µg/ml)<br>10) polyclonal rabbit anti human CHMP4B (abcam, #AB135154, Lot: GR3424084-3, 5µg/ml)<br>11) goat anti rabbit abberior STAR 635P (Abberior, #2-0012-007-2; Lot: 19092016Hp, 2µg/ml)<br>12) polyclonal rabbit anti mouse ASC (Adipogen, #AG-25B-0006-C100, Lot. A42092103, clone AL177, 5µg/ml (microscopy), 1µg/ml (WB))<br>13) Phospho-Tyrosine MultiMab™ rabbit mAB mix (Cell Signaling, #14017, 1:25 (20µl))<br>14) mouse anti NLRP3 (Adipogen, #AG-20B-0014-C100, clone Cryo-2, 1:1000)<br>15) polyclonal goat anti-mouse HRP (Jackson Immuno Research, #155-035-003, 1:15000)<br>16) APC conjugated rat anti mouse CD11a (eBioscience, #170111, Lot:1911677, clone M17/4, 5µg/ml)<br>17) BV510 conjugated rat anti mouse CD11b (Biolegend #:101245, Lot: B261558, clone M1/70, 5µg/ml)<br>18) FITC conjugated rat anti mouse CD18 (Pharmingen, #553292, Lot:22531, clone C71/16, 5µg/ml)<br>19) APC conjugated rat anti mouse CXCR2 (R&D Systems, #FAB2164A, Lot: LMC0816111, clone 242216, 5µg/ml)<br>20) Pacific Blue conjugated rat anti mouse Ly6G (Biolegend, #127612, clone 1A8, 5µg/ml)<br>21) APC conjugated rat IgG2a, κ Isotype Ctrl (Biolegend, #400512, clone RT2758, 5µg/ml)<br>22) BV510 conjugated rat IgG2b, κ Isotype Ctrl (Biolegend, #400645, Lot: B167358, clone RTK4530, 5µg/ml)<br>23) FITC conjugated rat IgG2a, κ Isotype Ctrl (eBioscience, #11-4321-85, Lot: 7054789, clone eBR2a, 5µg/ml)<br>24) APC conjugated rat IgG2a, κ Isotype Ctrl (Biolegend, #400512, Lot: B238056, clone RTK2758, 5µg/ml)<br>25) FITC conjugated anti human CD63 (Biolegend, #353006, clone H5C6, 5µg/ml)<br>26) Pacific blue conjugated anti human CD66b (Biolegend, #305112, clone G10F5, 5µg/ml)<br>27) APC/Fire™ 750 conjugated mouse anti human CD11b (Biolegend, #301352, clone ICRF44, 5µg/ml)<br>28)  FITC conjugated mouse IgG1, κ Isotype Ctrl (Biolegend, #400108, clone MOPC-21, 5µg/ml)<br>29) Pacific Blue mouse conjugated IgM, κ Isotype Ctrl (Biolegend, #401619, clone MM-30, 5µg/ml)<br>30) APC/Fire™ 750 conjugated mouse IgG1, κ Isotype Ctrl (Biolegend, #400196, clone MOPC-21, 5µg/ml)<br>31) APC conjugated anti human CD15 (Biolegend, #323007, clone W6D3, 5µg/ml) |
|---|---|
| Validation | Numbering refers to the list of antibodies above:<br>1) Used as reported in: (Pruenster M, Kurz AR, Chung KJ, Cao-Ehlker X, Bieber S, Nussbaum CF, Bierschenk S, Eggersmann TK, Rohwedder I, Heinig K, Immler R, Moser M, Koedel U, Gran S, McEver RP, Vestweber D, Verschoor A, Leanderson T, Chavakis T, Roth J, Vogl T, Sperandio M. Extracellular MRP8/14 is a regulator of β2 integrin-dependent neutrophil slow rolling and adhesion. Nat Commun. 2015 Apr 20;6:6915. doi: 10.1038/ncomms7915)<br>2) See 1)<br>3) https://www.merckmillipore.com/DE/de/product/Anti-GAPDH-Mouse-mAb-6C5,EMD_BIO-CB1001?ReferrerURL=https%3A%2F%2Fwww.google.com%2F#anchor_PDS<br>4) https://www.licor.com/documents/rfm2hw40wf33p06f3ndjrcorwi5usbft<br>5) https://www.licor.com/documents/7bohf1sfzugccz22fh0um00cvz8ocizf<br>6) https://www.cellsignal.com/datasheet.jsp?productId=2225&images=1<br>7) https://www.cellsignal.com/datasheet.jsp?productId=96458&images=1<br>8) https://www.abcam.com/cleaved-n-terminal-gsdmd-antibody-epr20829-408-ab215203.html<br>9) https://www.thermofisher.com/antibody/product/Donkey-anti-Rabbit-IgG-H-L-Highly-Cross-Adsorbed-Secondary-Antibody- |

Polyclonal/A-21206
10) https://www.abcam.com/chmp4b-antibody-ab135154.html
11) https://www.sigmaaldrich.com/DE/en/product/sigma/53399
12) https://adipogen.com/ag-25b-0006-anti-asc-pab-al177.html
13) https://www.cellsignal.de/products/antibody-conjugates/phospho-tyrosine-p-tyr-1000-multimab-rabbit-mab-mix-magnetic-bead-conjugate/14017
14) https://adipogen.com/ag-20b-0014-anti-nlrp3-nalp3-mab-cryo-2.html
15) https://www.jacksonimmuno.com/catalog/products/115-035-003
16) https://www.thermofisher.com/antibody/product/CD11a-LFA-1alpha-Antibody-clone-M17-4-Monoclonal/17-0111-80
17) https://www.biolegend.com/de-de/products/brilliant-violet-510-anti-mouse-human-cd11b-antibody-7993
18) https://www.bdbiosciences.com/en-us/products/reagents/flow-cytometry-reagents/research-reagents/single-color-antibodies-ruo/fitc-rat-anti-mouse-cd18.553292
19) https://www.rndsystems.com/products/mouse-cxcr2-il-8rb-apc-conjugated-antibody-242216_fab2164a
20) https://www.biolegend.com/de-de/products/pacific-blue-anti-mouse-ly-6g-antibody-6082
21) https://www.biolegend.com/de-de/products/apc-rat-igg2a-kappa-isotype-ctrl-1838
22) https://www.biolegend.com/de-de/products/brilliant-violet-510-rat-igg2b-kappa-isotype-ctrl-8018
23) https://www.thermofisher.com/antibody/product/Rat-IgG2a-kappa-clone-eBR2a-Isotype-Control/11-4321-85
24) https://www.biolegend.com/en-gb/products/apc-rat-igg2a-kappa-isotype-ctrl-1838
25) https://www.biolegend.com/de-de/products/fitc-anti-human-cd63-antibody-7434
26) https://www.biolegend.com/de-de/products/pacific-blue-anti-human-cd66b-antibody-9583
27) https://www.biolegend.com/de-de/products/apc-fire-750-anti-human-cd11b-antibody-13561
28) https://www.biolegend.com/de-de/products/fitc-mouse-igg1-kappa-isotype-ctrl-1406
29) https://www.biolegend.com/de-de/products/pacific-blue-mouse-igm-kappa-isotype-ctrl-3167
30) https://www.biolegend.com/de-de/products/apc-fire-750-mouse-igg1-kappa-isotype-ctrl-13011
31) https://www.biolegend.com/de-de/products/apc-anti-human-cd15-ssea-1-antibody-3702

# Animals and other research organisms

Policy information about studies involving animals; ARRIVE guidelines recommended for reporting animal research, and Sex and Gender in Research

| Laboratory animals | C57BL/6NCrl (WT) mice were purchased from Charles River Laboratories (Sulzfeld, Germany). Gsdmd-/- and Casp1-/-Casp11-/- mice were obtained from Petr Broz, Epalinges, Switzerland and Veit Hornung, Munich, Germany, respectively. Mrp14-/- were provided by Johannes Roth, Muenster, Germany. Sele-/- were provided by Dietmar Vestweber, Muenster, Germany. All mice, including Kcna3-/- mice were housed at the Biomedical Center, LMU, Planegg-Martinsried, Germany. 8-25 weeks old male and female mice were used for all experiments. <br><br> Room temperature and relative humidity ranged from 20 to 22°C and from 45–55%, respectively. The light cycle was adjusted to 12h light:12 h dark period. Room air was exchanged 11 times per hour and filtered with HEPA-systems. All mice were housed in individually ventilated cages (TypII long, Tecniplast, Germany) under specified-pathogen-free conditions. Hygiene monitoring was performed every three months based on recommendations of the FELASA-14 working group. All animals had free access to water and food (irradiated, 10mm pellet; 1314P, Altromin, Netherlands). The cages were equipped with nesting material (5 × 5cm, Nestlet, Datesand, UK), a red corner house (Tecniplast, Germany) and a rodent play tunnel (7.5 × 3.0cm, Datesand, UK). Soiled bedding (LASbedding, 3–6mm, PG3, Las vendi, Germany) was removed every 7 days. |
|---|---|
| Wild animals | No wild animals were used |
| Reporting on sex | Male and female mice were used for all experiments except for intravital microscopy studies in the cremaster muscle, where male mice were used only. Allocation of animals to groups was performed on the basis of their genotype. |
| Field-collected samples | Our study did not involve samples collected from the field. |
| Ethics oversight | Animal experiments were approved by the Government of Oberbayern (AZ.: ROB-55.2-2532.Vet_02-18-22 and ROB-55.2-2532.Vet_02-17-102) and carried out in accordance with the guidelines from Directive 2010/63/EU. |

Note that full information on the approval of the study protocol must also be provided in the manuscript.

# Flow Cytometry

## Plots

Confirm that:

☒ The axis labels state the marker and fluorochrome used (e.g. CD4-FITC).

☒ The axis scales are clearly visible. Include numbers along axes only for bottom left plot of group (a 'group' is an analysis of identical markers).

☒ All plots are contour plots with outliers or pseudocolor plots.

☒ A numerical value for number of cells or percentage (with statistics) is provided.

## Methodology

| Sample preparation | Human neutrophils were isolated from healthy blood donors using Polymorphprep (Axis Shield) or EasySepTM human |
|---|---|

| | |
|---|---|
| Sample preparation | neutrophil direct isolation kit (STEMCELL TECHNOLOGIES) according to manufacturer's protocols; Bone marrow mouse neutrophils were isolated using EasySepTM mouse neutrophil enrichment kit (STEMCELL TECHNOLOGIES) according to manufacturer's protocol. |
| Instrument | Flow cytometry was performed using a Beckman Coulter Gallios flow cytometer or CytoFLEX S, both Beckman Coulter. |
| Software | Data were analyzed using Kaluza (version 1.2) or Flowjo software (version 10.7 and newer) |
| Cell population abundance | Neutrophils were definded as CD15 positive (human) and Ly6G positive (murine) population. Purity after isolation was approx. 75-80% (murine samples) and approx. 90-95% (human samples). |
| Gating strategy | Neutrophils were definded as CD15 positive (human samples) and Ly6G positive (murine) population. |

☒ Tick this box to confirm that a figure exemplifying the gating strategy is provided in the Supplementary Information.

