## [Peer Review File · Nature Immunology]

Peer Review Information

Journal: Nature Immunology

Manuscript Title: E-selectin-mediated rapid NLRP3 inflammasome activation regulates S100A8/A9 release from neutrophils via transient gasdermin D pore formation

Corresponding author name(s): Professor Markus Sperandio

Reviewer Comments & Decisions:

Decision Letter, initial version:
--

21st May 2021

Dear Professor Sperandio,

Your Article, "Transient gasdermin D pores control S100A8/A9 release from rolling neutrophils" has now been seen by 3 referees. You will see from their comments copied below that while they find your work of considerable potential interest, they have raised quite substantial concerns that must be addressed.

In light of these comments, we cannot accept the manuscript for publication, but would be very interested in considering a revised version that addresses these serious concerns, mostly in line with the author response you already emailed me, as detailed below:

We are happy for you to generally follow the plan outlined in your response to the reviewers. In response to reviewer 1 we agree with your plan, but note that looking at triggering of the dTGN is optional. The critical mechanistic gap to work on is the link between E-selectin and K⁺ efflux as noted by reviewer 2 and 3.

As noted previously, reviewer 3 is our only neutrophil adhesion/rolling expert, so we take their concerns seriously. Where you have offered to perform further experiments in response to this reviewer we are satisfied, but for a few requests you decline and instead reference your previous work in Nature Communications. This response is sufficient for us to return to further review, but it is unclear to us if this will satisfy the reviewer, as internal controls in experiments are usually preferable. So if you choose not to provide these experiments here, please be aware that this might lead to a further round of revision. For example, with regards to the use of E-selectin KO mice where you say you previously did experiments with E-selectin antibody, please note that the reviewer might consider that an antibody does not fully prevent E-selectin function like a KO would.

We do agree that use of PSGL-1 KO mice would be useful to support claims about shedding as part of

the mechanism.

I am not really sure you have fully answered ref3 point 7, which was referring to other inflammasome activators, not just a chemokine – note you are offering to look at nigericin for reviewer 2, for example.

Finally, as noted in response to reviewer 3 section E, your conclusions are reliant on the idea that the pores are short lived, so although you are planning on not doing much in the way of showing ESCRT-mediated repair (as noted to reviewer 2), I think this would be supportive if you had some data in this area. Reviewer 2 stated fairly clearly that looking at NINJ1 will not explain the rapid repair. In any case, please ensure that it is very clear at least that the pores are indeed short lived.

If you choose to revise your manuscript taking into account all reviewer and editor comments, please highlight all changes in the manuscript text file in Microsoft Word format.

- * Include a "Response to referees" document detailing, point-by-point, how you addressed each referee comment. If no action was taken to address a point, you must provide a compelling argument. This response will be sent back to the referees along with the revised manuscript.
- * If you have not done so already please begin to revise your manuscript so that it conforms to our Article format instructions at <http://www.nature.com/ni/authors/index.html>. Refer also to any guidelines provided in this letter.
- * Include a revised version of any required reporting checklist. It will be available to referees (and, potentially, statisticians) to aid in their evaluation if the manuscript goes back for peer review. A revised checklist is essential for re-review of the paper.

The Reporting Summary can be found here:

When submitting the revised version of your manuscript, please pay close attention to our [href="https://www.nature.com/nature-research/editorial-policies/image-integrity">Digital Image Integrity Guidelines.](https://www.nature.com/nature-research/editorial-policies/image-integrity) and to the following points below:

Finally, please ensure that you retain unprocessed data and metadata files after publication, ideally archiving data in perpetuity, as these may be requested during the peer review and production

process or after publication if any issues arise.

[REDACTED]

If you wish to submit a suitably revised manuscript we would hope to receive it within 6 months. If you cannot send it within this time, please let us know. We will be happy to consider your revision so long as nothing similar has been accepted for publication at Nature Immunology or published elsewhere.

Nature Immunology is committed to improving transparency in authorship. As part of our efforts in this direction, we are now requesting that all authors identified as 'corresponding author' on published papers create and link their Open Researcher and Contributor Identifier (ORCID) with their account on the Manuscript Tracking System (MTS), prior to acceptance. ORCID helps the scientific community achieve unambiguous attribution of all scholarly contributions. You can create and link your ORCID from the home page of the MTS by clicking on 'Modify my Springer Nature account'. For more information please visit www.springernature.com/orcid.

Thank you for the opportunity to review your work.

Sincerely,

Nicholas Bernard
Consulting Editor
Nature Immunology

Referee expertise:

Referee #1:

Referee #2:

Referee #3:

Reviewers' Comments:

Reviewer #1:

Remarks to the Author:

Pruenster and colleagues document a mechanism behind transient GSDMD pores releasing specific

DAMPs from neutrophils. The mechanistic insight is the critical data, with evidence for E-selectin triggering the process, leading to slow neutrophil rolling and adhesion. L-selectin shedding prevents terminal neutrophil pyroptosis, which is presumably an important thresholding process. This is elegant biology to pair with the phenomena of pyroptosis resistant neutrophils, and is certain to aid in our understanding of these inflammatory processes during infection and sterile autoinflammatory diseases.

Major questions

Does E-selectin trigger dTGN formation to nucleate NLRP3? Presumably yes as it is K⁺ efflux dependent? The discussion suggests this as an "alternative" process, but there is no evidence to support that, or separate it from other potential upstream effects on mitochondria or lysosomes which are also implicated in NLRP3 activation.

Figure 4 demonstrates TAPI-0 inhibition prevents L-selectin release, so this should promote E-selectin mediated inflammasome activation (casp-1/GSDMD cleavage), could that be quantified? Is it feasible to perform deletion or inhibition of L-selectin to confirm that indeed the NLRP3 activation by E-selectin is specific to that receptor? What about other L-selectin triggers, do they also activate the NLRP3 inflammasome?

Given GSDMD activation but no terminal LDH release after E-selectin treatment, NINJ1 and ESCRT activity would be interesting to study, but I can see that this is beyond the scope of the current manuscript.

Minor points

Western blots are too narrow throughout, can't judge the quality easily.

Reviewer #2:

Remarks to the Author:

In this exciting article from the Sperandio lab, the authors demonstrate the novel findings that e-selectin-engaged (rolling) neutrophils trigger the NLRP3-CASP1-GSDMD signalling axis, to secrete S100 alarmins for firm adhesion to the vascular endothelium, allowing for neutrophil extravasation to tissue. The manuscript narrative is clear and logical, and manuscript experiments are generally appropriately designed, quantified, analysed, and discussed. There are several novel aspects of this study that will be of great interest to the immunology community, and several clever experiments (e.g. Fig 4b addition of PI at different time points after e-selectin addition – very nice!). There are however some 'missing mechanistic links' – further insight into these aspects of the biology would make this a strong study for publication in a high calibre journal such as Nature Immunology.

Major points:

1. According to the author's model, e-selectin engagement by neutrophils induces NLRP3 inflammasome assembly via K⁺ flux. By what mechanism does E-selectin trigger neutrophils K⁺ efflux?
2. Given that NLRP3 inflammasome assembly has not been reported to occur downstream of e-selectin

before, this aspect of the study would be strengthened by the addition of microscopy studies visualising the ASC speck (and perhaps active caspase-1 using FLICA too) in neutrophils exposed to +/- e-selectin (and +/- high extracellular K media).

3. The authors speculate, but do not show, that NLRP3-induced GSDMD pores may be repaired by the ESCRT machinery. This aspect of the study would be strengthened by data clearly showing ESCRT-mediated membrane repair (no doubt the Broz lab have the appropriate reagents to show this). Also, how does L-selectin shedding trigger membrane repair?

4. In the discussion, the authors suggest that e-selectin-stimulated neutrophils may maintain viability through membrane repair or low expression of NINJ1; however, low NINJ1 expression would not explain the rapid repair of GSDMD pores in Fig 4b. This should be acknowledged in the discussion.

5. In this study's experimental set up, E-selectin-induced inflammasome signalling (e.g. caspase-1 cleavage, PI uptake) appears occur extremely rapidly (e.g. within 10 mins). The speed is astonishing (and much quicker than that of macrophages stimulated with a strong inflammasome agonist such as nigericin). It would be interesting to unpack this a little further with some time course studies (perhaps these could be integrated into point 2 suggested microscopy studies) to compare neutrophil responses to e-selectin versus nigericin.

6. In this study's experimental set up, E-selectin-induced inflammasome signalling (e.g. caspase-1 cleavage, PI uptake) appears to occur without an inflammasome priming step. Can the authors explain this? How 'clean' is their E-selectin – is it contaminated with bacterial PAMPs that may provide a priming signal? Does the addition of E-selectin to neutrophils trigger NFkB activation in addition to K+ efflux? Do e-selectin-treated neutrophils release IL-1b in addition to S100s? How does e-selectin compare to nigericin as a stimulus in these studies? Does nigericin require a priming stimulus for NLRP3-induced GSDMD pores in the authors hands? If so, why is the priming step dispensable for e-selectin as a stimulus?

7. Given the speed with which neutrophils respond to e-selectin, one might hypothesise that NLRP3 activation in this setting is canonical (and not via the caspase-4/5/11 non-canonical pathway). Can the authors formally show this? E.g. they could incorporate WT vs Casp1/11 neutrophils (used elsewhere in the manuscript) into suggested microscopy studies (point 2) to show that ASC specks are blocked by MCC950 but not caspase-1/11 knockout.

8. Can the authors please clarify their source of murine neutrophils? They state that the neutrophils are "bone marrow derived" (which implies differentiation from progenitor cells, as for bone marrow-derived macrophages) but methods seem to indicate that neutrophils were purified directly from bone marrow without differentiation. If the latter is the case, suggest changing all instances of "bone marrow derived neutrophils" to simply "bone marrow neutrophils" as is standard in the literature.

9. Page 8 first paragraph and Supp Fig 3: the presumption here is that LPS+Nigericin-stimulated cells are undergoing pyroptosis and not another form of cell death. Nigericin is extremely toxic to cells, and can kill inflammasome-deficient cells through non-pyroptotic pathways. For the authors to indicate that LPS+nigericin offers a "positive control" for pyroptosis, as they have done, they should incorporate MCC950 or (better still) VX765 (caspase-1 inhibitor) into their experimental design. If they can show that LPS+Nigericin-induced death is blocked by caspase-1 inhibition they can indeed define this cell death mode as pyroptosis.

10. Some experiments (e.g. Fig 4) were performed 'in the presence of a PSGL-1 antibody' but it was unclear whether the antibody was added to all samples, of just the inhibitor-treated samples. Please clarify and justify this approach. If the PSGL-1 antibody was only added to the inhibitor-treated samples, then these samples should be shown +/- PSGL1 antibody to allow proper data interpretation.

11. Did MCC950 block S100 release in e-selectin-stimulated human neutrophils, similar to murine neutrophils? It would be helpful to show this in fig 2.

Minor suggestions to improve clarity and accuracy:

1. Line 20 page 6 "... indicating that blockade of NLRP3-dependent S100A8/A9 release leads to loss of S100A8/A9-TLR4-dependent beta2 integrin activation". Suggest rewording to ensure it is clear to your reader that this is speculation – S100A8/A9 signalling via TLR4 to integrins was not assessed in this study.
2. Page 7 PI uptake experiments: suggest adding a sentence or two at the outset of this paragraph to explain that PI can enter GSDMD pores while LDH is too large to exit through these pores, prior to recounting experiments using PI and LDH to measure GSDMD pores versus membrane rupture, respectively. Readers outside of the inflammasome or cell death fields are likely to be unaware that PI can be taken up by living cells with reversible membrane pores.
3. Line 6 page 13 "serine" is misspelt "serin"
4. Figure 3 schema at the top of the page: LDH release assay is misspelt "LHD"

Reviewer #3:

Remarks to the Author:

A. Summary of the key results:

Pruenster et al. describe an intriguing finding by showing data supportive of the hypothesis that binding of neutrophils to E-selectin under non-static conditions leads to activation of the inflammasome that in turn processes GSD into GSDMD pores that allows the release of calprotectin/S100A8/A9 from the cytosol of neutrophils into the extracellular space. This hypothesis can help understand the mechanism of release of the cytosolic protein S100A8/A9 that has been obscure for years.

B. Originality and significance: if not novel, please include reference:

The article is definitely original and logically follows on earlier work published by this group. Again the identification of a mechanism of S100A8/A9 release is also very important. Such mechanism if true is of great significance as modulation of this process seems to be a druggable target in inflammatory diseases or conditions. There are quite some arguments where the wish seems to be the father of the thought (see below). As it stands the data are not sufficiently supportive of the hypothesis.

C. Data & methodology: validity of approach, quality of data, quality of presentation:

The following needs to be carefully addressed (random order).

Major:

1. The authors refer to earlier work that E-selectin-Fc bound to a plate is not activating Fc-receptors of neutrophils. Please show that E-selectin induced release of S100A8/A9 by both mouse and human neutrophils is not affected by blocking CD32 and CD16 antibodies. Along similar lines, please show that the hypothesis is specific for E-selectin by showing that P-selectin-Fc is not active in the release of S100A8/A9.
2. The pores will presumably lead to a maximal increase in $[Ca^{2+}]_i$ that will lead to fusion of at least secretory vesicles and part of the specific granules. It is very difficult to understand that expression of activation markers originating from these granules is similar (at least CD11b) in wt and GSDMD^{-/-} mice. The absence of activation of neutrophils during expression of active GSDMD channels needs to be

explained or data showing this activation shown.

3. The article relies on pharmacological inhibitors where obvious experiments should have been done with relevant KO-mice. Three mice come into mind:

- E-selectin KO mice: these mice have a very limited inflammatory phenotype. Please show that TNF-induced activation in vivo does not lead to release of S100A8/A9 in these mice. Please explain what the consequence of the lack of clear phenotype means for the importance of the current hypothesis.

- MYD88 KO mice: redo the experiments relying on TAC242 and paquinimod.

- L-selectin KO mice or ADAM17 KO mice: please show that TNF-induced activation in vivo leads to modulated release of S100A8/A9.

4. The PI-uptake is of concern. It is presented as support for the hypothesis. It might also be interpreted as dying cells. The rescue is certainly an argument in favor of the hypothesis. So please show in a rolling assay under the fluorescent microscope that rolling neutrophils on TNF-activated HUVEC show transient PI-positive cells.

5. It is well known that L-selectin is shed upon activation in vitro, which is accompanied by upregulation of CD11b. In vivo, this is much less clear. Please provide evidence that L-selectin is shed in vivo in the mouse activated by TNF.

6. Neutrophil priming: page 11 line 4 implies that neutrophils are 'ready to go' which is quite a statement as it ignores a large literature on priming of neutrophils by an array of inflammatory mediators that prime neutrophils for all sorts of activation.

7. It is completely unclear how E-selectin induces signaling leading to activation of the inflammasome. Can the authors show that alternative activators of the inflammasome will also lead to release of S100A8/A9? What about chemokines?

Minor:

8. Are neutrophils first line defense? What about tissue resident macrophages/

9. Please indicate that neutrophils under rotating conditions in vitro binding to rec-Eselectin-Fc is not the same as rolling on endothelial cells in vivo.

10. It is important to know whether the differentiation and numbers and of leukocytes in the blood of GSDMD^{-/-} mice and Casp1/11^{-/-} mice is similar to the wt.

11. Page 10 line 8: it is implied that murine neutrophils were used originating from the peripheral blood. Is this correct? If so how were these cells isolated?

D. Appropriate use of statistics and treatment of uncertainties:

- Article is a bit basic on the statistics. E.G. Figure 1C and 1D are evaluated paired, which should be unpaired. The data are of different mice.

E. Conclusions: robustness, validity, reliability:

1. Conceptual problem: The central dogma around the adhesion paradigm is the glue-like interaction of selectin/selectin ligands during rolling adhesion (no inside-out likely little outside-in signals) by which cells can keep rolling in the absence of interaction with chemokines that are presented by activated endothelial cells. The data of the current study imply very marked activation of neutrophils by E-selectin irrespective of the presence of chemokines as the pores will lead to a maximum increase in Ca^{2+} as well as complete depolarization as the gradient Na^{+}/K^{+} will be completely gone. This situation seems hard to control and not very plausible. This needs very careful discussion which is completely absent now.

F. Suggested improvements: experiments, data for possible revision:
See above.

G. References: appropriate credit to previous work?:
Yes

H. Clarity and context: lucidity of abstract/summary, appropriateness of abstract, introduction and conclusions:
The hypothesis is clear, but the supportive evidence not sufficient (see above).

Author Rebuttal to Initial comments

See inserted PDF

Reviewer #1

(Remarks to the Author)

Pruenster and colleagues document a mechanism behind transient GSDMD pores releasing specific DAMPs from neutrophils. The mechanistic insight is the critical data, with evidence for E-selectin triggering the process, leading to slow neutrophil rolling and adhesion. L-selectin shedding prevents terminal neutrophil pyroptosis, which is presumably an important thresholding process. This is elegant biology to pair with the phenomena of pyroptosis resistant neutrophils, and is certain to aid in our understanding of these inflammatory processes during infection and sterile autoinflammatory diseases.

We thank reviewer 1 for their positive and constructive comments which we have addressed in the point-to-point response below.

Major questions

Does E-selectin trigger dTGN formation to nucleate NLRP3? Presumably yes as it is K⁺ efflux dependent? The discussion suggests this as an “alternative” process, but there is no evidence to support that, or separate it from other potential upstream effects on mitochondria or lysosomes which are also implicated in NLRP3 activation.

We thank the reviewer for his/her insightful comments. Indeed, rapid E-selectin-induced inflammasome activation leading to GSDMD pore formation and S100A8/A9 release is dependent on K⁺ efflux. We addressed the role of K⁺-efflux in rapid inflammasome activation in more detail in the revised version of the paper and could identify the voltage-gated K⁺ channel K_v1.3 to be essential for E-selectin triggered NLRP3 inflammasome activation and S100A8/A9 release using the specific small molecule K_v1.3 inhibitor PAP-1 as well as K_v1.3 (*Kcna3*^{-/-}) deficient mice (new Fig. 3 a, b and c).

In 2018, Chen and Chen showed substantial trans-Golgi network (TGN) disassembly induced by different NLRP3 stimuli (nigericin and ATP, after priming the cells with LPS). Dispersed TGN (dTGN) in turn supported NLRP3 aggregation and polymerization of the adaptor protein ASC allowing recruitment of caspase-1 to activate the downstream signaling cascade ¹.

To address a potential role for E-selectin in triggering TGN disassembly in neutrophils during rapid inflammasome activation we isolated human neutrophils and stimulated the cells with E-selectin or PBS (control) for 10min. After fixation and permeabilization, we stained the cells with a mouse anti-human TGN46 antibody (clone 2F7, human homolog of TGN38) followed by a secondary fluorescent labeled anti-mouse antibody. In unstimulated human neutrophils we did not find a clustered TGN located near the nucleus, but a rather diffused TGN network near the nucleus and in the cytosol (Reviewer1, Fig. 1). This did not substantially change after E-selectin treatment suggesting that E-selectin dependent NLRP3 inflammasome activation does not lead to significant TGN disassembly. This is in contrast to Chen et al. who used HeLa cells stably expressing NLRP3–GFP as well as bone marrow derived macrophages. They found a single perinuclear cluster of TGN under basal conditions, which was dispersed only after stimulation with different NLRP3 stimuli ¹.

Reviewer Figure 1.1

Reviewer Figure 1.1 Transgolgi network in primary human neutrophils. Isolated human neutrophils were stimulated with PBS (control) or E-selectin ($1\mu\text{g ml}^{-1}$), fixed, permeabilized and stained with mouse-anti-TGN46 antibody (clone 3TF1-3G3, Thermofisher) and DAPI (blue channel). Cells were analyzed using a Leica SP8X WLL confocal microscope. Representative micrographs are shown from experiments from $n=3$ healthy volunteers, scale bar= $5\mu\text{m}$).

We agree with the reviewer that we should not claim our finding of a rapid E-selectin induced inflammasome activation as an “alternative” process independent of lysosomal disruption, metabolic changes, trans-Golgi disassembly and/or mitochondrial dysfunction, as we did not formally prove this. The term was more related to the speed of inflammasome activation we describe in our study. However, it was misleading in this context and therefore we removed “alternative” in the revised version of the manuscript.

Figure 4 demonstrates TAPI-0 inhibition prevents L-selectin release, so this should promote E-selectin mediated inflammasome activation (casp-1/GSDMD cleavage), could that be quantified? Is it feasible to perform deletion or inhibition of L-selectin to confirm that indeed the NLRP3 activation by E-selectin is specific to that receptor? What about other L-selectin triggers, do they also activate the NLRP3 inflammasome?

Unfortunately, in the course of the revision we realized through additional experiments that shedding of L-selectin does not contribute to the transient nature of GSDMD-dependent pore formation. Therefore, we had to take out this part of the study. We apologize for this:

In the original manuscript we studied L-selectin shedding by flow cytometry using the well-established anti human L-selectin antibody clone DREG56 (old Figure 4c). However, from additional experiments (negative controls) we performed during the revision, we learned that, although cells were already fixed, L-selectin-DREG56 binding and the subsequent signal measured by flow cytometry was extremely sensitive to temperature fluctuations. We initially described that stimulation of E-selectin induced L-selectin shedding after 5min of incubation. However, we now know that this drop in L-selectin signal intensity was not due to E-selectin interacting with neutrophils, but rather due to temperature changes after neutrophils had been fixed and stained. Therefore, we had to take out those experiments.

To fill this gap and provide additional evidence for the transient nature of GSDMD pore formation and S100A8/A9 release, we focused on membrane repair mechanisms in neutrophils and uncovered the

induction of ESCRT-III dependent membrane repair during stimulation of neutrophils by E-selectin. Please also refer to Q3 of this reviewer for more information. Concerning the specificity of E-selectin - L-selectin interactions on NLRP3 inflammasome activation and S100A8/A9 release, we performed *in vitro* release assays in human neutrophils using two additional L-selectin ligands, MadCAM-1 and Endoglycan (Podocalyxin-like 2 protein). Neither MadCAM-1 nor Endoglycan were able to induce S100A8/A9 release after 10min of incubation time (new Fig. 3f), underlining the specificity of E-selectin in the induction of rapid NLRP3 inflammasome activation and GSDMD pore formation. Of note, binding requirements for human L-selectin to human E-selectin (where L-selectin acts as E-selectin ligand and needs to be properly glycosylated ²) differ from those reported for MAdCAM-1/Endoglycan (here, MAdCAM-1 and Endoglycan are L-selectin ligands with characteristic glycosylation requirements for binding to L-selectin ³).

These data are now included in the revised manuscript.

Finally, we also intended to disrupt E-selectin/L-selectin interactions using Rivipansel (Glycomimetics), a small molecule inhibitor blocking E-selectin / L-selectin interactions ⁴. However, the company withdrew the drug from the market and refused to send it to us.

Given GSDMD activation but no terminal LDH release after E-selectin treatment, NINJ1 and ESCRT activity would be interesting to study, but I can see that this is beyond the scope of the current manuscript.

In order to elucidate ESCRT activity after E-selectin treatment and expand on the mechanism of transient GSDMD pore formation in neutrophils, we stimulated neutrophils with E-selectin for 15min. Subsequently, subcellular localization of CHMP4B, an ESCRT-III associated protein, was analyzed using confocal microscopy. ESCRT-III proteins are known to form a punctate pattern during membrane repair and to translocate to the plasma membrane ^{5,6}. Consistently, stimulation of neutrophils with E-selectin resulted in the formation of CHMP4B puncta, presumably corresponding to functional assemblies of ESCRT-III, while unstimulated cells (control) displayed a rather weak and diffuse cytoplasmic staining of CHMP4B (new Fig. 5c and d).

These data are now included in the manuscript.

Minor points

Western blots are too narrow throughout, can't judge the quality easily.

We provide now uncropped blots for all presented WB analyses.

Reviewer #2

(Remarks to the Author)

In this exciting article from the Sperandio lab, the authors demonstrate the novel findings that e-selectin-engaged (rolling) neutrophils trigger the NLRP3-CASP1-GSDMD signalling axis, to secrete S100 alarmins for firm adhesion to the vascular endothelium, allowing for neutrophil extravasation to tissue. The manuscript narrative is clear and logical, and manuscript experiments are generally appropriately designed, quantified, analyzed, and discussed. There are several novel aspects of this study that will be of great interest to the immunology community, and several clever experiments (e.g. Fig 4b addition of PI at different time points after e-selectin addition – very nice!). There are however some ‘missing mechanistic links’ – further insight into these aspects of the biology would make this a strong study for publication in a high calibre journal such as Nature Immunology.

We thank the reviewer for his/her encouraging and constructive comments, which we addressed in our point-to-point response below.

Major points:

1. According to the author’s model, e-selectin engagement by neutrophils induces NLRP3 inflammasome assembly via K⁺ flux. By what mechanism does E-selectin trigger neutrophils K⁺ efflux?

We thank the reviewer for this important question.

Recently, our group was able to demonstrate that the voltage-gated potassium channel K_v1.3 is expressed on neutrophils and plays a critical role during neutrophil recruitment under inflammatory conditions ⁷. Based on these findings, we speculated that K_v1.3 might trigger K⁺-efflux and mediate rapid E-selectin triggered NLRP3 inflammasome activation and ensuing GSDMD pore formation.

Indeed, we found that blockade of K_v1.3 with the selective inhibitor 5-(4-Phenoxybutoxy)psoralen (PAP-1) reduced E-selectin induced caspase-1 cleavage (new Fig.3a). Furthermore, pharmacological inhibition of K_v1.3 (with PAP-1) or genetic deletion of K_v1.3 (isolated neutrophils from *Kcna3*^{-/-} mice) dampened E-selectin induced S100A8/A9 release compared to control or WT cells, respectively (new Fig.3b and c). Importantly, overall intracellular S100A8/A9 levels did not differ between WT and *Kcna3*^{-/-} neutrophils (new Supplementary Fig. 3). Together, our data demonstrate that potassium efflux via K_v1.3 is necessary for rapid NLRP3 inflammasome activation and S100A8/A9 release.

Interestingly, neither ATP, nor nigericin were able to induce S100A8/A9 release within 10min (new Fig. 3d and e), suggesting that E-selectin stimulation does not only mediate K⁺-efflux by K_v1.3, but also induces an additional downstream signal that is required for NLRP3 activation (e.g. posttranslational NLRP3 modification). Of note, concomitant stimulation with LPS and nigericin was also not able to induce S100A8/A9 release within 10 min (new Fig. 3e).

Furthermore, rapid inflammasome activation with E-selectin for 10min did not result in IL-1 β release *in vitro* (new Fig. 3g), presumably due to low amounts of pre-stored pro-IL-1 β in unprimed neutrophils. However, priming of neutrophils with LPS (for 2.5h) and subsequent stimulation of the cells with E-selectin (30min) induced IL-1 β release similar to classical inflammasome stimuli (LPS priming for 2.5h, nigericin activation for 30min), underlining a role of E-selectin as an endogenous inflammasome activator in unprimed and primed cells (new Fig. 3h).

These data are now included in the revised manuscript.

2. Given that NLRP3 inflammasome assembly has not been reported to occur downstream of e-selectin before, this aspect of the study would be strengthened by the addition of microscopy studies visualizing the ASC speck (and perhaps active caspase-1 using FLICA too) in neutrophils exposed to +/- e-selectin (and +/- high extracellular K media).

To address E-selectin mediated inflammasome activation and concomitant caspase-1 cleavage in more detail, we made use of the FAM-FLICA caspase-1 assay kit as suggested by this reviewer. We isolated human neutrophils from healthy blood donors, loaded the cells with FLICA dye and stimulated them with E-selectin or PBS (control) for 10min. Active caspase-1 was visualized using confocal microscopy. Indeed, we found speck-like aggregates of active caspase-1 in neutrophils within 10min of E-selectin treatment compared to PBS control (new Fig. 2d and 2e). In addition, E-selectin induced an increase in the overall intensity of the FLICA signal compared to PBS control (new Fig. 2f) underlining the capacity of E-selectin to activate the inflammasome machinery.

In addition and as suggested by this reviewer, we also studied ASC Speck formation induced by E-selectin. Indeed, we found ASC oligomers in neutrophils stimulated with E-selectin for 10min (new Supplementary Fig. 2a)

These data are now included in the revised manuscript.

3. The authors speculate, but do not show, that NLRP3-induced GSDMD pores may be repaired by the ESCRT machinery. This aspect of the study would be strengthened by data clearly showing ESCRT-mediated membrane repair (no doubt the Broz lab have the appropriate reagents to show this). Also, how does L-selectin shedding trigger membrane repair?

We thank the author for this important comment. As also discussed in the point-to-point response to reviewer 1, we now elucidated ESCRT activity after E-selectin treatment and stimulated neutrophils with E-selectin for 15min. Subcellular localization of CHMP4B, an ESCRT-III associated protein, was analyzed using confocal microscopy. ESCRT-III proteins are known to form a punctate pattern during membrane repair and to translocate to the plasma membrane^{5, 6}. Consistently, stimulation of neutrophils with E-selectin resulted in the formation of CHMP4B puncta, presumably corresponding to functional assemblies of ESCRT-III, while unstimulated cells (control) displayed a rather weak and diffuse cytoplasmic staining of CHMP4B (new Fig. 5c and d).

These data are now included in the revised manuscript.

Unfortunately, in the course of the revision we learned that shedding of L-selectin was not the regulating factor for transient GSDMD pore formation. We apologize for this:

In the original manuscript we studied L-selectin shedding by flow cytometry using the well-established anti human L-selectin antibody clone DREG56 (old Figure 4c). However, from additional experiments (negative controls) we performed during the revision, we learned that, although cells were already fixed, L-selectin-DREG56 binding and the subsequent signal measured by flow cytometry was extremely sensitive to temperature fluctuations. We initially described that stimulation of E-selectin induced L-selectin shedding after 5min of incubation. However, we now know that this drop in L-selectin signal intensity was not due to E-selectin interacting with neutrophils, but rather due to temperature changes after neutrophils had been fixed and stained. Therefore, we had to take out those experiments.

To fill this gap and provide additional evidence for the transient nature of GSDMD pore formation and S100A8/A9 release, we focused on membrane repair mechanisms in neutrophils and uncovered the induction of ESCRT-III dependent membrane repair during stimulation of neutrophils by E-selectin, as discussed above.

4. In the discussion, the authors suggest that e-selectin-stimulated neutrophils may maintain viability through membrane repair or low expression of NINJ1; however, low NINJ1 expression would not explain the rapid repair of GSDMD pores in Fig 4b. This should be acknowledged in the discussion.

As suggested by this reviewer we have removed this statement from the manuscript.

5. In this study's experimental set up, E-selectin-induced inflammasome signalling (e.g. caspase-1 cleavage, PI uptake) appears occur extremely rapidly (e.g. within 10 mins). The speed is astonishing (and much quicker than that of macrophages stimulated with a strong inflammasome agonist such as nigericin). It would be interesting to unpack this a little further with some time course studies (perhaps these could be integrated into point 2 suggested microscopy studies) to compare neutrophil responses to e-selectin versus nigericin.

We thank the reviewer for this valuable suggestion. Indeed, the speed of E-selectin mediated inflammasome activation and S100A8/A9 release is extremely rapid. In order to investigate whether other inflammasome activators were able to induce rapid S100A8/A9 release and to uncover the time course of the release with different stimuli, we performed the following *in vitro* release assays.

In a first set of experiments, we used isolated mouse neutrophils and stimulated the cells for 10min with PBS (negative control), E-selectin (positive control), nigericin, ATP, nigericin/LPS and CXCL1 (a chemokine binding to CXCR2). Interestingly, none of the well-known inflammasome activators ATP, nigericin and the combination of LPS/nigericin was able to induce S100A8/A9 release within 10min (new Fig. 3d and e). The same was true for CXCR2-binding chemokine CXCL1 (Reviewer Fig.3.3). These findings suggest that rapid E-selectin triggered NLRP3 inflammasome activation (10min) is neither induced by nigericin nor ATP nor by chemokine induced neutrophil activation (GPCR coupled signaling). In line with these findings, Morikis et al. 2017 demonstrated that CXCL8 (the human counterpart of CXCL1) was not able to induce S100A8/A9 release in human neutrophils ⁴. In addition, these findings suggest that E-selectin stimulation does not only mediate K⁺-efflux by K_v1.3, but in addition might induce downstream signaling leading to posttranslational NLRP3 modifications, a mechanism required for proper NLRP3 activation ^{8, 9, 10}.

In a second set of experiments, we stimulated isolated mouse neutrophils for 30min with PBS (negative control), E-selectin (positive control), nigericin and ATP. As expected and in line with previous findings ¹¹, incubation of neutrophils with E-selectin for 30min did not further increase the amount of released S100A8/A9 compared to 10min incubation time, underlining transient and limited pore formation in neutrophils activated with E-selectin. Interestingly, nigericin induced S100A8/A9 release after 30min, while ATP did not (Reviewer Fig. 2.2a, preliminary data). NLRP3 inflammasome activation with nigericin (stimulation for 45min) without preceding priming was described before in human monocytes ¹². Our observation that nigericin is able to induce S100A8/A9 release after 30min while ATP is not, is currently unclear, but cellular toxicity of nigericin might potentially contribute to this finding. This will be investigated in future studies.

Priming of neutrophils with LPS for 2.5h followed by stimulation with E-selectin or nigericin for 30min induced S100A8/A9 release under both conditions (Reviewer Fig. 2.2b, preliminary data), suggesting that both nigericin as well as E-selectin provide the appropriate K⁺-efflux to induce NLRP3 inflammasome activation in this setting.

Reviewer Figure 2.2

Reviewer Figure 2.2 E-selectin and nigericin induce S100 release after 30min of stimulation in unprimed and LPS primed neutrophils. (a) Unprimed and (b) LPS primed ($1\mu\text{g ml}^{-1}$ for 2.5h) isolated murine bone marrow neutrophils were stimulated for 30min with either E-selectin ($1\mu\text{g ml}^{-1}$), nigericin ($10\mu\text{M}$) or ATP (3mM ; unprimed cells only), respectively. Supernatants were collected and S100A8/A9 levels were analyzed. For (a) $n=3$, preliminary results, for (b) $n\geq 6$ mice per group, one-way ANOVA, Tukey's multiple comparison. Data is presented as mean \pm SEM

6. In this study's experimental set up, E-selectin-induced inflammasome signaling (e.g. caspase-1 cleavage, PI uptake) appears to occur without an inflammasome priming step. Can the authors explain this? How 'clean' is their E-selectin – is it contaminated with bacterial PAMPs that may provide a priming signal? Does the addition of E-selectin to neutrophils trigger NFkB activation in addition to K^+ efflux? Do e-selectin-treated neutrophils release IL-1b in addition to S100s? How does e-selectin compare to nigericin as a stimulus in these studies? Does nigericin require a priming stimulus for NLRP3-induced GSDMD pores in the authors hands? If so, why is the priming step dispensable for e-selectin as a stimulus?

We thank the reviewer for these important comments. We agree that it is fundamental to exclude any potential contaminations of the used reagents. All recombinant proteins used in this study were obtained from R&D Systems and dissolved under sterile conditions. To address whether PAMP contaminations could be responsible for rapid NLRP3 inflammasome activation, we tested the ability of lipopolysaccharide (LPS) to induce rapid S100A8/A9 release. For this approach we isolated murine bone marrow neutrophils and stimulated the cells with a combination of LPS/nigericin ($1\mu\text{g/ml}$ and $7.5\mu\text{g/ml}$, respectively) for 10min (see also answer to Q1 and Q5). LPS/nigericin was unable to induce S100A8/A9 release within 10min, demonstrating that LPS (contaminations) are not the priming signal in our experimental setting of rapid E-selectin induced inflammasome activation (new Fig. 3e).

In addition, we investigated NFkB signaling by analyzing Ikb phosphorylation in isolated human neutrophils after stimulating neutrophils for 10min with E-selectin or PBS (control) in the presence of Paquinimod and TAC242 (to avoid TLR4 signaling by released S100A8/A9) by Western Blot analyses. TNF- α stimulation (activating the NFkB signaling pathway via TNF receptor) for 30min was used as a positive control. 10min of E-selectin stimulation did not result in remarkable Ikb phosphorylation (Reviewer Figure 2.3), demonstrating that E-selectin does not lead to activation of the NFkB signaling pathway within 10min.

Reviewer Figure 2.3

Reviewer Figure 2.3 E-selectin does not induce IκB phosphorylation within 10min of stimulation. Isolated human neutrophils were stimulated with PBS (negative control), E-selectin ($1\mu\text{g ml}^{-1}$) in the presence of Paquinimod and the TLR4 inhibitor TAC242 or TNF- α (30min, positive control), respectively and activation of the NFκB signaling pathway was assessed by analysis of IκB phosphorylation using Western blotting ($n\geq 4-5$, one-way ANOVA, Tukey's multiple comparison). Data is presented as mean \pm SEM and representative Western blot.

In order to investigate IL-1 β release in E-selectin treated neutrophils, we stimulated isolated cells with PBS (negative control), E-selectin, nigericin or a combination of LPS/nigericin, respectively (please see also answer to Q1). None of the stimuli were able to induce IL-1 β release *in vitro* within 10min of stimulation (new Fig. 3g), presumably due to low amounts of pre-stored pro-IL-1 β in unprimed neutrophils. However, priming of neutrophils with LPS (for 2.5h) and subsequent stimulation with E-selectin (30min) induced IL-1 β release similar to canonical inflammasome activation (2.5h of LPS priming, 30min nigericin activation), although with reduced amounts of secreted IL-1 β , underlining the role of E-selectin as an endogenous inflammasome activator in unprimed and primed cells (new Fig. 3h).

These data are now included in the revised manuscript.

7. Given the speed with which neutrophils respond to e-selectin, one might hypothesize that NLRP3 activation in this setting is canonical (and not via the caspase-4/5/11 non-canonical pathway). Can the authors formally show this? E.g. they could incorporate WT vs Casp1/11 neutrophils (used elsewhere in the manuscript) into suggested microscopy studies (point 2) to show that ASC specks are blocked by MCC950 but not caspase-1/11 knockout.

We thank the reviewer for his/her comments. Indeed, we did not formally prove whether GSDMD dependent pore formation in our experimental setting (stimulation with E-selectin for 10min) is triggered by canonical or non-canonical inflammasome activation. Priming is described to be essential for non-canonical inflammasome activation in mice, probably due to the low expression of caspase-11 in resting cells (which might differ in human cells expressing caspase-4/5) ¹³. The rapid induction of GSDMD pores without priming in our experimental setting therefore points in the direction of canonical inflammasome activation. This is strongly supported by the finding that blocking NLRP3 by MCC950 (new Supplementary Fig. 2b) led to a similar reduction in S100A8/A9 release *in vivo* as observed in *Casp1^{-/-}Casp11^{-/-}* mice (Fig. 1a).

Can the authors please clarify their source of murine neutrophils? They state that the neutrophils are “bone marrow derived” (which implies differentiation from progenitor cells, as for bone marrow-derived macrophages) but methods seem to indicate that neutrophils were purified directly from bone marrow without differentiation. If the latter is the case, suggest changing all instances of “bone marrow derived neutrophils” to simply “bone marrow neutrophils” as is standard in the literature.

We thank the reviewer for this comment and agree that ‘bone marrow neutrophils’ is the appropriate term. Therefore, we renamed those cells as suggested by the reviewer throughout the manuscript.

9. Page 8 first paragraph and Supp Fig 3: the presumption here is that LPS+Nigericin-stimulated cells are undergoing pyroptosis and not another form of cell death. Nigericin is extremely toxic to cells, and can kill inflammasome-deficient cells through non-pyroptotic pathways. For the authors to indicate that LPS+nigericin offers a “positive control” for pyroptosis, as they have done, they should incorporate MCC950 or (better still) VX765 (caspase-1 inhibitor) into their experimental design. If they can show that LPS+Nigericin-induced death is blocked by caspase-1 inhibition they can indeed define this cell death mode as pyroptosis.

We thank the reviewer for this comment and agree that we did not formally show that LPS priming and nigericin stimulation induced pyroptosis in our experimental setting. Therefore, we removed the term in the manuscript in regard to this experimental setting.

10. Some experiments (e.g. Fig 4) were performed ‘in the presence of a PSGL-1 antibody’ but it was unclear whether the antibody was added to all samples, of just the inhibitor-treated samples. Please clarify and justify this approach. If the PSGL-1 antibody was only added to the inhibitor-treated samples, then these samples should be shown +/- PSGL1 antibody to allow proper data interpretation.

Unfortunately and as stated above, in the course of the revision we realized that shedding of L-selectin was not a regulating factor for transient GSDMD pore formation following stimulation with E-selectin. Therefore, we had to take out this part of the study (please see also answer to Q2, reviewer 1 and answer to Q3 for this reviewer).

11. Did MCC950 block S100 release in e-selectin-stimulated human neutrophils, similar to murine neutrophils? It would be helpful to show this in fig 2.

We apologize for not being clear here, but all experiments in Figure 2 have been performed using human neutrophils. In order to investigate E-selectin induced S100A8/A9 release in the presence of MCC950 using murine cells we performed *in vivo* and *in vitro* release assays using mouse neutrophils. We found that blockade of NLRP3 inflammasome activation with MCC950 prevented E-selectin induced S100A8/A9 release *in vivo* and *in vitro* (new Supplemental Fig. 2b and c). In addition, blockade of NLRP3 inflammasome activation with MCC950 affected leukocyte rolling velocities and significantly reduced number of adherent and extravasated cells in the cremaster mouse model of inflammation (new Supplementary Fig. 5).

These data are now included in the revised manuscript.

Minor suggestions to improve clarity and accuracy:

1. Line 20 page 6 “... indicating that blockade of NLRP3-dependent S100A8/A9 release leads to loss of S100A8/A9-TLR4-dependent beta2 integrin activation”. Suggest rewording to ensure it is clear to your reader that this is speculation – S100A8/A9 signalling via TLR4 to integrins was not assessed in this study.

We thank the reviewer for this comment. However, we think that this sentence is correct and not speculative. We have shown S100A8/A9-induced β 2 integrin activation via TLR4 in our previous work (Pruenster et al. 2015)¹¹ and in Fig. 1c and d (rolling velocities).

2. Page 7 PI uptake experiments: suggest adding a sentence or two at the outset of this paragraph to explain that PI can enter GSDMD pores while LDH is too large to exit through these pores, prior to recounting experiments using PI and LDH to measure GSDMD pores versus membrane rupture, respectively. Readers outside of the inflammasome or cell death fields are likely to be unaware that PI can be taken up by living cells with reversible membrane pores.

We added the text as requested by the reviewer (page 9, line 15 and page 10, line 6).

3. Line 6 page 13 “serine” is misspelt “serin”

Thank you, we corrected the typo.

4. Figure 3 schema at the top of the page: LDH release assay is misspelt “LHD”

Thank you, we corrected the typo in the Figure (now Figure 4).

Reviewer #3

(Remarks to the Author)

A. Summary of the key results:

Pruenster et al. describe an intriguing finding by showing data supportive of the hypothesis that binding of neutrophils to E-selectin under non-static conditions leads to activation of the inflammasome that in turn processes GSD into GSDMD pores that allows the release of calprotectin/S100A8/A9 from the cytosol of neutrophils into the extracellular space. This hypothesis can help understand the mechanism of release of the cytosolic protein S100A8/A9 that has been obscure for years.

B. Originality and significance: if not novel, please include reference:

The article is definitely original and logically follows on earlier work published by this group. Again the identification of a mechanism of S100A8/A9 release is also very important. Such mechanism if true is of great significance as modulation of this process seems to be a druggable target in inflammatory diseases or conditions. There are quite some arguments where the wish seems to be the father of the thought (see below). As it stands the data are not sufficiently supportive of the hypothesis.

C. Data & methodology: validity of approach, quality of data, quality of presentation:

The following needs to be carefully addressed (random order).

We thank the reviewer for his/her insightful and helpful comments, which we addressed in the point-to-point response below.

Major:

1. The authors refer to earlier work that E-selectin-Fc bound to a plate is not activating Fc-receptors of neutrophils. Please show that E-selectin induced release of S100A8/A9 by both mouse and human neutrophils is not affected by blocking CD32 and CD16 antibodies. Along similar lines, please show that the hypothesis is specific for E-selectin by showing that P-selectin-Fc is not active in the release of S100A8/A9.

We thank the reviewer for this valuable comment. In our earlier work on S100A8/A9 functions in neutrophils during neutrophil recruitment, we indeed showed that S100A8/A9 (MRP8/14) release is specific to E-selectin in murine neutrophils. P-selectin (CD62P) was unable to induce S100A8/A9 release (please refer to ¹¹, Fig. 1a). Fc control was unable to induce S100A8/A9 release either (please refer to ¹¹, Fig. 1b). However, as suggested by this reviewer, we also addressed S100A8/A9 release in the presence of CD16 and CD32 blocking antibodies (TruStain FcX™, BioLegend). Inhibition of Fc-receptors on murine neutrophils did not alter E-selectin-Fc chimera induced S100A8/A9 release, demonstrating no influence of the Fc-receptors on E-selectin induced S100A8/A9 release (Reviewer Fig. 3.1).

Reviewer Figure 3.1

Reviewer Figure 3.1 S100A8/A9 release is specific to E-selectin. Bone marrow neutrophils from WT mice were incubated with CD16 and CD32 blocking antibodies or control for 10min at room temperature and subsequently treated with E-selectin or PBS (control) for 10min. Supernatants were collected and S100A8/A9 levels were analyzed (n=3 mice per group). Data is presented as mean±SEM (two-way RM ANOVA, Sidak's multiple comparison), ns: not significant, *: p<0.05.

2. The pores will presumably lead to a maximal increase in [Ca²⁺]_i that will lead to fusion of at least secretory vesicles and part of the specific granules. It is very difficult to understand that expression of activation markers originating from these granules is similar (at least CD11b) in wt and GSDMD^{-/-} mice. The absence of activation of neutrophils during expression of active GSDMD channels needs to be explained or data showing this activation shown.

We apologize for not being clear regarding our experimental setup. FACS analysis of surface-expressed activation markers was performed in unstimulated WT and GSDMD deficient neutrophils to exclude any changes of rolling and adhesion relevant molecules under baseline conditions. No doubt, during inflammation, activation markers may certainly differ between WT and *Gsdmd*^{-/-} mice. This is also in line with the findings in Figure 6 showing that *Gsdmd*^{-/-} cells roll faster which is related to their reduced adhesion properties and reduced S100A8/A9 mediated β2 integrin activation.

3. The article relies on pharmacological inhibitors where obvious experiments should have been done with relevant KO-mice. Three mice come into mind:

- E-selectin KO mice: these mice have a very limited inflammatory phenotype. Please show that TNF-induced activation *in vivo* does not lead to release of S100A8/A9 in these mice. Please explain what the consequence of the lack of clear phenotype means for the importance of the current hypothesis.

We thank the reviewer for this valuable comment. Indeed, we performed similar experiments in our previous manuscript¹¹ using a monoclonal antibody against E-selectin (clone 9A9) *in vivo*. We found a strong reduction in serum S100A8/A9 levels in TNF-α treated WT mice in the presence of 9A9 compared to control mice (please refer to¹¹, Fig. 1d).

In order to address S100A8/A9 release in E-selectin deficient mice we injected rmTNF-α into the scrotum of C57BL/6 (WT) and E-selectin deficient (*Sele*^{-/-}) mice. S100A8/A9 serum levels were determined by ELISA before and 2h after TNF-α treatment. In WT mice, we found significantly increased serum levels of S100A8/A9 after TNF-α injection, which was absent in *Sele*^{-/-} animals (new Supplementary Fig. 1a). Importantly, number of neutrophils and monocytes, did not differ between WT and *Sele*^{-/-} animals (new Supplementary Fig. 1b and c).

These data are now included in the manuscript.

- MYD88 KO mice: redo the experiments relying on TAC242 and paquinimod.

In WT mice, we found significantly increased serum levels of S100A8/A9 after TNF- α injection, which surprisingly were absent in *MyD88*^{-/-} animals (Reviewer Fig. 3.2a). Interestingly, *MyD88*^{-/-} animals displayed normal amounts of GSDMD protein levels (Reviewer Fig. 3.2b), however the amount of cytosolic S100A8/A9 was significantly reduced in *MyD88*^{-/-} neutrophils as detected by Western Blot analysis (Reviewer Fig. 3.2c). To our knowledge, this reduction in cytosolic S100A8/A9 had not been described before, but could indeed explain the reduced S100A8/A9 serum levels after TNF- α application. To further elaborate on this, we also decided to investigate S100A8/A9 release *in vitro*, using the MyD88 inhibitor NBP2-29328, which leads to an acute inhibition of MyD88 instead of a permanent absence of MyD88 in *MyD88*^{-/-} mice. In this experimental setting, S100A8/A9 release was not affected (Reviewer Fig. 3.2d), suggesting that MyD88 is not substantially involved in acute and rapid E-selectin mediated GSDMD pore formation and S100A8/A9 release.

Reviewer Figure 3.2

Reviewer Figure 3.2 S100A8/A9 release upon genetic deletion and pharmacological inhibition of MyD88. **a)** Bone marrow neutrophils from WT and *MyD88*^{-/-} mice were treated with E-selectin or PBS (control) for 10min. (n=5 mice per group). Supernatants were collected and S100A8/A9 levels were analyzed. **b)** Basal GSDMD and **c)** S100A8/A9 protein levels of WT and *MyD88*^{-/-} neutrophils were determined by Western blot analysis (n \geq 5 mice per group). **d)** Bone marrow neutrophils were incubated with the MyD88 inhibitor NBP2-29328 or a control peptide (Ctrl-P) for 30min prior to stimulation with either E-selectin or PBS (control) for 10min. (n=5 mice per group). Supernatants were collected and S100A8/A9 levels were analyzed. Data is presented as mean \pm SEM (Two-way RM ANOVA, Sidak's multiple comparison for **a)** and **d)**, student's t-test for **b)** and **c)**), ns: not significant, *: p<0.05, **: p<0.01, ***: p<0.001.

- L-selectin KO mice or ADAM17 KO mice: please show that TNF-induced activation in vivo leads to modulated release of S100A8/A9.

Unfortunately and as mentioned in answer Q2 (reviewer 1) and Q3/Q10 (reviewer2) we had to take out the L-selectin shedding part. We apologize for this:

In the original manuscript we studied L-selectin shedding by flow cytometry using the well-established anti human L-selectin antibody clone DREG56 (old Figure 4c). However, from additional experiments (negative controls) we performed during the revision, we learned that, although cells were already fixed, L-selectin-DREG56 binding and the subsequent signal measured by flow cytometry was extremely sensitive to temperature fluctuations. We initially described that stimulation of E-selectin induced L-selectin shedding after 5min of incubation. However, we now know that this drop in L-selectin signal intensity was not due to E-selectin interacting with neutrophils, but rather due to temperature changes after neutrophils had been fixed and stained. Therefore, we had to take out those experiments.

To fill this gap and provide additional evidence for the transient nature of GSDMD pore formation and S100A8/A9 release, we focused on membrane repair mechanisms in neutrophils and uncovered the induction of ESCRT-III dependent membrane repair activity (CHMP4B staining, new Fig. 5 c and d) during stimulation of neutrophils by E-selectin.

4. The PI-uptake is of concern. It is presented as support for the hypothesis. It might also be interpreted as dying cells. The rescue is certainly an argument in favor of the hypothesis. So please show in a rolling assay under the fluorescent microscope that rolling neutrophils on TNF-activated HUVEC show transient PI-positive cells.

We thank the reviewer for his/her valuable comment. Since PI itself is toxic to cells, we think that it is not possible to test functional properties of neutrophils (e.g. rolling) after they have taken up PI. To account for cell viability in our assays, we measured lactate dehydrogenase (LDH) in E-selectin treated cell supernatants in a time dependent manner (old Figure 3g-i, 10min, 30min, 180min; new Fig. 4g-i). We were not able to detect significant amounts of LDH in the supernatants upon E-selectin treatment similar to PBS treated cells at none of the analyzed time points, suggesting that the cells do not undergo any form of cell death in response to E-selectin.

In order to investigate transient pore formation with another approach, we tested ESCRT-III activity after E-selectin treatment and stimulated neutrophils with E-selectin for 15min. Subcellular localization of CHMP4B, an ESCRT-III associated protein, was analyzed using confocal microscopy. ESCRT-III proteins are known to form a punctate pattern during membrane repair and to translocate to the plasma membrane^{5, 6}. Consistently, stimulation of neutrophils with E-selectin resulted in the formation of CHMP4B puncta, presumably corresponding to functional assemblies of ESCRT-III, while unstimulated cells (control) displayed a rather weak and diffuse cytoplasmic staining of CHMP4B (new Fig. 5c and d).

5. It is well known that L-selectin is shed upon activation in vitro, which is accompanied by upregulation of CD11b. In vivo, this is much less clear. Please provide evidence that l-selectin is shed in vivo in the mouse activated by TNF.

Please see our response to Q2 (reviewer 1) and Q3/Q10 (reviewer2) and Q3 of this reviewer.

6. Neutrophil priming: page 11 line 4 implies that neutrophils are ‘ready to go’ which is quite a statement as it ignores a large literature on priming of neutrophils by an array of inflammatory mediators that prime neutrophils for all sorts of activation.

We thank the reviewer for this comment. We agree with the reviewer that neutrophils were underestimated for many years regarding their functional adaptability and phenotypic heterogeneity. There is emerging evidence that neutrophils are versatile ^{14, 15, 16} and besides their classical function during acute inflammation, neutrophils play a crucial role in homeostatic tissue function and repair ¹⁷, in chronic inflammation ¹⁸ and tumor growth ¹⁹. In our context of rapid inflammasome activation however, we investigate the “classical” function of neutrophils during their recruitment to sites of inflammation. Within our setting of transient GSDMD-dependent pore formation and S100A8/A9 release, neutrophils are indeed “ready to go” as they do not need transcriptional/translational upregulation of cytokines or other proinflammatory mediators during the recruitment process, which only takes a few minutes. Therefore, we think that wording is correct in our context.

7. It is completely unclear how E-selectin induces signaling leading to activation of the inflammasome. Can the authors show that alternative activators of the inflammasome will also lead to release of S100A8/A9? What about chemokines?

We thank the reviewer for this important comment and agree with the reviewer that it is indeed very interesting to study additional NLRP3 inflammasome activators and chemokines. Please see also answer Q1 and Q5 of reviewer 2.

To test potential other NLRP3 inflammasome activators, we used isolated mouse neutrophils and stimulated the cells for 10min with PBS (negative control), E-selectin (positive control), nigericin, ATP, nigericin+LPS and CXCL1 (a chemokine binding to CXCR2). None of the well-known inflammasome activators such as ATP, nigericin and the combination of LPS/nigericin was able to induce S100A8/A9 release within 10min. The same was true for CXCR2-binding chemokine CXCL1 (new Fig. 3d and e and Reviewer Fig. 3.3). These findings suggest that rapid E-selectin triggered NLRP3 inflammasome activation is independent of ATP, nigericin or GPCR (CXCR2) signaling. Concerning chemokines, this is in line with Morikis et al. showing that CXCL8 (human analog to murine CXCL1) does not induce S100A8/A9 release in human neutrophils ⁴. As discussed in answer Q1, reviewer 2, potassium efflux is induced via K_v1.3 (new Fig. 3a-c, and new Supplementary Fig. 3a).

These data are now included in the manuscript.

Reviewer Figure 3.3

Reviewer Figure 3.3 Rapid S100A8/A9 release is specific to E-selectin stimulation. Bone marrow neutrophils from WT mice were treated with PBS, E-selectin, ATP (3mM) or CXCL1 (10nM) for 10min. Supernatants were collected and S100A8/A9 levels were analyzed ($n \geq 5$ mice per group). Data is presented as mean \pm SEM (one-way ANOVA, Tukey's multiple comparison); **: $p < 0.01$, ***: $p < 0.001$.

Minor:

8. Are neutrophils first line defense? What about tissue resident macrophages/

As elucidated in answer to Q6, we are aware of the growing number of neutrophil functions reaching beyond its role in first line defense and its interplay with tissue resident macrophages. We now acknowledged this in the discussion (page 14, line 19).

9. Please indicate that neutrophils under rotating conditions *in vitro* binding to rec-Eselectin-Fc is not the same as rolling on endothelial cells *in vivo*.

We apologize for not being clear regarding our experimental setup. We added a sentence (page 17, line 15).

10. It is important to know whether the differentiation and numbers and of leukocytes in the blood of GSDMD^{-/-} mice and Casp1/11^{-/-} mice is similar to the wt.

We agree with the reviewer that a proper interpretation of the *in vivo* S100A8/A9 release data (Fig. 1a) requires basal white blood cell (WBC) counts, especially neutrophil counts. Therefore, we measured number of leukocytes and of leukocyte subsets in peripheral blood samples of WT, *Casp-1^{-/-}* *Casp-11^{-/-}* and *Gsdmd^{-/-}* mice using a hemocytometer (ProCyte Dx; IDEXX Laboratories). We did not detect any differences neither in overall WBC counts, nor in the numbers of blood neutrophils, lymphocytes or monocytes between knockout animals and corresponding WT controls (which were coming out from the same animal facility as the respective ko mice) (number of neutrophils and monocytes in new Supplemental Figure 1d-g).

These data are now included in the manuscript.

11. Page 10 line 8: it is implied that murine neutrophils were used originating from the peripheral blood. Is this correct? If so how were these cells isolated?

We thank the reviewer for this comment and apologize for not being clear regarding the origin of the neutrophils we used. We used bone marrow neutrophils, which we isolated as described in the materials and method section. As suggested by reviewer 2 we now used the term bone marrow neutrophils throughout the revised version of the manuscript.

D. Appropriate use of statistics and treatment of uncertainties:

- Article is a bit basic on the statistics. E.G. Figure 1C and 1D are evaluated paired, which should be unpaired. The data are of different mice.

We have performed all statistical test in agreement with our Core Facility for Bioinformatics, Biomedical Center, LMU, Munich. Paired comparison accounts for potential variations in coating of the flow chambers and experimental procedure over different experimental days. To ensure comparability and reproducibility of single experiments, cells from both genotypes were used at each experimental day. Therefore, we decided to leave statistical testing as initially described, as we think it is correct.

E. Conclusions: robustness, validity, reliability:

1. Conceptual problem: The central dogma around the adhesion paradigm is the glue-like interaction of selectin/selectin ligands during rolling adhesion (no inside-out likely little outside-in signals) by which cells can keep rolling in the absence of interaction with chemokines that are presented by activated endothelial cells. The data of the current study imply very marked activation of neutrophils by E-selectin irrespective of the presence of chemokines as the pores will lead to a maximum increase in Ca^{2+} as well as complete depolarization as the gradient Na^+/K^+ will be completely gone. This situation seems hard to control and not very plausible. This needs very careful discussion which is completely absent now.

We agree with the reviewer that neutrophil activation is a rather complex event that can occur via multiple pathways, including TLR4 and GPCR signaling²⁰ but also via E-selectin as shown previously by us and others^{11, 21}. Here, we report on a new signaling function of E-selectin inducing NLRP3 inflammasome activation and transient GSDMD pore formation in neutrophils leading to neutrophil activation, but not to cell death. This is supported by the finding that extracellular LDH did not increase during E-selectin stimulation (Figure 4g-i). In addition, adding PI to isolated human neutrophils at various time points after the onset of E-selectin stimulation revealed that the formed pores (via which PI is entering neutrophils) are only transiently and reversibly present in the plasma membrane (Figure 5a and b).

As cell depolarization rather impairs than stimulates immune cell activation, uncontrolled pore formation would indeed lead to dramatic changes in the intracellular milieu and finally induce cell death. To address this important point, we investigated calcium influx induced by E-selectin and the chemokine CXCL8 in the presence and absence of MCC950 using the ratiometric calcium indicator Indo-1-AM and flow cytometry, an established method in our lab to assess calcium signaling⁷. As shown in Reviewer Figure 3.4, CXCL8 induced significant calcium influx, while E-selectin only led to a rather weak calcium influx. Of note, MCC950 did not affect calcium influx induced by chemokine or E-selectin. Therefore, these findings strongly suggest that transient GSDMD pore formation does not lead to an uncontrolled and potentially lethal situation for neutrophils.

Reviewer Figure 3.4

Reviewer Figure 3.4 E-selectin and CXCL8 dependent intracellular Ca²⁺ increase. Human neutrophils were stained with Indo-1 AM for 45 min at 37°C and then pre incubated with MCC950 (1 μ M) or PBS for 10 min. E-selectin (1 μ g ml⁻¹) or CXCL8 (10nM) stimuli were applied 30sec after baseline recording and changes in intracellular calcium measured by flow cytometry. (n=6 individual experiments). Data is presented as mean.

F. Suggested improvements: experiments, data for possible revision:

See above.

G. References: appropriate credit to previous work?:

Yes

H. Clarity and context: lucidity of abstract/summary, appropriateness of abstract, introduction and conclusions:

The hypothesis is clear, but the supportive evidence not sufficient (see above).

Reference List

1. Chen, J. & Chen, Z.J. PtdIns4P on dispersed trans-Golgi network mediates NLRP3 inflammasome activation. *Nature* **564**, 71-76 (2018).
2. Chase, S.D., Magnani, J.L. & Simon, S.I. E-selectin ligands as mechanosensitive receptors on neutrophils in health and disease. *Ann Biomed Eng* **40**, 849-859 (2012).
3. Rosen, S.D. Ligands for L-selectin: homing, inflammation, and beyond. *Annu Rev Immunol* **22**, 129-156 (2004).
4. Morikis, V.A. *et al.* Selectin catch-bonds mechanotransduce integrin activation and neutrophil arrest on inflamed endothelium under shear flow. *Blood* **130**, 2101-2110 (2017).

5. Ruhl, S. *et al.* ESCRT-dependent membrane repair negatively regulates pyroptosis downstream of GSDMD activation. *Science* **362**, 956-960 (2018).
6. Jimenez, A.J. *et al.* ESCRT machinery is required for plasma membrane repair. *Science* **343**, 1247136 (2014).
7. Immler, R. *et al.* The voltage-gated potassium channel KV1.3 regulates neutrophil recruitment during inflammation. *Cardiovasc Res* (2021).
8. Swanson, K.V., Deng, M. & Ting, J.P. The NLRP3 inflammasome: molecular activation and regulation to therapeutics. *Nat Rev Immunol* **19**, 477-489 (2019).
9. Song, N. *et al.* NLRP3 Phosphorylation Is an Essential Priming Event for Inflammasome Activation. *Mol Cell* **68**, 185-197 e186 (2017).
10. McKee, C.M. & Coll, R.C. NLRP3 inflammasome priming: A riddle wrapped in a mystery inside an enigma. *J Leukoc Biol* **108**, 937-952 (2020).
11. Pruenster, M. *et al.* Extracellular MRP8/14 is a regulator of beta2 integrin-dependent neutrophil slow rolling and adhesion. *Nat Commun* **6**, 6915 (2015).
12. Gritsenko, A. *et al.* Priming Is Dispensable for NLRP3 Inflammasome Activation in Human Monocytes In Vitro. *Front Immunol* **11**, 565924 (2020).
13. Yang, Y., Wang, H., Kouadir, M., Song, H. & Shi, F. Recent advances in the mechanisms of NLRP3 inflammasome activation and its inhibitors. *Cell Death Dis* **10**, 128 (2019).
14. Adrover, J.M. *et al.* A Neutrophil Timer Coordinates Immune Defense and Vascular Protection. *Immunity* **51**, 966-967 (2019).
15. Ballesteros, I. *et al.* Co-option of Neutrophil Fates by Tissue Environments. *Cell* **183**, 1282-1297 e1218 (2020).
16. Nemeth, T., Sperandio, M. & Mocsai, A. Neutrophils as emerging therapeutic targets. *Nat Rev Drug Discov* **19**, 253-275 (2020).
17. Phillipson, M. & Kuberski, P. The Healing Power of Neutrophils. *Trends Immunol* **40**, 635-647 (2019).
18. Soehnlein, O., Steffens, S., Hidalgo, A. & Weber, C. Neutrophils as protagonists and targets in chronic inflammation. *Nat Rev Immunol* **17**, 248-261 (2017).
19. Coffelt, S.B., Wellenstein, M.D. & de Visser, K.E. Neutrophils in cancer: neutral no more. *Nat Rev Cancer* **16**, 431-446 (2016).
20. Futosi, K., Fodor, S. & Mocsai, A. Neutrophil cell surface receptors and their intracellular signal transduction pathways. *Int Immunopharmacol* **17**, 638-650 (2013).
21. Block, H. *et al.* Crucial role of SLP-76 and ADAP for neutrophil recruitment in mouse kidney ischemia-reperfusion injury. *J Exp Med* **209**, 407-421 (2012).

Decision Letter, first revision:

24th Jun 2022

Dear Dr. Sperandio,

We have now finished reviewing your manuscript entitled "Transient gasdermin D pores control S100A8/A9 release from rolling neutrophils", reference number NI-A31849A.

Unfortunately, although our two inflammasome reviewers are really impressed by the paper, as you know, our neutrophil (rolling) expert had some major issues. We carefully looked at the criticisms raised by this reviewer and at your author response to those criticisms and decided that another opinion was warranted. So we recruited another neutrophil (rolling) expert to mediate (reviewer 4). As you will see below, this reviewer agrees with all of the comments from reviewer 3 and we are left with little choice now but to let you know that we cannot publish your paper.

Please note that we disagree with reviewer 4 comments regarding a lack of novelty in the paper. This is not the reason for the decision and is clearly not a view shared by the other reviewers, even by reviewer 3, or by the editorial team. There is clear novelty in the link between E-selectin and potassium efflux and in the transient nature of the gsdm pore and the role of ESCRT in the repair process, as well as this all being a mechanism for the release of S100.

However, we do agree with reviewer 3 and 4 that there are some major inconsistencies with regards to the proposed role of E-selectin mediated S100 release and how this might work in a physiological scenario.

The neutrophil reviewers are both clear that in their opinion and the majority of the published literature that the mechanism you are proposing does not quite gel with what is understood about constitutive expression and signalling by E-selectin. I have taken another look at your paper and I tend to agree that this concern is not resolved. Unfortunately, the reviewers have not provided clear guidance as to how you can resolve this issue, and as this is the second set of reviews to make this comment, we have now decided to reject the paper.

If you choose to appeal this decision, I would say that perhaps the only chance of success you might have would be if you can better plug some of the gaps in the proposed mechanism. Your paper states "we wanted to uncover the precise mechanism and the involved intracellular processes of inflammation-driven S100A8/A9 release from neutrophils". It seems to us that you have not quite achieved this goal, as we think a critical mechanistic gap here is in a missing link between E-selectin and the activation of the potassium channels. How does E-selectin interact with or regulate the potassium channel? Is there not room in here somewhere for chemokine or other activation signals that enable your proposed mechanism, that might appease our reviewer concerns? It might be the case that if you can define this link mechanistically then some of the broader complaints might be resolved, but even this is not clear to us and the reviewers were not explicitly asking you to better define this particular mechanistic link, so we can't really offer you a chance to revise the paper by further studying this link. This is just an editorial suggestion that we think might help an appeal or submission elsewhere.

We realize that this is disappointing. I hope that you continue to consider Nature Immunology for your results most significant for the immunology community and wish you well in your future investigations.

As noted above, I am happy to chat about this.

Sincerely,

Nick Bernard, PhD
Senior Editor
Nature Immunology

Reviewers' comments:

Reviewer #1 (Remarks to the Author):

My comments have been addressed.

Reviewer #2 (Remarks to the Author):

In this exciting article from the Sperandio lab, the authors demonstrate the novel findings that e-selectin-engaged (rolling) neutrophils trigger the NLRP3-CASP1-GSDMD signalling axis, to secrete S100 alarmins for firm adhesion to the vascular endothelium, allowing for neutrophil extravasation to tissue. This manuscript has been thoroughly revised to address the Reviewers' comments, and now includes further mechanistic data that expands on, and refines, the model of the original manuscript submission; many of 'missing mechanistic links' of the previous submission are now very nicely filled. The discovery that Kv1.3 mediates K⁺ efflux is a particularly interesting addition to this work. My congratulations to the authors on this very exciting study. I have only minor suggestions for manuscript improvement (listed below). My recommendation is for publication with only minor revisions.

Minor concerns/suggestions:

1. Page 2 abstract end: suggest replacing "independent of prior priming" with "independent of prior inflammasome priming" for greater clarity.
2. Supplementary Figures 2a-b: these data for ASC oligomerisation are puzzling and do not support the proposed signalling model as suggested ("ASC oligomerization in neutrophils [was] induced by E-selectin"). It is unclear whether the method was appropriate for identifying ASC oligomers – was a crosslinking agent used, or were gels non-denaturing? My reading of these methods suggests that neither of these approaches were used, in which case, how were "ASC oligomers" captured by this method? Regardless, the figure indicates significant bands purported to be ASC oligomers in both resting and stimulated cells, and these conditions are almost indistinguishable from each other. Given that the neutrophils are coated onto slides prior to analysis, it would have been much simpler to detect ASC specks by microscopy rather than by biochemical means – suggest replacing S2a-b with microscopic evaluation of ASC specks (if you are hunting for a suitable antibody, adipogen antibody

#AL177 works well for human ASC microscopy).

3. Page 7 line 10 sentence: "Interestingly, blockade of NLRP3 activation with MCC950 resulted in reduced S100A8/A9 release upon E-selectin stimulation in vitro and in vivo which was similar to the reduction observed in Casp-1^{-/-}Casp-11^{-/-} mice (Supplementary Fig. 2b and 2c) suggesting a predominant role of the canonical inflammasome pathway in E-selectin triggered S100A8/A9 release." Actually the MCC950 data indicate that NLRP3 triggers S100A8/9 release, but does not provide strong evidence that *canonical NLRP3* signalling is involved. Suggest rewording to "... suggesting a predominant role of the NLRP3 inflammasome pathway in E-selectin triggered S100A8/A9 release". The data presented immediately prior to this sentence (showing that MCC950 blocks GSDMD cleavage) does in fact support a role for canonical NLRP3 signalling here – as caspase-11/4/5-induced (noncanonical) NLRP3 signalling would place GSDMD cleavage above NLRP3 in the signalling hierarchy (i.e. Caspase-11/4/5 -> GSDMD cleavage -> K⁺ efflux -> NLRP3 signalling). Suggest changing this earlier sentence to "...suggesting that canonical NLRP3 activation contributes to GSDMD cleavage and GSDMD-NT surface mobilization following stimulation of neutrophils with E-selectin".

4. Page 8 line 15 "K⁺-efflux is known to be induced in response to most canonical NLRP3 stimuli, including extracellular ATP and nigericin" – suggest removing the word "canonical" here, as noncanonical NLRP3 activation also requires K⁺ efflux.

5. Page 10 text describing S4A-D – while the authors have technically avoided stating that human neutrophil death induced by nigericin is pyroptosis, it is very strongly implied. Nigericin is very well known to induce human neutrophil NETosis rather than pyroptosis – so this is misleading. Suggest adding a sentence acknowledging that human neutrophil death induced by nigericin is likely to reflect NETosis rather than pyroptosis.

6. Throughout the manuscript the CHMP4B microscopy data (Fig 5) is reported to show that the neutrophil self-repair program is ESCRT-III dependent. While the CHMP4B data is consistent with an ESCRT-III dependent self-repair program, it does not actually show that self-repair requires the ESCRT machinery (ie. it is an association, rather than establishing causation). To make claims that self-repair indeed requires ESCRT-III, it would be necessary to perturb the ESCRT-III system (e.g. by knockout).

7. Recent papers indicate that IL-1 β cleavage is required for its exit via GSDMD pores (DOI: 10.1016/j.celrep.2018.07.027) because GSDMD pores function as an electrostatic filter, allowing passage of neutral and positively charged cargo but excluding negatively charged cargo (doi:10.1038/s41586-021-03478-3). What is the charge of S100A8/A9, and what is its expected electrostatic interactions with the GSDMD pore? This point would be interesting to raise in the discussion.

Reviewer #3 (Remarks to the Author):

Prüenster and colleagues have responded to many issues raised by the reviewers and changed their manuscript accordingly. The article has definitely improved. However, there are some lingering and important issues as the authors did not answer all my questions to my satisfaction. It all concerns the translation of their findings/suggestions/ conclusions to findings made in vivo.

E1. Conclusions: robustness, validity, reliability

Activation of neutrophils by E-selectin irrespective of preactivation. The authors decided not to engage in the discussion that the dogma in the field that rolling neutrophils do not arrest unless chemokines activate the beta-2 integrins does not seem to be compatible by their data. Their model of E-selectin directly activating neutrophils surely leads to firm attachment of rolling neutrophils without the

requirement of chemokines or other inflammatory mediators. This is rather essential as constitutive expression of E-selectin also in homeostasis would always lead to firm attachment which is not seen in vivo. Also the idea that marginated neutrophils are at least in part rolling cells on selectins is not compatible with the current study. Demargination by steroids or exercise leads to massive increase in cell numbers without indications of activation of the cells that you would expect when the model of the article is correct.

C1. Most experiments on bone marrow murine neutrophils and only few with human neutrophils. This issue is important as there are major differences between bone marrow neutrophils in the mouse and peripheral blood neutrophils in man. Please compare murine and human neutrophils when reasonably possible. For instance:

- The experiments of release of S100A8/A9 by neutrophils activated on rotating culture flasks coated with E-selectin must also be performed with human cells.
- Experiments with blocking antibodies against Fc-receptors must also be carried out with human cells as the Fc-receptor repertoire is completely different in mice. Also a control is missing that the Fc-receptors are sufficiently inhibited under the conditions of the experiment.

So please add experiments with human neutrophils in all experiment with murine cells if possible (surely experiments relying on KO-mice cannot be compared).

C2. Please show whether there is activation of murine and human neutrophils by rolling on E-selectin in the context of neutrophil activation markers such as CD11b, CD62L, CD66, CD63. Again the translation to in vivo (presence of persistent rolling cells) should be carefully addressed.

C3. I am concerned that the number of neutrophils (bone marrow?) and monocytes (bone marrow?) did not differ in the Sele-/- mice where you might expect neutrophilia such as found in LAD-2 patients. Please discuss.

C6. Priming of neutrophil responses by chemokines and chemotaxins is almost instantaneous (seconds). Please rediscuss the original issue as the timing issue does not seem convincing.

Reviewer figure 3.4 is a mystery to me. If the pores are large enough to allow PI and S100 proteins to pass the membrane I would have expected saturation of the INDO-1 inside the cells. It is even of concern that the INDO-1 in its 'acid' form would also diffuse out of the cells along the same pores.

Reviewer #4 (Remarks to the Author):

I would like to thank you for inviting me to review the revised manuscript and the review process. In this paper, the authors investigated the molecular mechanism of E-selectin-mediated signaling and demonstrated that E-selectin triggers Kv1.3 channel-mediated potassium efflux resulting in NLRP3 inflammasome activation.

Reviewer 3 raises very good points that are valid and are not adequately addressed in the response to reviewers letter.

To be honest, I am quite surprised that this manuscript made it into the review process, because of the lack of novelty, methodological issues, and lack of citing existent literature. My major concerns are:

- 1) The link between Kv1.3 and the activation of the NLRP3 is not novel, because this link has been already established (e.g. PMID: 32652099).
- 2) In accordance with reviewer 3, the authors ignore the existing literature on E-selectin-mediated signaling and claim that E-selectin engagement induces slow leukocyte rolling and arrest (page 4, line 21). This is not true as shown in many papers (PMID: 34435628; PMID: 29592875; PMID: 22511754; PMID: 20299514; PMID: 20445017; PMID: 22021370; PMID: 17543554). Binding of E-selectin to its receptor on neutrophils induces the extended conformation of LFA-1 which mediates slow leukocyte rolling, but not arrest (PMID: 34435628; PMID: 29592875; PMID: 22511754; PMID: 20299514; PMID: 20445017; PMID: 22021370; PMID: 17543554). Leukocyte arrest requires the full activation of LFA-1 which is triggered by chemokines. This has also been shown in in vivo experiments. Blocking E-selectin alone does not reduce the number of adherent or transmigrated cells in vivo (PMID: 15466624; PMID: 17543554). These data do not go along with the data shown in Figure 6 as Gsdmd^{-/-} mice have a reduced leukocyte adhesion and transmigration, suggesting that the elimination of gasdermin D is also involved in other processes.
- 3) In line with point 2, gasdermin D is expressed in different tissues. However, in figure 6 the authors used the full knock-out mouse. These experiments cannot exclude that the seen phenotype is caused by the lack of gasdermin D in other tissues.
- 4) It has been shown that E-selectin binding triggers different signaling pathways. As integrin activation occurs instantaneously after E-selectin engagement, it is not comprehensible how the gasdermin/NLRP3 pathway is involved in E-selectin-mediated slow leukocyte rolling.

Appeal Letter:

Dear Professor Sperandio,

Thank you for your letter asking us to reconsider our decision on your manuscript, "Transient gasdermin D pores control S100A8/A9 release from rolling neutrophils".

Now that I have had a chance to discuss the matter carefully with my colleagues, I am happy to say that we would consider sending your manuscript back to external review if you do indeed provide some clearer mechanistic insight to the link between E-selectin and the activation of the potassium channels as outlined in depth in my last decision letter. I'm sure, however, that you'll understand that we cannot predict the outcome of the review process and we will need some positive sentiments from reviewers other than our inflammasome experts here.

Once you have made your revisions, please use the URL below to submit the revised manuscript with figures and a revised version of the life sciences reporting summary. It will be available to referees (and, potentially, statisticians) to aid in their evaluation if the manuscript goes back for peer review. A revised checklist is essential for re-review of the paper.

The Reporting Summary can be found here:
<https://www.nature.com/documents/nr-reporting-summary.pdf>

The Editorial Policy Checklist can be found here: <https://www.nature.com/documents/nr-editorial-policy-checklist.pdf>

[REDACTED]

Please let us know how you wish to proceed and when we can expect your revised manuscript.

With kind regards,

Nick Bernard, PhD
Senior Editor
Nature Immunology

Author Rebuttal, Appeal:

See inserted PDF

Point to point discussion:

After another round of revisions we gained additional insights into the molecular mechanism of E-selectin induced rapid NLRP3 inflammasome activation.

We found that E-selectin engagement of neutrophils results in Btk-dependent tyrosine phosphorylation of NLRP3. Btk is known to signal downstream of E-selectin engagement ¹ and Btk dependent activation of the NLRP3 inflammasome was described before ^{2, 3, 4}. Hence, using the Btk inhibitor ibrutinib, which already made it into clinical trials, we were able to block E-selectin-induced NLRP3 tyrosine phosphorylation and S100A8/A9 release in vitro. Interestingly, tyrosine phosphorylation of NLRP3 did not require potassium efflux, as E-selectin-induced NLRP3 tyrosine phosphorylation was found to be fully functional in Kv1.3 deficient cells. Downstream of NLRP3 tyrosine phosphorylation, we detected E-selectin-induced ASC oligomerization as demonstrated by western blot analysis and, as new data, confocal microscopy (see also reviewer 2, point 2 and Fig. 4d and e). Interestingly, ASC oligomerization and ASC speck formation was dependent on potassium efflux via Kv1.3, as we could not detect ASC oligomers/ASC specks using Kcna3^{-/-} neutrophils (Fig. 4f and g).

Further, we confirmed rapid, E-selectin-induced release of the small cytoplasmic alarmins S100A8/A9 by mass spectrometry-based proteomics of supernatants from E-selectin versus PBS stimulated human neutrophils (Supplementary Fig. 3c and d). In addition to S100A8/A9, we detected Macrophage migration-inhibitory factor (MIF) in the supernatants of E-selectin stimulated cells (Supplementary Fig. 3c and d). Taken together, we found a significant enrichment of the term "small cytosolic alarmin" in an unbiased 1D annotation enrichment (Supplementary Fig. 3e), while larger alarmins (such as HSP70 and HSP90, Supplementary Fig. 3c-e) and granule content (reviewer figure 3.1.d-f) were not enriched in the supernatants after 10min E-selectin stimulation. This data clearly support our hypothesis that E-selectin-induced NLRP3 inflammasome activation and reversible GSDMD pore formation in neutrophils is a fine tuned mechanism resulting in the release of small cytosolic alarmins, without general (over-) activation (as indicated by granule release).

Finally, we determined E-selectin-induced pore formation as reflected by changes in membrane potential of E-selectin stimulated primary neutrophils using patch-clamp recordings (Supplementary Fig. 4a-c). Preventing pore formation by incubating the cells with the Kv1.3 inhibitor PAP-1 or the GSDMD blocker disulfiram prevented depolarization of the cells.

Please find below our specific responses to the reviewers' comments:

Reviewers' comments:

Reviewer #1 (Remarks to the Author):

My comments have been addressed.

We thank reviewer 1 for supporting our work to be published in Nature Immunology

Reviewer #2 (Remarks to the Author):

In this exciting article from the Sperandio lab, the authors demonstrate the novel findings that e-selectin-engaged (rolling) neutrophils trigger the NLRP3-CASP1-GSDMD signalling axis, to secrete S100 alarmins for firm adhesion to the vascular endothelium, allowing for neutrophil extravasation to tissue. This manuscript has been thoroughly revised to address the Reviewers' comments, and now includes further mechanistic data that expands on, and refines, the model of the original manuscript submission; many of 'missing mechanistic links' of the previous submission are now very nicely filled. The discovery that Kv1.3 mediates K⁺ efflux is a particularly interesting addition to this work. My congratulations to the authors on this very exciting study. I have only minor suggestions for manuscript improvement (listed below). My recommendation is for publication with only minor revisions.

We thank reviewer 2 for his/her favorable comments on our revised manuscript and for advocating publication in Nature Immunology after minor revisions.

Minor concerns/suggestions:

1. Page 2 abstract end: suggest replacing “independent of prior priming” with “independent of prior inflammasome priming” for greater clarity.

We changed the abstract in the current version of the manuscript. The statement was removed.

2. Supplementary Figures 2a-b: these data for ASC oligomerization are puzzling and do not support the proposed signalling model as suggested (“ASC oligomerization in neutrophils [was] induced by E-selectin”). It is unclear whether the method was appropriate for identifying ASC oligomers – was a crosslinking agent used, or were gels non-denaturing? My reading of these methods suggests that neither of these approaches were used, in which case, how were “ASC oligomers” captured by this method? Regardless, the figure indicates significant bands purported to be ASC oligomers in both resting and stimulated cells, and these conditions are almost indistinguishable from each other. Given that the neutrophils are coated onto slides prior to analysis, it would have been much simpler to detect ASC specks by microscopy rather than by biochemical means – suggest replacing S2a-b with microscopic evaluation of ASC specks (if you are hunting for a suitable antibody, adipogen antibody #AL177 works well for human ASC microscopy).

We thank reviewer 2 for this valuable comment. We now repeated the experiments with an adopted protocol (crosslinking of ASC specks) according to Hoss et al ⁵ using WT and Kcna3^{-/-} bone marrow neutrophils. Again, we found an increase in ASC oligomers after stimulating neutrophils for 10min with E-selectin when compared to control cells (Fig. 4d). Interestingly, we were unable to detect an increase in ASC oligomers in E-selectin-treated Kcna3^{-/-} neutrophils compared to control (PBS) treatment (Fig. 4f), indicating that potassium efflux via K_v1.3 is a prerequisite for ASC oligomerization in E-selectin-induced NLRP3 inflammasome assembly.

As suggested by reviewer 2, we also investigated E-selectin-induced ASC speck formation by confocal microscopy using the #AL177 antibody from Adipogen. In line with the western blot results, bone marrow neutrophils from WT mice exhibited an increased area of ASC specks upon E-selectin stimulation compared to control (Fig.4e). Again, E-selectin induced ASC speck formation was absent in Kcna3^{-/-} cells (Fig. 4g).

We have included these results into the revised version of the manuscript.

3. Page 7 line 10 sentence: “Interestingly, blockade of NLRP3 activation with MCC950 resulted in reduced S100A8/A9 release upon E-selectin stimulation in vitro and in vivo which was similar to the reduction observed in Casp-1^{-/-}-Casp-11^{-/-} mice (Supplementary Fig. 2b and 2c) suggesting a predominant role of the canonical inflammasome pathway in E-selectin triggered S100A8/A9 release.” Actually the MCC950 data indicate that NLRP3 triggers S100A8/9 release, but does not provide strong evidence that *canonical NLRP3* signalling is involved. Suggest rewording to “... suggesting a predominant role of the NLRP3 inflammasome pathway in E-selectin triggered S100A8/A9 release”. The data presented immediately prior to this sentence (showing that MCC950 blocks GSDMD cleavage) does in fact support a role for canonical NLRP3 signalling here – as caspase-11/4/5-induced (noncanonical) NLRP3 signalling would place GSDMD cleavage above NLRP3 in the signalling hierarchy (i.e. Caspase-11/4/5 -> GSDMD cleavage -> K⁺ efflux -> NLRP3 signalling). Suggest changing this earlier sentence to “...suggesting that canonical NLRP3 activation contributes to GSDMD cleavage and GSDMD-NT surface mobilization following stimulation of neutrophils with E-selectin”.

As suggested, we changed this in the current version of the manuscript.

4. Page 8 line 15 “K⁺-efflux is known to be induced in response to most canonical NLRP3 stimuli, including extracellular ATP and nigericin” – suggest removing the word “canonical” here, as noncanonical NLRP3 activation also requires K⁺ efflux.

As suggested, we changed this in the current version of the manuscript.

5. Page 10 text describing S4A-D – while the authors have technically avoided stating that human neutrophil death induced by nigericin is pyroptosis, it is very strongly implied. Nigericin is very well known to induce human neutrophil NETosis rather than pyroptosis – so this is misleading. Suggest adding a sentence acknowledging that human neutrophil death induced by nigericin is likely to reflect NETosis rather than pyroptosis.

We thank the reviewer for this comment. As suggested, we now stated that “several reports implicated GSDMD pores to drive neutrophil pyroptosis ⁶ and to play a role in the generation of neutrophil extracellular traps ^{7, 8} (page 12, line 19f.).

6. Throughout the manuscript the CHMP4B microscopy data (Fig 5) is reported to show that the neutrophil self-repair program is ESCRT-III dependent. While the CHMP4B data is consistent with an ESCRT-III dependent self-repair program, it does not actually show that self-repair requires the ESCRT machinery (ie. it is an association, rather than establishing causation). To make claims that self-repair indeed requires ESCRT-III, it would be necessary to perturb the ESCRT-III system (e.g. by knockout).

We have discussed this question with our ESCRT-III expert Petr Broz (co-author on the paper). Demonstrating the involvement of the ESCRT machinery would have been quite tricky as manipulating the cells by either expressing dominant-negative versions of ESCRT proteins or knocking down ESCRT components is not feasible in short lived neutrophils. Therefore, we decided to change the wording in the text and now speak of an association.

7. Recent papers indicate that IL-1beta cleavage is required for its exit via GSDMD pores (DOI: 10.1016/j.celrep.2018.07.027) because GSDMD pores function as an electrostatic filter, allowing passage of neutral and positively charged cargo but excluding negatively charged cargo (doi:10.1038/s41586-021-03478-3). What is the charge of S100A8/A9, and what are its expected electrostatic interactions with the GSDMD pore? This point would be interesting to raise in the discussion.

This is an interesting point and at the moment we only have a hypothesis how it might work. S100A8 and S100A9 have isoelectric points (IEPs) which are much lower (IEP_{S100A8} : pH 5.7, IEP_{S100A9} : pH 5.7) than processed IL-1beta ($IEP_{IL-1beta}$: pH 8.8 compared to full length pro-IL-1beta $IEP_{pro-IL-1beta}$: pH 4.6) ⁹. This would mean that S100A8 and S100A9 is negatively charged at pH 7.2 (intracellular pH in neutrophils) and according to the work by Monteleone et al.⁹, it might impair the release of S100A8/A9 through GSDMD pores. Interestingly, it has been demonstrated that calcium binding by S100A8/A9 has significant impact on its net

charge thereby increasing its hydrophobicity and making it more positively charged¹⁰. Assuming now the presence of open GSDMD pores, high free extracellular Ca²⁺ concentration will trigger Ca²⁺ influx through the pores into the cell (this is in fact indirectly supported by our finding of pore-induced depolarization, Suppl. Fig 4a). Local increase in free calcium ions at pore sites might then lead to an increase in calcium binding by S100A8/A9 making it more positively charged and allowing its release through the pores. We have discussed this in the new version of the manuscript (page 18, line 9-11)

Reviewer #3 (Remarks to the Author):

Pruenster and colleagues have responded to many issues raised by the reviewers and changed their manuscript accordingly. The article has definitely improved. However, there are some lingering and important issues as the authors did not answer all my questions to my satisfaction. It all concerns the translation of their findings/suggestions/ conclusions to findings made in vivo.

We thank the reviewer for his/her comments and appreciate that reviewer 3 acknowledges our efforts in improving the present manuscript.

In the following point-to-point response we tried to engage on the raised concerns and we hope that we can convince reviewer 3 on the biological relevance of the proposed mechanism focusing on how the alarmin and DAMP molecule S100A8/A9 is released by neutrophils during their recruitment into inflamed tissue.

E1. Conclusions: robustness, validity, reliability

Activation of neutrophils by E-selectin irrespective of preactivation. The authors decided not to engage in the discussion that the dogma in the field that rolling neutrophils do not arrest unless chemokines activate the beta-2 integrins does not seem to be compatible by their data. Their model of E-selectin directly activating neutrophils surely leads to firm attachment of rolling neutrophils without the requirement of chemokines or other inflammatory mediators. This is rather essential as constitutive expression of E-selectin also in homeostasis would always lead to firm attachment which is not seen in vivo. Also the idea that marginated neutrophils are at least in part rolling cells on selectins is not compatible with the

current study. Demargination by steroids or exercise leads to massive increase in cell numbers without indications of activation of the cells that you would expect when the model of the article is correct.

In the present study, we did not intend to question the importance of chemokines in beta2 integrin activation of neutrophils and we agree with reviewer 3 that under in vivo conditions E-selectin downstream signaling cooperates with additional signaling cascades (i.e. chemokines and other signaling molecules) in mediating neutrophil firm arrest. However, we disagree with some of the raised statements and therefore we would like to clarify some misunderstandings.

The dogma, that “rolling neutrophils do not arrest unless chemokines activate the beta-2 integrins” does to our understanding not completely cover the current picture of beta2 integrin activation. There is solid evidence in the literature (including our own work) showing that various mediators are able to activate beta2 integrins independent of chemokines, f.e. TNF- α ¹¹, amyloid- β ¹², TLR2 and TLR5 agonists¹³ and TLR4 agonists^{14, 15}.

From our point of view, we never claimed that E-selectin directly fully activates beta2 integrins, but we agree with reviewer 3 that the sentence he/she is most likely referring to (“Consequently, neutrophils reduce their rolling velocity and undergo firm arrest”) may be misleading to some extent. Accordingly, we changed this sentence in the current version of the manuscript. A few years ago, we reported that E-selectin stimulation induces S100A8/A9 release from neutrophils and that released S100A8/A9 in turn mediates integrin activation via TLR4 signaling¹⁴. To our knowledge, this concept has not been refuted. In fact, other groups confirmed the finding that S100A8/A9 and other TLR4 agonists are able to mediate beta2 integrin activation^{15, 16} affecting also neutrophil adhesion^{17, 18}.

Furthermore, reviewer 3 raised concerns regarding translational aspects of our findings with findings in vivo. We are convinced that our mechanism published in 2015¹⁴ fits well to the in vivo observations described in the current manuscript. Engagement of E-selectin on rolling neutrophils was described by the McEver group to promote $G_{\alpha i}$ -independent (chemokine-independent) beta2 integrin activation in vivo resulting in slow rolling and affecting neutrophil recruitment (peritonitis model)¹⁹. In line with the data by Yago et al.¹⁹ and Smith et al.³³, our group has demonstrated that E-selectin and chemokines ($G_{\alpha i}$ signaling) cooperate

in an overlapping fashion to mediate firm leukocyte arrest in TNF-alpha stimulated cremaster muscle venules³⁴. Blocking either CXCR2 (with pertussis toxin or using Cxcr2^{-/-} mice) or E-selectin (blocking mAb 9A9 or using Sele^{-/-} mice) alone did not change the number of adherent leukocytes in TNF-alpha stimulated cremaster muscle venules. However, blocking both, CXCR2 and E-selectin almost completely abolished leukocyte adhesion in this neutrophil driven inflammatory disease model^{20, 21}. Hence, our model of E-selectin induced S100A8/A9 release and subsequent TLR4 dependent integrin activation published in 2015 is to our understanding not conflicting with all of the published models, but can be considered as an extension. Accordingly, we rephrased the text (p.5, l.17ff.) into: "We have shown before that secreted S100A8/A9 binds to TLR4 in an autocrine manner thereby activating β 2 integrins on neutrophils. This resulted in deceleration of rolling neutrophils along the inflamed vessel wall (slow rolling), facilitating firm leukocyte arrest." ¹⁴.

Concerning constitutive expression of E-selectin (and VCAM-1), this was described in 1996 on bone marrow endothelial cells by indirect immunoperoxidase staining²². This constitutive expression, although rather weak, was associated with homing of hematopoietic progenitor cells (HPCs). This was later functionally confirmed by the Hidalgo group²³, who showed that the requirements for homing were mostly based on the expression of ESL-1 on HPCs and not PSGL-1 (the main E-selectin ligand for S100A8/A9 release in murine neutrophils as demonstrated by our group in 2015¹⁴). Furthermore, the same study by Sreeramkumar et al. also found that ESL-1 is not involved in beta2 integrin activation²³. Therefore, it might be difficult to transfer these findings from the bone marrow microvasculature to our model of E-selectin induced inflammasome activation during neutrophil recruitment into inflamed peripheral tissue.

Regarding the margined pool: this term often refers to a pool of neutrophils found in the lung of unstimulated mice. Earlier experiments by Nancy Hogg and Claire Doerschuk showed that this pool of neutrophils in the lung is localized to capillaries of the lung²⁴. These capillaries show a characteristic architecture promoting neutrophil 'margination' which is independent of E-selectin (and P-selectin) as shown later by the Doerschuk laboratory²⁵. For other tissues, margination of neutrophils has been considered a purely rheological phenomenon caused by the presence of red blood cells pushing leukocytes from the central flow towards the

vessel wall ^{26, 27}, which is independent of selectins or leukocyte-endothelial interactions. Interestingly, a recent report in PNAS by Fay et al. revealed that demargination of leukocytes induced by systemic glucocorticoids or catecholamines (exercise) is again a purely rheological and biomechanical phenomenon explained by a reduction in leukocyte stiffness (with redistribution of WBCs in the flowing blood) and independent of leukocyte-endothelial interactions ²⁸.

C1. Most experiments on bone marrow murine neutrophils and only few with human neutrophils. This issue is important as there are major differences between bone marrow neutrophils in the mouse and peripheral blood neutrophils in man. Please compare murine and human neutrophils when reasonably possible. For instance:

- The experiments of release of S100A8/A9 by neutrophils activated on rotating culture flasks coated with E-selectin must also be performed with human cells.
- Experiments with blocking antibodies against Fc-receptors must also be carried out with human cells as the Fc-receptor repertoire is completely different in mice. Also a control is missing that the Fc-receptors are sufficiently inhibited under the conditions of the experiment.

So please add experiments with human neutrophils in all experiment with murine cells if possible (surely experiments relying on KO-mice cannot be compared).

We have performed most of our experiments also with human neutrophils (please refer to Fig. 2, Fig. 3a,b,g, Fig. 5a,b,c,g,h,i,j,k,l,m, Supplementary Fig. 2d, Supplementary Fig. 3a,c-e, Supplementary Fig. 4), so we do not agree that only a few experiments have been conducted with human neutrophils.

For all human neutrophil experiments (with one exception, Supplementary Fig. 2d), we have used recombinant E-selectin without Fc-tag. E-selectin induced caspase-1 cleavage, GSDMD pore formation, S100A8/A9 release, CHMP4B translocation and changes in the membrane potential are therefore not relying on Fc. Hence, we feel that additional experiments blocking Fc receptors in the human system are not necessary. Only one experiment in the flow chamber, shown in Supplementary Fig. 2d, was conducted with human recombinant E-selectin-Fc, as this construct binds much better to the flow chamber surface.

C2. Please show whether there is activation of murine and human neutrophils by rolling on E-selectin in the context of neutrophil activation markers such as CD11b, CD62L, CD66, CD63. Again, the translation to *in vivo* (presence of persistent rolling cells) should be carefully addressed.

We thank the reviewer for this valuable comment. Due to the low number of neutrophils we get from flow chamber experiments (with cells rolling on E-selectin) and the fact that we cannot distinguish rolling from free flowing neutrophils in the flow chamber when collecting the cells, we used isolated human neutrophils incubated with soluble E-selectin instead.

We assessed potential changes in surface expression levels of CD63, CD66b and CD11b upon E-selectin treatment using flow cytometry. 10min of E-selectin stimulation did not result in upregulation of CD63, CD66b or CD11b compared to negative control (Reviewer Fig. 3.1a-c), suggesting that E-selectin does not induce the secretion of azurophilic (CD63), specific (CD66b, CD11b), gelatinase, or secretory vesicles (both CD11b). Because of the issues with the L-selectin antibody in the original version of the manuscript we decided not to investigate L-selectin levels here.

In addition, we performed an unbiased secretome analyses and did not detect an increase in the amount of azurophilic (MPO, neutrophil elastase, cathepsin G), specific (cathelicidin, lactoferrin, MMP8) and gelatinase (MMP2, lysozyme) granules in the supernatants of E-selectin stimulated neutrophils compared to control (Reviewer Figure 3.1 d-f).

Reviewer Figure 3.1

Reviewer Figure 3.1 E-selectin does not induces granule release in human neutrophils within 10min of stimulation. *a-c* Isolated human neutrophils were stimulated with E-selectin ($1\mu\text{g ml}^{-1}$), PMA (100nM, positive control) or PBS (control) for 10min at 37°C under shaking conditions. Surface expression of **a** CD63, **b** CD66b and **c** CD11b was assessed using flow cytometry ($n=4$ independent experiments). **d-f** Enrichment of granule content (azurophilic gr., specific gr., gelatinase gr.) in supernatants of E-selectin ($1\mu\text{g ml}^{-1}$) and PBS (control) stimulated human neutrophils was analyzed by mass spectrometry ($n=3$ independent experiments). Data is presented as mean \pm SEM for **a-c** (one-way ANOVA, Tukey's multiple comparison), as representative volcano plot for **d**, as heat map for **e** and box-whisker-plots for **f** with dots indicating proteins outside the 95% confidence interval. $-\log_{10}$ transformed p -values depicted in the volcano plot were determined by Welch's two-sided t -test. Enrichment of small granule content in **e** was determined by a 1D annotation enrichment with Benjamin-Hochberg FDR=0.02. ns: not significant; *: $p<0.05$; ***: $p<0.001$.

C3. I am concerned that the number of neutrophils (bone marrow?) and monocytes (bone marrow?) did not differ in the $\text{Sele}^{-/-}$ mice where you might expect neutrophilia such as found in LAD-2 patients. Please discuss.

We apologize for not being clear here. In Supplementary Figure 1 both, neutrophil and monocyte counts of $\text{Sele}^{-/-}$, $\text{Gsdmd}^{-/-}$ and $\text{Casp1}^{-/-}\text{Casp11}^{-/-}$ animals were analyzed in peripheral blood samples. We found normal leukocyte counts in the blood circulation of $\text{Sele}^{-/-}$ mice compared to littermate controls, which is in line with previous reports ^{29, 30, 31} indicating that the loss of E-selectin can be compensated under homeostatic conditions. This has also been reported for other adhesion deficient mice including f.e. L- and P-selectin deficient mice ^{32, 33 34, 35}. However, some publications have also reported moderately elevated leukocytes counts in 'single selectin' deficient mice which might be related to differences in mouse strains and animal facilities.

C6. Priming of neutrophil responses by chemokines and chemotaxins is almost instantaneous (seconds). Please rediscuss the original issue as the timing issue does not seem convincing.

We thank the reviewer for this comment. Work by Klaus Ley's group has demonstrated that neutrophils roll on average 270 μm along inflamed endothelium (TNF-alpha cremaster muscle model in vivo) until they firmly adhere ³⁶. At a rolling

velocity of around $4\mu\text{m/s}$, this corresponds to around 70 seconds and plenty of time for neutrophils to interact with E-selectin ³⁶. This clearly indicates that neutrophils do not 'almost instantaneously' go from rolling to arrest under *in vivo* conditions although chemokines are expressed on the inflamed endothelium. We have confirmed this *in vivo* observation many times over the last 20 years ^{37, 38}.

In contrast, almost instantaneous transition from rolling to arrest can be observed for neutrophils only in an artificial situation where recombinant chemokine (f.e. CXCL1 or CXCL8) is injected in high concentration into the blood circulation of a living mouse (as reported f.e. by our group ²¹). In this scenario, neutrophils go into firm arrest within seconds.

However, to increase the temporal resolution of E-selectin-mediated S100A8/A9 release in human neutrophils and further addressing this issue, we performed S100A8/A9 release assays using shorter stimulation times (1min, 5min). Intriguingly, S100A8/A9 levels in the supernatant were already detectable after 1min of E-selectin stimulation and further increased with longer stimulation time (5min, 10min), demonstrating that E-selectin-mediated inflammasome activation and ensuing S100A8/A9 release occurs within minutes.

Reviewer Figure 3.2

Reviewer Figure 3.2 S100A8/A9 is released by human neutrophils in a time dependent manner. Isolated human neutrophils were stimulated with PBS (control) or E-selectin ($1\mu\text{g ml}^{-1}$) for 1min, 5min or 10min and S100A8/A9 levels in the supernatants was assessed by ELISA ($n \geq 3$ individual experiments; two-way ANOVA, Sidak's multiple comparison). Data is presented as mean \pm SEM. ns: not significant; *: $p < 0.05$; ***: $p < 0.001$.

Values from the 10min time point (control and E-selectin stimulation) are the same as in Figure 3g.

LUDWIG-
MAXIMILIANS-
UNIVERSITÄT
MÜNCHEN

WALTER-BRENDEL-ZENTRUM
BIOMEDIZINISCHES CENTRUM MÜNCHEN
INSTITUT FÜR KARDIOVASKULÄRE PHYSIOLOGIE UND PATHOPHYSIOLOGIE

Reviewer figure 3.4 is a mystery to me. If the pores are large enough to allow PI and S100 proteins to pass the membrane I would have expected saturation of the INDO-1 inside the cells. It is even of concern that the INDO-1 in its 'acid' form would also diffuse out of the cells along the same pores.

We apologize for not being clear here. In this assay, cells are loaded with Indo-1 AM and then washed to remove any extracellular dye, which might diffuse into the cell during stimulation.

In addition, stimulation with CXCL8 leads to a Ca^{2+} signal with a fast decay. However, CXCL8 does not induce the release of S100A8/A9, as shown by Morikis et al in the human system ¹⁶. Therefore, we also do not expect pore formation and diffusion through pores by CXCL8 stimulation.

Reviewer #4 (Remarks to the Author):

I would like to thank you for inviting me to review the revised manuscript and the review process. In this paper, the authors investigated the molecular mechanism of E-selectin-mediated signaling and demonstrated that E-selectin triggers Kv1.3 channel-mediated potassium efflux resulting in NLRP3 inflammasome activation.

Reviewer 3 raises very good points that are valid and are not adequately addressed in the response to reviewers letter. To be honest, I am quite surprised that this manuscript made it into the review process, because of the lack of novelty, methodological issues, and lack of citing existent literature.

We thank reviewer 4 for his/her comments on our manuscript. We strongly disagree with this reviewer that our findings are lacking novelty. This is the first report describing E-selectin-induced rapid activation of the NLRP3 inflammasome in neutrophils. In addition, we also show for the first time that E-selectin stimulation of neutrophils mediates rapid activation of the NLRP3 inflammasome in a Kv1.3-dependent fashion leading to transient Gasdermin D dependent pore formation with release of the alarmin S100A8/A9. None of these findings was described before.

My major concerns are:

1) The link between Kv1.3 and the activation of the NLRP3 is not novel, because this link has been already established (e.g. PMID: 32652099).

Indeed, Kv1.3 has been linked to inflammasome activation in microglia cells ³⁹. However, direct evidence is completely missing and only an association between inflammasome activation and the presence of Kv1.3 is shown. It is needless to say that almost any experimental setting where the inflammatory process is impaired will lead to a similar association. To our knowledge, there is no publication showing a direct link between Kv1.3 and inflammasome activation in neutrophils.

2) In accordance with reviewer 3, the authors ignore the existing literature on E-selectin-mediated signaling and claim that E-selectin engagement induces slow leukocyte rolling and arrest (page 4, line 21). This is not true as shown in many papers (PMID: 34435628; PMID: 29592875; PMID: 22511754; PMID: 20299514;

PMID: 20445017; PMID: 22021370; PMID: 17543554). Binding of E-selectin to its receptor on neutrophils induces the extended conformation of LFA-1 which mediates slow leukocyte rolling, but not arrest (PMID: 34435628; PMID: 29592875; PMID: 22511754; PMID: 20299514; PMID: 20445017; PMID: 22021370; PMID: 17543554). Leukocyte arrest requires the full activation of LFA-1 which is triggered by chemokines. This has also been shown in in vivo experiments. Blocking E-selectin alone does not reduce the number of adherent or transmigrated cells in vivo (PMID: 15466624; PMID: 17543554). These data do not go along with the data shown in Figure 6 as *Gsdmd*^{-/-} mice have a reduced leukocyte adhesion and transmigration, suggesting that the elimination of gasdermin D is also involved in other processes.

We thank the reviewer for his/her comment. In the present study, we did not intend to question the importance of chemokines in beta2 integrin activation in neutrophils. We agree with reviewer 4 that under in vivo conditions E-selectin downstream signaling cooperates with additional signaling (via chemokines, but also other signaling molecules) in mediating neutrophil full arrest. However, we disagree with some of the raised statements and therefore we would like to clarify some misunderstandings.

As already discussed in comment E1 of reviewer 3, the dogma, that "leukocyte arrest requires the full activation of LFA-1 which is triggered by chemokines" does to our understanding not completely cover the current picture of beta2 integrin activation. There is solid evidence from us and others showing that various mediators are able to activate beta2 integrins independent of chemokines as well, f.e. TNF- α ¹¹, amyloid- β ¹², TLR2 and TLR5 agonists¹³ and TLR4 agonists^{14, 15}. This is also demonstrated by Smith et al. (cited by reviewer 4, PMID: 15466624,³³) showing that both loss of CXCR2 or E-selectin alone do not affect TNF-alpha induced adhesion in the same cremaster muscle in vivo model demonstrating that E-selectin and CXCR2 have redundant and cooperative roles in facilitating firm adhesion.

A few years ago, we published that E-selectin stimulation induces S100A8/A9 release from neutrophils and that released S100A8/A9 in turn mediates integrin activation via TLR4 signaling¹⁴. As already stated in our response to reviewer 3 (please refer to E1), to our knowledge, this concept has not been refuted. In fact,

other groups confirmed the finding that S100A8/A9 and other TLR4 agonists are able to mediate beta2 integrin activation ^{15, 16} and to affect neutrophil adhesion ^{17, 18}. For further discussion, please refer to reviewer 3, section E1.

Hence, our model of E-selectin-induced S100A8/A9 release and subsequent TLR4 dependent integrin activation is not conflicting with all the published models, but an extension.

Regarding the findings in Figure 6 (Gsdmd^{-/-} mice IVM data), please refer to the next point (#3) of reviewer 4 for details.

3) In line with point 2, gasdermin D is expressed in different tissues. However, in figure 6 the authors used the full knock-out mouse. These experiments cannot exclude that the seen phenotype is caused by the lack of gasdermin D in other tissues.

We agree with reviewer 4 that the observed immune phenotype in Gsdmd^{-/-} mice (Figure 6, as well as in MCC950 treated WT mice, Supplementary Figure 6) in vivo might not exclusively be attributed to impaired inflammasome mediated S100A8/A9 release from neutrophils (which we never claimed) and we apologize for not stating it clear enough.

During an inflammatory process, genetic deletion of GSDMD (and pharmacological inhibition of the NLRP3 inflammasome) does not only directly affect neutrophil recruitment by inhibiting S100A8/A9 release, but likely also attenuate inflammasome activation in other cell types including tissue macrophages and endothelial cells leading to a reduction in neutrophil adhesion and extravasation. In the new version of the revised manuscript we state now that “genetic deletion of GSDMD or pharmacological inhibition of NLRP3 inflammasome activation in this in vivo model will most likely also attenuate inflammasome activation in cell types other than neutrophils, which might additionally contribute to the reduction in the inflammatory response observed in the in vivo setting where GSDMD is deleted or NLRP3 blocked by MCC950.” (page 15, line 8ff.)

Nevertheless, we are convinced that the increased rolling velocities in Gsdmd^{-/-} (and MCC950 treated) neutrophils compared to WT neutrophils in TNF- α stimulated postcapillary venules in the mouse cremaster are to a great extent a result of

LUDWIG-
MAXIMILIANS-
UNIVERSITÄT
MÜNCHEN

WALTER-BRENDEL-ZENTRUM
BIOMEDIZINISCHES CENTRUM MÜNCHEN
INSTITUT FÜR KARDIOVASKULÄRE PHYSIOLOGIE UND PATHOPHYSIOLOGIE

impaired S100A8/A9 release. Evidence to support this comes from our flow chamber assays where we perfused isolated neutrophils through flow chambers coated with E-selectin and ICAM-1 (thereby excluding any contribution of the endothelial or tissue/macrophage compartment) and found an increase in neutrophil rolling velocities in the absence of Gasdermin D or when pretreating the cells with MCC950.

4) It has been shown that E-selectin binding triggers different signaling pathways. As integrin activation occurs instantaneously after E-selectin engagement, it is not comprehensible how the gasdermin/NLRP3 pathway is involved in E-selectin-mediated slow leukocyte rolling.

Please refer to our response to reviewer 3 (C6).

In addition we would like to state, that in the present study, we did not intend to question the importance of E-selectin (or chemokines) on beta2 integrin activation in neutrophils and their role in E-selectin mediated slow rolling. Our focus was to elucidate E-selectin induced rapid NLRP3 inflammasome activation resulting in transient pore formation and the release of S100A8/A9 through these GSDMD pores.

Reference list

1. Mueller, H. *et al.* Tyrosine kinase Btk regulates E-selectin-mediated integrin activation and neutrophil recruitment by controlling phospholipase C (PLC) gamma2 and PI3Kgamma pathways. *Blood* **115**, 3118-3127 (2010).
2. Bittner, Z.A. *et al.* BTK operates a phospho-tyrosine switch to regulate NLRP3 inflammasome activity. *J Exp Med* **218** (2021).
3. Ito, M. *et al.* Bruton's tyrosine kinase is essential for NLRP3 inflammasome activation and contributes to ischaemic brain injury. *Nat Commun* **6**, 7360 (2015).
4. Liu, X. *et al.* Human NACHT, LRR, and PYD domain-containing protein 3 (NLRP3) inflammasome activity is regulated by and potentially targetable through Bruton tyrosine kinase. *J Allergy Clin Immunol* **140**, 1054-1067 e1010 (2017).
5. Hoss, F., Rolfes, V., Davanzo, M.R., Braga, T.T. & Franklin, B.S. Detection of ASC Speck Formation by Flow Cytometry and Chemical Cross-linking. *Methods Mol Biol* **1714**, 149-165 (2018).
6. Chauhan, D. *et al.* GSDMD drives canonical inflammasome-induced neutrophil pyroptosis and is dispensable for NETosis. *EMBO Rep*, e54277 (2022).
7. Chen, K.W. *et al.* Noncanonical inflammasome signaling elicits gasdermin D-dependent neutrophil extracellular traps. *Sci Immunol* **3** (2018).
8. Sollberger, G. *et al.* Gasdermin D plays a vital role in the generation of neutrophil extracellular traps. *Sci Immunol* **3** (2018).
9. Monteleone, M. *et al.* Interleukin-1beta Maturation Triggers Its Relocation to the Plasma Membrane for Gasdermin-D-Dependent and -Independent Secretion. *Cell Rep* **24**, 1425-1433 (2018).
10. Champaiboon, C., Sappington, K.J., Guenther, B.D., Ross, K.F. & Herzberg, M.C. Calprotectin S100A9 calcium-binding loops I and II are essential for keratinocyte resistance to bacterial invasion. *J Biol Chem* **284**, 7078-7090 (2009).
11. Bromberger, T., Klapproth, S., Sperandio, M. & Moser, M. Humanized beta2 Integrin-Expressing Hoxb8 Cells Serve as Model to Study Integrin Activation. *Cells* **11** (2022).

12. Zenaro, E. *et al.* Neutrophils promote Alzheimer's disease-like pathology and cognitive decline via LFA-1 integrin. *Nat Med* **21**, 880-886 (2015).
13. Chung, K.J. *et al.* A novel pathway of rapid TLR-triggered activation of integrin-dependent leukocyte adhesion that requires Rap1 GTPase. *Mol Biol Cell* **25**, 2948-2955 (2014).
14. Pruenster, M. *et al.* Extracellular MRP8/14 is a regulator of beta2 integrin-dependent neutrophil slow rolling and adhesion. *Nat Commun* **6**, 6915 (2015).
15. Uhl, B. *et al.* Aged neutrophils contribute to the first line of defense in the acute inflammatory response. *Blood* **128**, 2327-2337 (2016).
16. Morikis, V.A. *et al.* Selectin catch-bonds mechanotransduce integrin activation and neutrophil arrest on inflamed endothelium under shear flow. *Blood* **130**, 2101-2110 (2017).
17. Sprenkeler, E.G.G. *et al.* S100A8/A9 Is a Marker for the Release of Neutrophil Extracellular Traps and Induces Neutrophil Activation. *Cells* **11** (2022).
18. Ryckman, C., Vandal, K., Rouleau, P., Talbot, M. & Tessier, P.A. Proinflammatory activities of S100: proteins S100A8, S100A9, and S100A8/A9 induce neutrophil chemotaxis and adhesion. *J Immunol* **170**, 3233-3242 (2003).
19. Yago, T. *et al.* E-selectin engages PSGL-1 and CD44 through a common signaling pathway to induce integrin alphaLbeta2-mediated slow leukocyte rolling. *Blood* **116**, 485-494 (2010).
20. Smith, M.L., Olson, T.S. & Ley, K. CXCR2- and E-selectin-induced neutrophil arrest during inflammation in vivo. *J. Exp. Med* **200**, 935-939 (2004).
21. Frommhold, D. *et al.* Sialyltransferase ST3Gal-IV controls CXCR2-mediated firm leukocyte arrest during inflammation. *J Exp Med* **205**, 1435-1446 (2008).
22. Schweitzer, K.M. *et al.* Constitutive expression of E-selectin and vascular cell adhesion molecule-1 on endothelial cells of hematopoietic tissues. *Am J Pathol* **148**, 165-175 (1996).
23. Sreeramkumar, V. *et al.* Coordinated and unique functions of the E-selectin ligand ESL-1 during inflammatory and hematopoietic recruitment in mice. *Blood* **122**, 3993-4001 (2013).

24. Wiggs, B.R. *et al.* Contributions of capillary pathway size and neutrophil deformability to neutrophil transit through rabbit lungs. *J Appl Physiol* (1985) **77**, 463-470 (1994).
25. Mizgerd, J.P. *et al.* Selectins and neutrophil traffic: Margination and *Streptococcus pneumoniae*-induced emigration in murine lungs. *J. Exp. Med* **184**, 639-645 (1996).
26. Fedosov, D.A., Peltomaki, M. & Gompper, G. Deformation and dynamics of red blood cells in flow through cylindrical microchannels. *Soft Matter* **10**, 4258-4267 (2014).
27. Melder, R.J., Yuan, J., Munn, L.L. & Jain, R.K. Erythrocytes enhance lymphocyte rolling and arrest in vivo. *Microvasc Res* **59**, 316-322 (2000).
28. Fay, M.E. *et al.* Cellular softening mediates leukocyte demargination and trafficking, thereby increasing clinical blood counts. *Proc Natl Acad Sci U S A* **113**, 1987-1992 (2016).
29. Forlow, S.B. *et al.* Severe inflammatory defect and reduced viability in CD18 and E-selectin double-mutant mice. *J Clin. Invest* **106**, 1457-1466 (2000).
30. Robinson, S. *et al.* Multiple, targeted deficiencies in selectins reveal a predominant role for P-selectin in leukocyte recruitment. *PNAS* **96**, 11452-11457 (1999).
31. Kunkel, E.J. & Ley, K. Distinct phenotype of E-selectin deficient mice: E-selectin is required for slow leukocyte rolling in vivo. *Circ. Res* **79**, 1196-1204 (1996).
32. Jung, U., Bullard, D.C., Tedder, T.F. & Ley, K. Velocity differences between L- and P-selectin-dependent neutrophil rolling in venules of mouse cremaster muscle in vivo. *Am J Physiol* **271**, H2740-2747 (1996).
33. Ley, K. *et al.* Sequential contribution of L- and P-selectin to leukocyte rolling in vivo. *J Exp Med* **181**, 669-675 (1995).
34. Mayadas, T.N., Johnson, R.C., Rayburn, H., Hynes, R.O. & Wagner, D.D. Leukocyte rolling and extravasation are severely compromised in P selectin-deficient mice. *Cell* **74**, 541-554 (1993).
35. Arbones, M.L. *et al.* Lymphocyte homing and leukocyte rolling and migration are impaired in L-selectin-deficient mice. *Immunity* **1**, 247-260 (1994).
36. Dunne, J.L., Goobic, A.P., Acton, S.T. & Ley, K. A novel method to analyze leukocyte rolling behavior in vivo. *Biol. Proced. Online* **6**, 173-179 (2004).

37. Sperandio, M. *et al.* Severe impairment of leukocyte rolling in venules of core 2 glucosaminyltransferase-deficient mice. *Blood* **97**, 3812-3819 (2001).
38. Yang, W.H., Nussbaum, C., Grewal, P.K., Marth, J.D. & Sperandio, M. Coordinated roles of ST3Gal-VI and ST3Gal-IV sialyltransferases in the synthesis of selectin ligands. *Blood* **120**, 1015-1026 (2012).
39. Ma, D.C., Zhang, N.N., Zhang, Y.N. & Chen, H.S. Kv1.3 channel blockade alleviates cerebral ischemia/reperfusion injury by reshaping M1/M2 phenotypes and compromising the activation of NLRP3 inflammasome in microglia. *Exp Neurol* **332**, 113399 (2020).

Decision Letter, second revision:

2nd Jun 2023

Dear Professor Sperandio,

Your Article, "E-selectin-mediated rapid NLRP3 inflammasome activation regulates S100A8/A9 release from neutrophils via transient gasdermin D pore formation" has now been seen by referees.

Reviewers 2 and 3 are the original reviewers. Reviewer 1 was already happy so was not consulted again. Reviewer 4 declined to re-review. As you know, reviewer 3 was and still is fairly negative about the paper, as you will see below, so we recruited yet another reviewer (number 5) who is a neutrophil expert. This reviewer was asked to mediate and you will be pleased to know that they mostly side with the authors over the negative comments of reviewer 3 and 4. However they also raise some important concerns mostly about missing controls.

As such, we have decided to offer you the chance for a final revision to provide these controls and importantly to also edit your text to ensure that the conclusions are not overstated and that you insert suitable caveats regarding the physiological relevance of the mechanisms being described (to address reviewer 3 complaints). If you do this, then we would consider that the scientific community can decide as to whether reviewer 3 is correct or not. If the paper makes it to publication, we would also expect that you opt in to publish the peer review history for full transparency of the conflicting opinions about these data.

If you choose to revise your manuscript taking into account the reviewer and editor comments, please highlight all changes in the manuscript text file in Microsoft Word format.

- * Include a "Response to referees" document detailing, point-by-point, how you addressed each referee comment. If no action was taken to address a point, you must provide a compelling argument. This response will be sent back to the referees along with the revised manuscript.
- * If you have not done so already please begin to revise your manuscript so that it conforms to our Article format instructions at <http://www.nature.com/ni/authors/index.html>. Refer also to any guidelines provided in this letter.
- * Include a revised version of any required reporting checklist. It will be available to referees (and, potentially, statisticians) to aid in their evaluation if the manuscript goes back for peer review. A revised checklist is essential for re-review of the paper.

The Reporting Summary can be found here:

[REDACTED]

If you wish to submit a suitably revised manuscript we would hope to receive it within 6 months. If you cannot send it within this time, please let us know. We will be happy to consider your revision so long as nothing similar has been accepted for publication at Nature Immunology or published elsewhere.

Nature Immunology is committed to improving transparency in authorship. As part of our efforts in this direction, we are now requesting that all authors identified as 'corresponding author' on published papers create and link their Open Researcher and Contributor Identifier (ORCID) with their account on the Manuscript Tracking System (MTS), prior to acceptance. ORCID helps the scientific community achieve unambiguous attribution of all scholarly contributions. You can create and link your ORCID from the home page of the MTS by clicking on 'Modify my Springer Nature account'. For more information please visit www.springernature.com/orcid.

Thank you for the opportunity to review your work.

Sincerely,

Nick Bernard, PhD
Senior Editor
Nature Immunology

Reviewers' Comments:

Reviewer #2:

Remarks to the Author:

The authors have performed another very thorough round of review, and in so doing have addressed my former minor points and greatly increased the impact of their manuscript (particularly with further mechanistic insight regarding BTK). My congratulations to the authors on this exciting study, which I now recommend for publication.

This reviewer notes the comments on Reviewers 3 and 4- experts in neutrophil rolling and trafficking - regarding novelty. Whilst I cannot comment on the current study's novelty with respect to neutrophil dynamics, I would like to emphasise that the current study is extremely novel in the inflammasome field. The finding that e-selectin can trigger NLRP3 signalling in neutrophils, leading to vital cell release of S100s is an entirely novel usage of the NLRP3 inflammasome pathway and greatly extends our understanding of the physiological functions of NLRP3.

Reviewer #3:

Remarks to the Author:

The article has improved, but still lacks sufficient in vivo evidence for a seemingly highly inefficient mechanism.

The overall concept in the field is that rolling of neutrophils on selectins is slowing the cells down facilitating firm adhesion mediated by integrins activated by inflammatory mediators. "No mediators no slowing down" and cells can detach and go elsewhere without an necessary effect on the fitness of the cells. The authors claim that this hypothesis is wrong and come up with a complex alternative model/mechanism relying in the end on cell death at the vascular surface (inflammasome activation, pore formation and release of cytosolic proteins such as S100/A8/A9 proteins) as this is the consequence of leaky cells. As also commented on in the first review this seems a very inefficient mechanism. On the other hand, this seems testable in vivo as rolling cells on activated endothelium should be accompanied by leaky neutrophils that can be visualized by Ca²⁺ dyes, DNA dyes etc. Although the individual experiments are performed well, they do not support an overall logic of the model as this is highly inefficient.

Experiments with soluble E-selectin and cells in suspension as 'surrogate' for cells rolling on immobilized E-selectin is just not good enough. There is no effects of putative cross-linking of e-selectin ligands on neutrophils and an absence of shear-stress that might modulate cellular functions.

The issue that leukocytosis is found in LAD2-(human) patients whereas in mouse models this is absent has still not been addressed. This is important as this is quite a fundamental process when data are translated from mouse to the human in vivo situation. Also the comments that this might be caused by differences in mouse strains and animal facilities is disturbing.

The arguments (C6) raised in response to the statement regarding priming of neutrophil responses by chemokines are not to the point. These are in fact almost instantaneous and this concerns many neutrophil responses (degranulation, activation of the NADPH-oxidase, lipid release etc). I am not

disagreeing with Ley's data on the 270 microns tracks in vivo, but I raised the issue why there is this delay when the signaling of chemokines/chemotaxin receptors is so fast.

The point of INDO-1 diffusing out of the cells and Ca²⁺ diffusing in when pores are formed is completely missed by the authors. How can any reliable Ca²⁺ transients be measured when the cells are leaky.

Reviewer #5:

Remarks to the Author:

The work E-selectin-mediated rapid NLRP3 inflammasome activation...by Pruenster et al., provides a new mechanistic understanding of how the alarmin S100A8, important in neutrophil function, is released from neutrophils during transmigration. Although the concept that E-selectin stimulates S100A8 release was demonstrated before by the authors, and the knowledge that this alarmin is secreted from the cytosol independently from granule mobilization was previously established by others, the current work's novelty rests on the elucidation of a new mechanisms in which E-selectin transiently activates the NLRP3 inflammasome to induce S100A8 through Gasdermin-D pores, in a rapid manner.

Questions were raised by R3 and R4 on the apparent lack of agreement of this study, with previous work showing that chemokines are necessary, together with E-selectin, as a coordinated and double-hit stimulation, to fully activate LFA1. In this reviewer's opinion those concerns were alleviated by the authors' response, as they explain that double-hit stimulation can be achieved through non-cytokine mechanisms, including S100A8's own autocrine effect on TLRs.

Furthermore, the authors demonstrate that in vitro E-selectin can induce S100A8 in a timely manner without cytokine stimulation. This new figure, which has been included as "Reviewer Figure 3.2", should be included within the manuscript. The authors also addressed a concern about granule mobilization, clearly showing that this is not the case, but the data was again included as reviewer figure only (Reviewer Figure 3.1 a-c) but should instead be part of the manuscript (the points were addressed but please include figures in the main manuscript).

The authors also now performed new mass spectrometry analysis of the secretome released in response to E-selectin stimulation to support their conclusions. These experiments are important and a complete list of peptides identified should be included within the manuscript or supplemental data.

The question about mouse vs human neutrophils was also properly addressed.

In conclusion to this part of the study, it is perceived that the authors have been responsive by clarifying their interpretation of the neutrophil biological process with reasonable arguments and new data.

However, I would like to highlight some points where the manuscript could be improved to better support their conclusions. One of the main points, if not the main point, of the manuscript is to demonstrate that E-selectin activates the NLRP3 inflammasome, transiently, to allow for pore forming Gasdermin D to permit S100A8 exit. In my opinion, it would be in the best interest of the manuscript and the authors to make these data stronger. One of the weaker pieces of data are the immunoblots which need improvement for the following reasons: To show that caspase 1 is activated, the authors do a Western blot for caspase 1 p20, in the cell supernatants. A closer look at the uncropped membrane shows many bands, much more stronger than p20 and of higher molecular weight are also increased in E-selectin-treated cells in this lane, suggesting that either a) these are non-specific bands and the amount of sample included in the lane corresponding to E-selectin stimulated cells is much more than in other lanes (therefore no proof of activation of caspase 1); or b) all these bands

are specific for truncations of caspase 1 and, in this case, this sample simply has more caspase, not more caspase 1p20 activation. Whichever is the case, the blot also brings attention to the lack of selectivity of the secretion mechanism and raises the question that, if caspase1p20 exits the cells, it would not be able to activate intracellular Gasdermin. Furthermore, although the caspase activation probe (FLICA) is shown to be increased in E-selectin treated samples, I strongly suggest that a) the author include the Caspase1-KO as negative control in the FLICA assay and that they quantify Caspase 1p20 in the total cell lysate, pellet+supernatant.

The single image of immunofluorescence from which the authors derive the conclusion that Gasdermin-D form pores at the PM, seems to show puncta that resembles granules or vesicles near to, but not necessarily at the plasma membrane. Translocation of Gasdermin D to granules has been demonstrated previously by Karmakar et al, using super-resolution techniques that would be very useful here to determine to which degree or if Gasdermin D is detectable at the PM. The plasma membrane should have been labeled with a marker for identification.

The antibody used to detect Gasdermin-D-NT in this immunofluorescence is mentioned but not described. The experiment is central, and the anti-truncated GSDMD-NT must be clearly described. The GAPDH equal loading could be improved in 2a and 4d.

The PI experiment shown in 5J K is very interesting, supports transient opening, and to address previous concerns, I do not have any problem with this figure.

Overall, this is a novel story. Some major points have been addressed. I added a few comments on pieces of data that should be improved to support the findings.

Author Rebuttal, second revision:

See inserted PDF

Reviewers' Comments:

Reviewer#2:

Remarks to the Author:

The authors have performed another very thorough round of review, and in so doing have addressed my former minor points and greatly increased the impact of their manuscript (particularly with further mechanistic insight regarding BTK). My congratulations to the authors on this exciting study, which I now recommend for publication.

This reviewer notes the comments on Reviewers 3 and 4- experts in neutrophil rolling and trafficking - regarding novelty. Whilst I cannot comment on the current study's novelty with respect to neutrophil dynamics, I would like to emphasise that the current study is extremely novel in the inflammasome field. The finding that e-selectin can trigger NLRP3 signalling in neutrophils, leading to vital cell release of S100s is an entirely novel usage of the NLRP3 inflammasome pathway and greatly extends our understanding of the physiological functions of NLRP3.

We thank reviewer 2 for his encouraging comments and supporting our work for publication in Nature Immunology

Reviewer #3:

Remarks to the Author:

The article has improved, but still lacks sufficient in vivo evidence for a seemingly highly inefficient mechanism.

The overall concept in the field is that rolling of neutrophils on selectins is slowing the cells down facilitating firm adhesion mediated by integrins activated by inflammatory mediators. “No mediators no slowing down” and cells can detach and go elsewhere without an necessary effect on the fitness of the cells. The authors claim that this hypothesis is wrong and come up with a complex alternative model/mechanism relying in the end on cell death at the vascular surface (inflammasome activation, pore formation and release of cytosolic proteins such as S100/A8/A9 proteins) as this is the consequence of leaky cells. As also commented on in the first review this seems a very inefficient mechanism. On the other hand, this seems testable in vivo as rolling cells on activated endothelium should be accompanied by leaky neutrophils that can be visualized by Ca²⁺ dyes, DNA dyes etc. Although the individual experiments are performed well, they do not support an overall logic of the model as this is highly inefficient.

We thank the reviewer for her/his comments.

We would like to emphasize once again that we did not claim in the current manuscript that the hypothesis “selectins slowing down the cells facilitating firm adhesion mediated by integrins” is wrong. In addition (and very important for the concept of our entire study), we do not state at all that neutrophils undergo cell death after E-selectin induced NLRP3 inflammasome activation within the vasculature. This is clearly stated not only in the results part but also at the end of the abstract, at the end of the introduction, at the end of the discussion and in the legend of the graphical abstract!

In contrast, we demonstrate that cells do not die upon E-selectin stimulation (please refer to Fig. 5g-i) and propose cell survival of E-selectin stimulated cells involving activation of the ESCRT membrane repair machinery (please refer to Fig. 5l and m).

Experiments with soluble E-selectin and cells in suspension as ‘surrogate’ for cells rolling on immobilized E-selectin is just not good enough. There is no effects of putative cross-linking of e-selectin ligands on neutrophils and an absence of shear-stress that might modulate cellular functions.

We agree with the reviewer that investigating E-selectin induced inflammasome activation in a human system in vivo would be optimal. However, due to technical limitations and as already highlighted in the previous point-to-point response, we used in vivo and in vitro models to mimic E-selectin interaction with rolling neutrophils. We performed experiments using soluble E-selectin and experiments using

surface bound E-selectin under shaking conditions mimicking inflamed endothelium and leukocyte rolling (coated glass capillaries and dishes). We are aware of limitations of in vitro models, which is a general issue for studying neutrophil activation (i.e. stimulating cells with soluble chemokine instead of GAG-bound chemokine).

The issue that leukocytosis is found in LAD2-(human) patients whereas in mouse models this is absent has still not been addressed. This is important as this is quite a fundamental process when data are translated from mouse to the human in vivo situation. Also the comments that this might be caused by differences in mouse strains and animal facilities is disturbing.

In our opinion, citing LAD2 mouse vs. human and comparing this to our study is rather far-fetched. We are quite aware that many immunological studies show differences between human and mouse. In our study, we have conducted numerous experiments in the mouse and human systems and concerning the effects of E-selectin-triggered GSDMD pore formation we find very similar results for human and mouse neutrophils.

The arguments (C6) raised in response to the statement regarding priming of neutrophil responses by chemokines are not to the point. These are in fact almost instantaneous and this concerns many neutrophil responses (degranulation, activation of the NADPH-oxidase, lipid release etc). I am not disagreeing with Ley's data on the 270 microns tracks in vivo, but I raised the issue why there is this delay when the signaling of chemokines/chemotaxin receptors is so fast.

We agree that priming of neutrophil responses with chemokine are not the point and the focus of the current study. We in addition agree that chemokine signaling is fast and induces neutrophil responses almost instantaneously. We disagree on a delay of E-selectin induced signaling. We have shown that E-selectin signaling leads to rapid inflammasome activation (and slow leukocyte rolling) occurring very rapidly after engagement and leading to S100A8/A9 release within minutes (please refer to Supplementary Fig. 3c).

The point of INDO-1 diffusing out of the cells and Ca²⁺ diffusing in when pores are formed is completely missed by the authors. How can any reliable Ca²⁺ transients be measured when the cells are leaky.

INDO-1 AM enters the cell where it is cleaved by intracellular esterases into INDO-1, which is not cell permeable. Although we did not formally prove whether some INDO-1 might leave the cell through GSDMD pores, it is very unlikely that all INDO-1 leaves the cell even before Ca²⁺ is entering the cell. In addition, we want to state again that E-selectin dependent GSDMD pore formation is a very limited and transient process in neutrophils. This is also reflected by the fact that we only see a mild change in membrane potential after stimulation with E-selectin (please refer to Supplementary Fig. 4a-c).

Reviewer #5:

Remarks to the Author:

The work E-selectin-mediated rapid NLRP3 inflammasome activation....by Pruenster et al., provides a new mechanistic understanding of how the alarmin S100A8, important in neutrophil function, is released from neutrophils during transmigration. Although the concept that E-selectin stimulates S100A8 release was demonstrated before by the authors, and the knowledge that this alarmin is secreted from the cytosol independently from granule mobilization was previously established by others, the current work's novelty rests on the elucidation of a new mechanisms in which E-selectin transiently activates the NLRP3 inflammasome to induce S100A8 through Gasdermin-D pores, in a rapid manner.

Questions were raised by R3 and R4 on the apparent lack of agreement of this study, with previous work showing that chemokines are necessary, together with E-selectin, as a coordinated and double-hit stimulation, to fully activate LFA1. In this reviewer's opinion those concerns were alleviated by the authors' response, as they explain that double-hit stimulation can be achieved through non-cytokine mechanisms, including S100A8's own autocrine effect on TLRs.

We thank the reviewer for her/his positive and motivating comments and for recognizing the concept and the novelty of the current study.

Furthermore, the authors demonstrate that in vitro E-selectin can induce S100A8 in a timely manner without cytokine stimulation. This new figure, which has been included as "Reviewer Figure 3.2", should be included within the manuscript. The authors also addressed a concern about granule mobilization, clearly showing that this is not the case, but the data was again included as reviewer figure only (Reviewer Figure 3.1 a-c) but should instead be part of the manuscript (the points were addressed but please include figures in the main manuscript).

We thank the reviewer for this valuable comment and as suggested, we included the time series of S100A8/A9 release (earlier Reviewer Fig. 3.2) into the revised version of the manuscript (Supplementary Fig. 3c). In addition, we included the granule release data (mass spectrometry analysis and flow cytometry data, earlier Reviewer Fig. 3.1) into the revised version of the manuscript (Supplementary Fig. 3g-m).

The authors also now performed new mass spectrometry analysis of the secretome released in response to E-selectin stimulation to support their conclusions. These experiments are important and a complete list of peptides identified should be included within the manuscript or supplemental data.

We thank the reviewer for this valuable comment. We apologize for not being clear here. A complete list of peptides identified has been deposited to the ProteomeXchange Consortium via the PRIDE partner repository with the dataset identifier PXD041652, as stated in the “reporting summary /availability of data” provided by nature portfolio.

The question about mouse vs human neutrophils was also properly addressed.

In conclusion to this part of the study, it is perceived that the authors have been responsive by clarifying their interpretation of the neutrophil biological process with reasonable arguments and new data.

However, I would like to highlight some points where the manuscript could be improved to better support their conclusions. One of the main points, if not the main point, of the manuscript is to demonstrate that E-selectin activates the NLRP3 inflammasome, transiently, to allow for pore forming Gasdermin D to permit S100A8 exit. In my opinion, it would be in the best interest of the manuscript and the authors to make these data stronger. One of the weaker pieces of data are the immunoblots which need improvement for the following reasons: To show that caspase 1 is activated, the authors do a Western blot for caspase 1 p20, in the cell supernatants. A closer look at the uncropped membrane shows many bands, much more stronger than p20 and of higher molecular weight are also increased in E-selectin-treated cells in this lane, suggesting that either a) these are non-specific bands and the amount of sample included in the lane corresponding to E-selectin stimulated cells is much more than in other lanes (therefore no proof of activation of caspase 1); or b) all these bands are specific for truncations of caspase 1 and, in this case, this sample simply has more caspase, not more caspase 1p20 activation. Whichever is the case, the blot also brings attention to the lack of selectivity of the secretion mechanism and raises the question that, if caspase1p20 exits the cells, it would not be able to activate intracellular Gasdermin. Furthermore, although the caspase activation probe (FLICA) is shown to be increased in E-selectin treated samples, I strongly suggest that a) the author include the Caspase1-KO as negative control in the FLICA assay and that they quantify Caspase 1p20 in the total cell lysate, pellet+supernatant.

We thank the reviewer for this very important comment. We apologize for not being clear in explaining the procedure of western blot analysis.

To analyze caspase-1 cleavage upon E-selectin stimulation, we stimulated 1×10^7 cells per condition with E-selectin or PBS (or as indicated) and stopped the reaction by addition of ice-cold HBSS followed by centrifugation (5min at 300g). Supernatants were separated from the cell pellets and the cells were lysed and homogenized using modified RIPA lysis buffer (100 μ l per 10^6 cells). Protein extraction from

supernatants was performed with either chloroform-methanol or trichloroacetat precipitation. The precipitate was then resuspended in 30µl of modified RIPA lysis buffer and SDS PAGE gels were loaded with the entire volume. In addition, 30µl of cell lysate per condition was resolved by SDS-PAGE gel electrophoresis, too. In line with literature ¹, we analyzed the amount of cleaved caspase-1 in the supernatants. To account for potential variations in cell numbers among the respective samples, we normalized the cleaved caspase-1 signal from the supernatants to the GAPDH signal from the corresponding cell lysate.

We agree with the reviewer that GSDMD pore formation requires active caspase-1 within the cells. Therefore, we quantified caspase-1 cleavage in cell lysates of E-selectin vs PBS treated human neutrophils. In line with the increase of cleaved caspase-1 in the supernatants of E-selectin versus PBS stimulated human neutrophils, we detected a significant increase of cleaved caspase-1 (caspase-1 p20/p22) within the cell lysates. This supports our hypothesis that caspase-1 activation leads to GSDMD pore formation thereby facilitating s100A8/A9 release. We included this important observation into the revised version of the manuscript (Supplementary Figure 2a).

In addition, we performed the FAM-FLICA assay using the capase-1 inhibitor VX-765 (as the initial FLICA experiments have been carried out using human neutrophils, we decided not to change species using Casp-1^{-/-} mice). We did not detect active caspase-1 in VX-765 pretreated and E-selectin stimulated human neutrophils. We included these experiments into the revised version of the manuscript (Supplementary Fig. 2b and c).

The single image of immunofluorescence from which the authors derive the conclusion that Gasdermin-D form pores at the PM, seems to show puncta that resembles granules or vesicles near to, but not necessarily at the plasma membrane. Translocation of Gasdermin D to granules has been demonstrated previously by Karmakar et al, using super-resolution techniques that would be very useful here to determine to which degree or if Gasdermin D is detectable at the PM. The plasma membrane should have been labeled with a marker for identification.

We thank the reviewer for this important point. As suggested, we performed STED microscopy and detected GSDMD-NT localized at the membrane (stained with WGA) of E-selectin stimulated neutrophils. We included these experiments into the revised version of the manuscript (Supplementary Fig. 2d and e).

The antibody used to detect Gasdermin-D-NT in this immunofluorescence is mentioned but not described. The experiment is central, and the anti-truncated GSDMD-NT must be clearly described.

We thank the reviewer for this comment. As suggested, we included a short description of the GSDMD-NT antibody in the main text of the manuscript (page 7, lane 15 ff).

The GAPDH equal loading could be improved in 2a and 4d.

We thank the reviewer for this comment. Please see our comment on the western blot method above. We used the same amount of lysed cells in all conditions and not the same amount of protein. Finally, we normalized all samples (active caspase-1 in supernatants, GSDMD-NT and active caspase-1 in cell lysates) to the corresponding GAPDH signal.

The PI experiment shown in 5J K is very interesting, supports transient opening, and to address previous concerns, I do not have any problem with this figure.

Overall, this is a novel story. Some major points have been addressed. I added a few comments on pieces of data that should be improved to support the findings.

1. Chauhan, D. *et al.* GSDMD drives canonical inflammasome-induced neutrophil pyroptosis and is dispensable for NETosis. *EMBO Rep*, e54277 (2022).

Decision Letter, third revision:

14th Aug 2023

Dear Dr. Sperandio,

Thank you for submitting your revised manuscript "E-selectin-mediated rapid NLRP3 inflammasome activation regulates S100A8/A9 release from neutrophils via transient gasdermin D pore formation" (NI-A31849D). It has now been seen only by the mediating reviewer #5 again and their comments are below.

Just to remind you, unusually we had a total of 5 reviewers over the course of the revisions to this manuscript. The inflammasome reviewers 1 and 2 were previously happy so they were not consulted again. The neutrophil rolling adhesion reviewer 3 was not keen in an early stage of the manuscript and a mediating reviewer 4 was in agreement with them. However, as mediating reviewer 4 declined to re-review this paper we had to seek an alternative mediating reviewer who happens to be much more favourable of the paper and we are happy to say that their most recent re-review here is very positive. As such, we'll be happy in principle to publish it in Nature Immunology, pending minor revisions to comply with our editorial and formatting guidelines.

PLEASE NOTE: given the complexity of the issues debated between yourself and reviewer 3/4 and reviewer 5, we have decided that it is critical that you 'opt-in' to publishing the transparent peer reviews so that the wider scientific community can see the debate for themselves and make up their own minds about the data. Please ensure you check this option when resubmitting through the online system. Ordinarily, transparent peer review is something optional for the authors to choose or not, but here we do not feel comfortable publishing your article without this discussion made accessible to the public. We hope you understand this position, but if this is a concern please do contact me, as otherwise we are now pushing on with our editorial checks as outlined below.

We will now perform detailed checks on your paper and will send you a checklist detailing our editorial and formatting requirements in about a week. Please do not upload the final materials and make any revisions until you receive this additional information from us.

If you had not uploaded a Word file for the current version of the manuscript, we will need one before beginning the editing process; please email that to immunology@us.nature.com at your earliest convenience.

Thank you again for your interest in Nature Immunology Please do not hesitate to contact me if you have any questions.

Sincerely,

Nick Bernard, PhD
Senior Editor
Nature Immunology

Reviewer #5 (Remarks to the Author):

The authors have been responsive and adequately addressed my previous comments. Importantly, they include new and convincing imaging analysis of the transient translocation of GSDMD to the plasma membrane. Personally, I would have included these data in the main body of the manuscript; however, the data is appropriately described in the text and therefore the issue of whether GSDMD transiently translocate to the PM has been addressed and quantified. Other pieces of data that were originally included as "reviewer only" data are now included in the Supplementary data as I requested, and the data are properly described in the text. A control for the Flica experiment was also added to the manuscript as requested. In my opinion, this is a novel story and now all major points have been addressed.

Final Decision Letter:

Dear Dr. Sperandio,

I am delighted to accept your manuscript entitled "E-selectin-mediated rapid NLRP3 inflammasome activation regulates S100A8/A9 release from neutrophils via transient gasdermin D pore formation" for publication in an upcoming issue of Nature Immunology.

Over the next few weeks, your paper will be copyedited to ensure that it conforms to Nature Immunology style. Once your paper is typeset, you will receive an email with a link to choose the appropriate publishing options for your paper and our Author Services team will be in touch regarding any additional information that may be required.

Please note that *Nature Immunology* is a Transformative Journal (TJ). Authors may publish their research with us through the traditional subscription access route or make their paper immediately open access through payment of an article-processing charge (APC). Authors will not be required to make a final decision about access to their article until it has been accepted. [Find out more about Transformative Journals](https://www.springernature.com/gp/open-research/transformative-journals).

Your paper will be published online soon after we receive your corrections and will appear in print in the next available issue. Content is published online weekly on Mondays and Thursdays, and the embargo is set at 16:00 London time (GMT)/11:00 am US Eastern time (EST) on the day of publication. Now is the time to inform your Public Relations or Press Office about your paper, as they might be interested in promoting its publication. This will allow them time to prepare an accurate and satisfactory press release. Include your manuscript tracking number (NI-A31849E) and the name of the journal, which they will need when they contact our office.

About one week before your paper is published online, we shall be distributing a press release to news organizations worldwide, which may very well include details of your work. We are happy for your institution or funding agency to prepare its own press release, but it must mention the embargo date and Nature Immunology. Our Press Office will contact you closer to the time of publication, but if you or your Press Office have any enquiries in the meantime, please contact press@nature.com.

Also, if you have any spectacular or outstanding figures or graphics associated with your manuscript - though not necessarily included with your submission - we'd be delighted to consider them as candidates for our cover. Simply send an electronic version (accompanied by a hard copy) to us with a possible cover caption enclosed.

If you have not already done so, we strongly recommend that you upload the step-by-step protocols used in this manuscript to the Protocol Exchange. Protocol Exchange is an open online resource that

allows researchers to share their detailed experimental know-how. All uploaded protocols are made freely available, assigned DOIs for ease of citation and fully searchable through nature.com. Protocols can be linked to any publications in which they are used and will be linked to from your article. You can also establish a dedicated page to collect all your lab Protocols. By uploading your Protocols to Protocol Exchange, you are enabling researchers to more readily reproduce or adapt the methodology you use, as well as increasing the visibility of your protocols and papers. Upload your Protocols at www.nature.com/protocolexchange/. Further information can be found at www.nature.com/protocolexchange/about .

Please note that we encourage the authors to self-archive their manuscript (the accepted version before copy editing) in their institutional repository, and in their funders' archives, six months after publication. Nature Portfolio recognizes the efforts of funding bodies to increase access of the research they fund, and strongly encourages authors to participate in such efforts. For information about our editorial policy, including license agreement and author copyright, please visit www.nature.com/ni/about/ed_policies/index.html

Sincerely,

Nick Bernard, PhD
Senior Editor
Nature Immunology